# MPS: A Multi-Perspective Benchmark For Assessing Spurious Correlations in Text Classification

## Abstract

Text classification is especially susceptible to diverse spurious correlations, such as those related to word-frequency and concept-level patterns. Nevertheless, there is a lack of a comprehensive and standardized benchmark for evaluating the robustness of models against these spurious correlations. To address this crucial issue, we present MPS (**M**ulti - **P**erspective Benchmark For Assessing **S**purious Correlations in Text Classification). To construct this benchmark, we collect eight widely used text classification datasets and introduce five categories of spurious correlations for each of them, producing 40 variants of datasets for comprehensively evaluating spurious correlations in diverse settings. We then extensively evaluate various text classification models and state-of-the-art anti-spurious correlation methods on this benchmark, which uncovers the vulnerabilities of these models and methods to diverse spurious correlations. A follow-up comparative analysis on this benchmark is performed to assess the performance of these anti-spurious correlation methods and humans in diverse settings.

## 1 Introduction

In recent years, despite considerable advancements in models and machine learning algorithms for text classification tasks, empirical studies have revealed that due to the inherent nature of the learning process, models display significantly heightened sensitivity to various spurious correlations relative to the extraction and learning of essential features. This phenomenon can be largely explained by the imbalanced distribution of features within training datasets Chen et al. (2022); Sagawa et al. (2020); McCoy et al. (2019). Such spurious correlations substantially impede further performance improvements in text classification Tang et al. (2023).

To the best of our knowledge, research on spurious correlations commonly categorizes them into two primary types: statistics-based and concept-based. The statistics-based one is characterized by erroneous associations between specific words and a target label, which occurs when such words co-occur with unusually high frequency within that label. For example, in movie review datasets, terms like 'Spielberg' or 'Titanic' frequently demonstrate spurious correlations with positive sentiment Wang & Culotta (2020). One special subtype within this category is the negation-based spurious correlation. Unlike other words, negation words, e.g., 'not', typically exhibit global semantic impact on one entire text sample Joshi et al. (2022). As documented in Joshi et al. (2022); Williams et al. (2017), spurious correlations can occur between negation words and arbitrary classification labels. On the other hand, prior concept-level analysis Zhou et al. (2023) has uncovered a novel spurious correlation between a classification label and specific semantic features commonly shared across a broad corpus of text, which is termed concept-level spurious correlation.

Approaches to mitigating spurious correlations encompass traditional data augmentation methods (e.g., downsampling) as well as loss-function-level techniques—including distributionally robust optimization (DRO) regularization Sagawa et al. (2019), JTT loss design Liu et al. (2021), and embedding-space analysis Chew et al. (2023). However, these methods are typically evaluated under the influence of only one type. For example, datasets used in DRO focus exclusively on negation-type spurious correlations, while those for the NFL loss family address only concept-level spurious correlations. As a result, these methods lack comprehensive evaluation across diverse spurious cor-

| Categories | $\mathcal{A}$ | Example Text(Categories Label) | Explanation |
|---|---|---|---|
| SCS | sentence-level concepts | **Too small very small size I never use** (Clothing & Shoes Experience) | The label provides a comprehensive summarization of the sentence's core message. |
| CCS | core-word concepts | Not the best **herzog** perhaps, but unmistakably **herzog**.(Directors) | The label summarizes the essence of the core terms. |
| NBS | negation-based | I have **no** respect for the sodomy-loving pedophile priest club.(Nega) | The label denotes whether or not negative semantics are present in the text. |
| QBS | question-based | I came to a grocery store to pick up some stuff ,but guess who i saw**?**(Question) | The label denotes whether or not interrogative semantics are present in the text. |
| WFS | word-frequency based | **Love** @ Union Square, San Francisco(Love) | The label denotes the specific high-frequency word from the dataset that appears in the text. |

Table 1: **Spurious Correlation Categories and Explanation.** The table presents text examples for five categories of spurious correlations alongside their explanations. The red markings highlight which specific portions of the text correspond to each respective category.

relation types. Moreover, the datasets employed in these studies are constructed using conventional text classification corpora in an ad-hoc way rather than based on unified and standard protocols. As a broadly known benchmark for evaluating spurious correlation, WILDS Koh et al. (2021) can serve this purpose. Although this benchmark includes text classification datasets with spurious correlations based on statistics and concepts, many critical subtypes, such as the negation-based ons, are not included. This narrow focus hinders holistic evaluation across multiple spurious correlation types. Given the inherent difficulty in acquiring subgroup labels and the often limited sample size of certain subgroups, the worst-group accuracy measured under such conditions may fail to adequately reflect the extent to which spurious correlations affect model learning Liu et al. (2021).

To address this research gap, we propose the MPS benchmark, which integrates eight text classification datasets spanning diverse domains, varying label categories, and high popularity. On this benchmark, we evaluate five types of spurious correlations: SCS (Sentence-level Concept Spurious correlations), CCS (Core-word Concept Spurious correlations), NBS (Negation-Based Spurious correlations), QBS (Question-Based Spurious correlations), and WFS (Word-Frequency-based Spurious correlations). Among these, SCS and CCS represent concept-level spurious correlations, while NBS, QBS, and WFS fall under statistical-level spurious correlations. Notably, QBS is a novel spurious correlation type first introduced in MPS that has not been previously studied. Experimental results on MPS further demonstrate that existing mitigation methods consistently underperform in addressing QBS-type spurious correlations. Examples illustrating the impact of these five categories of spurious correlations on model learning, along with detailed explanations of each type, are provided in Appendix Figure 2 and Table 1. Furthermore, the dataset construction process based on MPS can effectively establish an imbalanced distribution between subgroups and task labels to induce spurious correlations. The worst-group accuracy, computed directly from task labels, serves as an effective metric for assessing the robustness of models against such spurious correlations.

Comprehensive testing and analysis of diverse models and anti-spurious correlations methods on the MPS dataset reveal that these models and methods struggle to mitigate the adverse effects induced by multiple spurious correlation types during learning. Additionally, comparisons with human-level performance in text classification tasks indicate significant potential for improving models' ability to resolve spurious correlations in such tasks. In conclusion, our paper makes the following contributions:

· To the best of our knowledge, we present the first comprehensive benchmark, MPS, for evaluating diverse spurious correlations in text classification tasks.
· We conducted extensive evaluations using MPS on a wide range of existing models and anti-spurious correlation methods, revealing their vulnerabilities to the diverse spurious correlations.
· We further perform human evaluations using MPS to compare the ability of humans and existing methods in dealing with spurious correlations.
· Based on MPS, we have developed a novel framework for evaluating spurious correlations in text classification tasks. This approach effectively circumvents certain limitations inherent in traditional evaluation metrics, such as the issue of scarce subgroup samples.

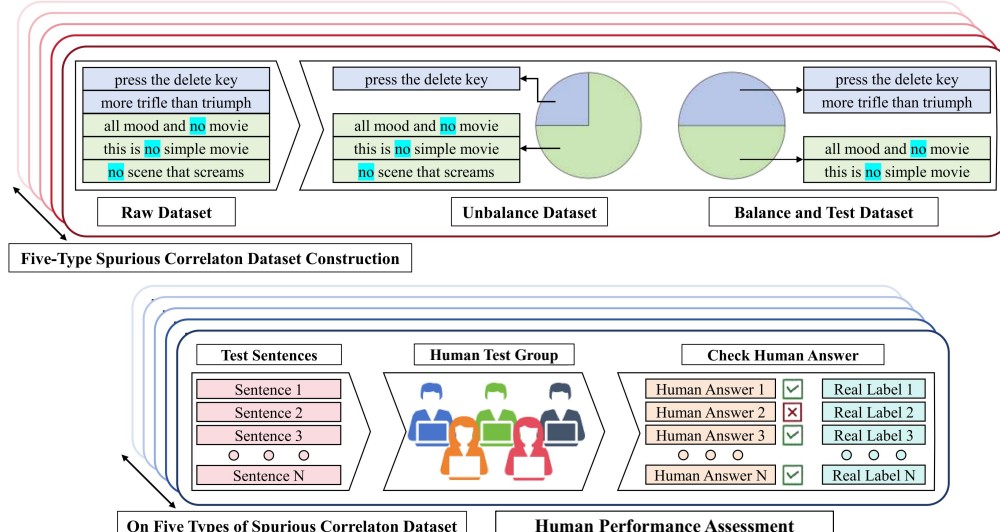

Figure 1: **MPS Benchmark Construction Process and Human Test Workflow Diagram.** To illustrate, we use negation-based spurious correlations as an example. First, we detect instances containing negation semantics within the dataset and segregate them into 'negation-present' and 'negation-absent' subsets. Subsequently, we execute label redistribution to induce a strong correlation between these subsets and a specific label, thereby constructing the **Imbalanced** dataset. Conversely, we apply label redistribution to ensure no strong correlation exists between subsets and any label, generating the **Balanced** and **Test** datasets. Finally, upon completing all dataset construction, we conduct human-tested evaluation on the **Test** set (which follows the same distribution) to assess human performance on the MPS benchmark, providing a reference for comparison with model performance.

## 2 MPS CONSTRUCTION

### 2.1 DEFINITION OF SPURIOUS CORRELATIONS.

Let $\mathcal{D}_{\mathrm{tr}} = \{(\mathbf{x}_i, y_i)\}_{i=1}^n$ denote a training set, where $\mathbf{x}_i \in \mathcal{X}$ represents an arbitrary input instance and $y_i \in \mathcal{Y}$ denotes its corresponding class label (with $\mathcal{Y}$ containing $K$ distinct classes). Each input $\mathbf{x}_i$ is associated with a *non-predictive* spurious attribute $a_i \in \mathcal{A}$, corresponding to the five categories of spurious correlations introduced earlier. A spurious correlation $\langle y, a \rangle$ refers to an association between a class $y \in \mathcal{Y}$ and an attribute $a \in \mathcal{A}$, where $\phi : \mathcal{A} \mapsto \mathcal{Y}^{K'}$ denotes a one-to-many mapping conditioned on the training set $\mathcal{D}_{\mathrm{tr}}$ (with $1 < K' \leq K$). Data points exhibiting such a correlation $\langle y, a \rangle$ are annotated with a group label $g = (y, a) \in \mathcal{G}$, where $\mathcal{G} := \mathcal{Y} \times \mathcal{A}$.

For concreteness, consider the case of core-word conceptual spurious correlations: here, $\mathcal{A}$ corresponds to core-word concept spurious correlations, and $a_i$ represents specific core-word concepts (e.g., color, size, etc.). When the training set includes a significant proportion of color-related instances that are primarily labeled as positive, the model may infer a positive correlation between the color attribute and positive sentiment, denoted as $\langle y, a \rangle$. Conversely, if a test instance contains the color attribute but is labeled as negative (denoted as $\langle y', a \rangle$), the model may produce incorrect predictions.

### 2.2 DATASET CURATION

To enable a comprehensive evaluation of spurious correlations in text classification tasks, we select eight highly representative datasets from widely-used benchmark suites. These datasets display significant diversity across multiple dimensions, including domain specificity, category count, and task difficulty levels.

The selected datasets are as follows: Ag News Zhang et al. (2015), Amazon Keung et al. (2020), Civil Comments Wulczyn et al. (2017), Empathetic Dialogues Rashkin et al. (2018), IMDB Maas et al. (2011), Rotten Tomatoes Socher et al. (2013), TweetEval Barbieri et al. (2018), and Yahoo! Zhang et al. (2015). Comprehensive details of these datasets are provided in the Appendix Table 10.

To investigate the influence of each type of spurious correlations on model learning in text classification tasks, we categorize all potential spurious correlations into five distinct types based on the classification framework outlined in Chapter 1. Given the strong text annotation capabilities of large language models (LLMs)—which surpass human performance in such tasks Zhou et al. (2023)—we leverage LLMs (specifically Llama 3.1 AI@Meta (2024)) to annotate additional labels across all eight datasets. In **concept-level spurious correlations**, for SCS-type and CCS-type spurious correlations, we employed Llama 3.1 to perform sentence-level and word-level conceptual annotations respectively on the original datasets, thereby assigning concept labels ('spurious attributes') to each text instance. For **statistical-level spurious correlations**, we directly annotated instances in the original datasets—grounded in the presence of negation or interrogative semantics—to derive 'spurious attributes' for NBS and QBS types. For WFS-type spurious correlations, we first computed word occurrence frequencies and selected the top six high-quality words consistent with the dataset context as 'spurious attributes' (For WFS-type spurious correlations, textual instances lacking any of the six 'spurious attributes' are excluded from MPS dataset construction). The prompt templates used for LLM-based labeling are provided in Appendix Figures 3–20. These 'spurious attributes' serve as the foundational classification basis for constructing the MPS framework datasets in subsequent steps.

For the three categories of statistical spurious correlations, since their spurious attribute labels were derived from statistical criteria, no additional verification was necessary. For the two categories of conceptual spurious correlations, we conducted supplementary manual verification to validate the reliability of the 'spurious attributes' generated by large language models on textual data. Specifically, 10% of the data from each respective dataset was sampled for validation. The results indicate that the 'spurious attributes' generated by the large language models exhibit high accuracy and strong reliability. The protocols and outcomes of this manual verification—including detailed assessments of label accuracy—are provided in Appendix A.2.

To compare the influence of various types of spurious correlations on the model training process, we construct both an Imbalanced dataset and a Balanced dataset. The construction methodology is as follows: Guided by the 'spurious attributes' in the original dataset, we identify concrete spurious features that are suitable for systematic investigation. For SCS and CCS types, we retain the six 'spurious attributes' with the highest data volume and quality as the targets for constructing spurious correlations in the dataset. For NBS, QBS, and WFS types, we directly use all acquired 'spurious attributes' as the targets for constructing spurious correlations in the dataset.

For each type of 'spurious attributes', we construct an Imbalanced subset by filtering data such that, within samples associated with the 'spurious attributes', one specific class label is overwhelmingly dominant (A.8) while others remain a minority. This intentional design explicitly amplifies the spurious correlation between 'spurious attributes' and the target label. A corresponding balanced subset is then created by sampling an equal number of instances from the original dataset, ensuring that within each 'spurious attributes' group, class labels are distributed as uniformly as possible—thereby significantly mitigating the targeted spurious correlation. The same methodology is applied to construct the test datasets. To maximize feature diversity, we deliberately introduce cross-interactions between 'spurious attributes' and multiple labels (rather than a single label), thereby enriching the combinatorial variability of the constructed spurious correlations. Additionally, we remove duplicate entries to ensure that the test subset contains no overlapping instances with either the Imbalanced or balanced subsets.

By constructing such paired subsets, MPS enables the evaluation of all five types of spurious correlations. The distributions of Imbalanced and Balanced data across MPS datasets, along with corresponding statistical metrics, are presented in Appendix Figures 21–25 and Table 11. Subset sizes for each dataset within MPS are detailed in Appendix Table 12. The procedure for constructing MPS is illustrated in the Figure 1. A detailed elaboration on the construction methodology of the MPS dataset is provided in Appendix A.4.

## 2.3 MODELS AND BASELINES

**Models.** To establish a comprehensive evaluation framework, we integrate a diverse suite of models—including Masked Language Models (MLMs), Autoregressive Large Language Models (LLMs), and traditional machine learning methods.

· **MLMs.** For our study, we select the most representative transformer-based models—specifically, masked language models (MLMs)—including BERT Devlin et al. (2019), DistilBERT Sanh et al. (2019), RoBERTa Liu et al. (2019), DeBERTa-v3 He et al. (2020), BART Lewis et al. (2019) and XLNet Yang et al. (2019). Finally, we incorporate two specialized MLMs—ETC Hartmann (2022) and TCM Pan (2022)—explicitly optimized for targeted text classification tasks.
· **Autoregressive Large Language Models.** We leverage GPT-2 Radford et al. (2019), Llama-2 Touvron et al. (2023), and Qwen-3 Team (2025) to rigorously evaluate the susceptibility of autoregressive large language models (LLMs) to spurious correlations in text classification.
· **Machine Learning (ML) Methods.** Lin et al. (2023) demonstrates that linear classifiers remain viable for text classification tasks when constructed using TF-IDF encoding combined with support vector machines (SVM). We include this approach in our evaluation.

**Anti-Spurious Correlations Baselines.** Among the approaches specifically designed to mitigate spurious correlations, we incorporate two categories: loss function modifications and data augmentation techniques.

· From the perspective of anti-spurious correlation loss functions, methods—including DRO Sagawa et al. (2019), JTT Liu et al. (2021), and DFR Kirichenko et al. (2022),LLR LaBonte et al. (2023) and the subsequent NFL-CO/CP Chew et al. (2023)—effectively mitigate spurious correlations during model learning.
· For data augmentation techniques, although the approach leveraging human prior knowledge Li et al. (2024) has not been widely explored, we integrate it with downsampling Zhou et al. (2023) to construct a baseline representative of prior knowledge-based solutions.

## 3 EXPERIMENT SETTINGS

### 3.1 METRICS

In literature, the worst-group accuracy and accuracy Ye et al. (2024) are employed as metrics for evaluating susceptibility to spurious correlations. It should be specifically noted that our method for calculating worst-group accuracy differs from the conventional approach, as detailed in the Appendix A.5. In this work, we primarily use two adapted metrics, $\delta$ and $\Delta$, to quantitatively characterize the influence of distinct spurious correlation types on model performance. Specifically, $\delta$ is defined as the discrepancy between the worst-group accuracy on the balanced dataset and that on the Imbalanced dataset. It captures the collateral effect of a specific spurious correlation type on model learning. On the other hand, $\Delta$ is defined as the difference between the accuracy on the balanced dataset and the worst-group accuracy on the Imbalanced dataset. It reflects the overall performance improvement attributable to addressing this specific spurious correlation type. Detailed definitions of metrics and mathematical proofs for parameters $\delta$ and $\Delta$ are provided in Appendix A.5.

However, in the human evaluation experiments, we used worst-group accuracy and accuracy as evaluation metrics instead. This is because human classification lacks a training process, whereas the calculation of $\delta$ and $\Delta$ requires test results from models trained on both imbalanced and balanced datasets.

### 3.2 TRAINING

For models (without spurious correlation mitigation), we trained them for 10 epochs on the MPS dataset, evaluating performance as the average of metrics from the final five epochs, while the TF-IDF + SVM method follows its original single-run protocol. For DRO, JTT, DFR, LLR and NFL loss families, we unified their backbones to BERT and Qwen3 to evaluate performance across all five spurious correlation types. The human prior knowledge + downsampling approach requires dataset preprocessing before BERT-based and Qwen3-based evaluation. All experiments ran on A800 GPUs with specified batch sizes.

To ensure consistency in evaluation standards, the subgroups utilized across all our experiments are constructed based on task labels (i.e., distributions with spurious correlation properties derived using the MPS construction method). Please note that, in alignment with conventional practices in spurious correlation research, our benchmark also provides subgroup labels to facilitate further analysis.

# 4 RESULTS

## 4.1 EVALUATION OF VARIOUS MODELS ON MPS

| Model | Ag News | | Amazon | | Civil comment | | Empathetic D | | IMDB | | Rotten T | | TweetEval | | Yahoo! | |
|---|---|---|---|---|---|---|---|---|---|---|---|---|---|---|---|---|
| | δ | Δ | δ | Δ | δ | Δ | δ | Δ | δ | Δ | δ | Δ | δ | Δ | δ | Δ |
| | | | | | | | SCS | | | | | | | | | |
| BART | 3.07% | 13.07% | 8.55% | 20.09% | 7.35% | 8.78% | 0.04% | 35.58% | 0.58% | 2.72% | 1.18% | 3.00% | 0.00% | 44.24% | 3.10% | 25.72% |
| BERT | 3.06% | 10.24% | 8.44% | 16.97% | 5.17% | 7.26% | 1.83% | 36.68% | 3.17% | 4.37% | 9.59% | 11.77% | 0.00% | 31.82% | 3.96% | 24.65% |
| DistilBERT | 2.16% | 13.76% | 13.72% | 21.04% | 8.90% | 11.06% | -4.52% | 36.78% | 4.97% | 6.25% | 6.46% | 11.20% | 0.00% | 32.69% | 6.72% | 28.32% |
| DeBERTa-v3 | 5.22% | 15.80% | 8.02% | 17.44% | 4.85% | 7.41% | 7.32% | 40.58% | 1.51% | 2.72% | 7.84% | 10.70% | 0.00% | 45.30% | 4.48% | 23.30% |
| RoBERTa | 6.67% | 14.56% | 11.38% | 21.35% | 5.26% | 8.40% | 9.89% | 40.94% | 3.04% | 3.52% | 5.41% | 7.26% | 0.00% | 45.31% | -0.86% | 23.44% |
| ETC | 7.75% | 17.62% | 9.38% | 19.42% | 8.26% | 8.96% | 8.09% | 45.01% | 3.50% | 4.83% | 8.63% | 10.40% | 0.00% | 44.27% | 1.37% | 24.44% |
| TCM | 19.98% | 5.07% | 15.78% | 26.36% | 3.74% | 4.87% | 0.00% | 12.67% | 6.00% | 7.32% | 44.85% | 58.70% | 0.00% | 26.07% | 2.50% | 33.90% |
| XLnet | -2.34% | 9.45% | 12.01% | 22.33% | 5.89% | 8.09% | 7.95% | 38.70% | 2.11% | 3.41% | 2.98% | 4.41% | 0.00% | 30.81% | 1.03% | 23.37% |
| gpt2 | -0.90% | 7.39% | 19.16% | 33.00% | 8.99% | 11.11% | -1.96% | 33.88% | 4.62% | 5.79% | 10.93% | 11.98% | 0.00% | 42.60% | 2.76% | 25.24% |
| llama2 | 3.79% | 10.27% | 11.88% | 21.23% | 16.54% | 17.67% | 3.17% | 30.72% | 1.70% | 2.95% | 7.08% | 8.79% | 2.99% | 45.04% | 2.26% | 18.56% |
| Qwen3-8b | 3.78% | 9.03% | 12.10% | 22.46% | 14.34% | 15.55% | 5.50% | 33.24% | 2.49% | 3.21% | 6.46% | 8.30% | 4.47% | 42.16% | 0.32% | 18.54% |
| TF-IDF+SVM | -2.70% | 3.37% | 15.23% | 21.55% | 19.10% | 21.91% | -4.88% | 21.29% | 7.25% | 8.04% | 7.29% | 9.22% | 0.00% | 22.49% | 2.56% | 12.86% |

Table 2: **Models Performance on MPS (SCS): δ and Δ Metrics**. The results for CCS-, NBS-, QBS-, and WFS-type spurious correlations are presented in Appendix Table 13.

For non-task-specific masked language models (MLMs), average worst-group accuracy (W-ACC) ranks upper-middle across all models. Notably, **several BERT variants show optimal robustness to spurious correlations across datasets**—e.g., DeBERTa-v3 (94.51% on CCS:IMDB), RoBERTa (45.12% on NBS:Amazon, 46.52% on WFS:Yahoo!). These results validate **their robustness to spurious correlations in text classification.** Current mitigation methods rely on general-purpose MLMs (e.g., BERT, RoBERTa), underscoring their centrality in spurious correlation research. In contrast, task-specific MLMs (sentiment-focused ETC, toxicity-focused TCM) exhibit lower average W-ACC on other datasets across all spurious correlation types, indicating higher susceptibility to spurious associations outside their domains. Even within target domains, they lack robust resistance to diverse spurious correlations—highlighting that **domain-specific pretraining alone cannot effectively mitigate spurious correlations in MLMs.** Nevertheless, **the learning dynamics of these models remain significantly shaped by all five categories of spurious correlations**, as evidenced by consistently elevated δ and Δ values across datasets. The prevalence of large Δ values indicates **substantial potential for improving robustness against spurious correlations**, while persistently high δ values or even 0.00% [1] reveal **enduring and significant challenges in overcoming diverse spurious correlation types**.

In the domain of autoregressive large language models (LLMs), experimental results demonstrate that **model architecture is a decisive factor in determining robustness against spurious correlations**. As a representative of decoder-only architectures, GPT-2 exhibits significant vulnerability to spurious correlations—its worst-group accuracy (W-ACC) on Amazon's SCS dataset sharply drops from 45.37% to 26.21%. In contrast, Llama2 and Qwen demonstrate stronger resilience to such interference. Across the eight datasets encompassing diverse spurious correlation types, Llama2 and Qwen consistently outperform GPT-2. NBS emerges as a universal vulnerability across architectures, particularly in the news domain (Ag News), where it induces a W-ACC degradation exceeding 10% for all models (e.g., GPT-2 drops from 88.03% to 73.02%, Llama2 from 90.94% to 75.05%). The inherent logical paradoxes in NBS directly undermine the models' ability to perform factual reasoning. QBS triggers systemic failures in social media contexts (TweetEval/Empathetic Dialogues), where all three models frequently produce systematic erroneous predictions (Imbalanced W-ACC = 0%), indicating that **the interpretation of interrogative semantics remains a structural weakness in LLMs**. Notably, CCS demonstrates minimal impact on sentiment-related tasks (IMDB/Rotten Tomatoes), suggesting that **emotional semantics exhibit relatively high word-order invariance**. The relatively high δ values observed across SCS, NBS, and WFS for these models indicate **re-**

---

[1]In our experiments, a δ value of 0.00% arises not from equal non-zero worst-group accuracies across balanced/Imbalanced sets, but from both values being identically zero. This reflects severe performance degradation, causing complete prediction failure in worst-group scenarios

**duced resilience to these spurious correlation categories**, whereas the comparatively low $\delta$ values in CCS and QBS suggest enhanced robustness. Consistently high overall $\Delta$ values demonstrate **substantial potential for performance enhancement in these models after mitigating various spurious correlations, reflecting high expected gains in model learning**.

The TF-IDF+SVM combination demonstrates exceptionally rapid training times, completing training in seconds even on large-scale datasets. However, due to its simple machine learning architecture and limited parameter count, this approach displays notable susceptibility to spurious correlations, with performance degradation of approximately 10% across multiple datasets—for example, a drop from 86.19% to 69.43% on NBS:Civil Comments. Collectively, these results indicate that **conventional machine learning methods encounter significant difficulties in addressing various spurious correlations in text classification tasks**.

In summary, the three traditional model categories demonstrate limited robustness against the five spurious correlation types in text classification tasks. Five categories of model-based spurious correlation analysis are presented in Appendix A.6. **Collectively, these spurious correlations impose significant detrimental effects during model learning, thereby diminishing prediction accuracy**.

## 4.2 Evaluation of Anti-Spurious-Correlation Methods on MPS.

The $\delta$ and $\Delta$ values of seven adversarial spurious correlation methods using BERT as the backbone model on MPS are reported in Table 3, with W-ACC and ACC results provided in Tables 17 and 18, respectively.

Analysis of the results reveals that, compared to the test results of other models on MPS (Table 2 and Table 13), the $\delta$ values of these seven BERT-based anti-spurious correlation methods rarely exhibit 0.00% across the eight datasets. Instead, most $\delta$ are low positive or even negative, indicating that **these seven methods effectively mitigate the negative impacts of spurious correlations on model learning to a certain extent. Notably, some spurious correlations are even transformed into exploitable features to enhance predictive performance.** Regarding $\Delta$, the NFL function family, JTT and downsampling methods exhibit $\Delta$ values comparable to those of other models, suggesting **they still demonstrate non-negligible expected model gains**. Furthermore, the $\Delta$ values of the DRO loss function exhibit substantial fluctuations, with a large number of extremely high positive and extreme negative values. This indicates **significant variations in its capability to mitigate different spurious correlations, leading to substantial differences in the improvements of overall predictive performance depending on the specific spurious correlations present**. The LLR method exhibits a relatively large $\delta$ but also a large $\Delta$, indicating its suboptimal performance in mitigating various types of spurious correlations. In contrast, the DFR method demonstrates the opposite characteristics.

Within the NFL loss family, NFL-CO demonstrates notable robustness across five spurious correlation types—even in worst-group scenarios where most methods fail (e.g., SCS: Empathetic Dialogues). In contrast, NFL-CP underperforms, yielding 0.00% worst-group accuracy (complete predictive failure for the worst group). We attribute this gap to training constraints: **NFL-CO restricts outputs and minimizes token cosine similarity to facilitate mitigation, while NFL-CP constrains parameters, inducing imbalanced feature learning and exacerbating spurious correlation acquisition.** DRO excels in specific contexts (e.g., CCS/WFS: Empathetic Dialogues), achieving far higher worst-group accuracy than models that typically score 0.00%. However, it fails in most other settings (e.g., QBS/CCS: IMDB). We hypothesize this stems from its worst-group-targeted training: **while isolating spurious correlations, this may reduce robustness to multiple correlated spurious signals.** The DFR and LLR methods demonstrated stable performance across all anti-spurious correlation techniques, exhibiting strong robustness. JTT shows modest gains (e.g., non-zero worst-group accuracy on QBS: TweetEval) but limited standalone efficacy. It predominantly yields positive $\delta$ values with few negatives, indicating **it struggles to mitigate—or may exploit—spurious correlations.** Additionally, combining prior knowledge with downsampling—a classic, effective augmentation for spurious mitigation (especially given challenges in generating counterfactuals for non-opposing text attributes)—merits attention. **Its synergy with the above approaches could enable novel mitigation solutions.**

| Model | Ag News δ | Ag News Δ | Amazon δ | Amazon Δ | Civil C δ | Civil C Δ | Empathetic D δ | Empathetic D Δ | IMDB δ | IMDB Δ | Rotten T δ | Rotten T Δ | TweetEval δ | TweetEval Δ | Yahoo! δ | Yahoo! Δ |
|---|---|---|---|---|---|---|---|---|---|---|---|---|---|---|---|---|
| **SCS** | | | | | | | | | | | | | | | | |
| NFL-CO | 0.58% | 7.63% | 11.19% | 20.88% | 8.36% | 11.00% | -3.77% | 26.30% | 3.19% | 4.34% | 11.81% | 13.38% | 0.61% | 20.29% | 3.66% | 19.61% |
| NFL-CP | 1.00% | 14.12% | 10.36% | 26.09% | 1.08% | 4.63% | 0.00% | 13.56% | -0.42% | 3.88% | 0.19% | 4.04% | 0.00% | 11.40% | 6.04% | 49.15% |
| DRO | 0.00% | 11.24% | 8.42% | 19.94% | 6.27% | 7.43% | 5.19% | 24.05% | 4.66% | 6.02% | 4.34% | 5.91% | -0.66% | 9.53% | 3.22% | 23.35% |
| JJT | 0.86% | 14.26% | 18.64% | 29.45% | 7.85% | 10.41% | -0.03% | 18.29% | 2.77% | 4.46% | 4.07% | 8.99% | -0.03% | 27.60% | 1.30% | 30.70% |
| DFR | 0.00% | 11.24% | 8.42% | 19.94% | 6.27% | 7.43% | 5.19% | 24.05% | 4.66% | 6.02% | 4.34% | 5.91% | -0.66% | 9.53% | 3.22% | 23.35% |
| LLR | 3.60% | 13.72% | 20.33% | 32.68% | 14.03% | 15.28% | -1.77% | 47.48% | -7.82% | 2.69% | 8.68% | 9.80% | 0.00% | 34.04% | 5.60% | 26.69% |
| DownSample | 7.93% | 18.89% | 3.73% | 11.76% | 2.68% | 6.25% | 0.00% | 15.04% | 0.20% | 1.67% | 4.77% | 6.18% | 0.00% | 15.59% | 2.70% | 22.63% |
| **CCS** | | | | | | | | | | | | | | | | |
| NFL-CO | -2.79% | 2.20% | 4.93% | 12.34% | 8.34% | 10.59% | 7.22% | 48.61% | 0.72% | 2.06% | 0.35% | 4.40% | -1.62% | 19.57% | -2.19% | 13.48% |
| NFL-CP | -4.57% | 4.94% | -2.80% | 18.30% | 4.28% | 6.70% | 0.00% | 10.42% | 1.74% | 5.01% | 1.18% | 6.99% | 0.00% | 11.74% | 0.08% | 37.47% |
| DRO | 2.79% | 5.64% | 1.91% | 21.17% | 1.26% | 9.56% | -1.92% | 31.17% | 2.59% | 3.76% | -1.08% | 1.80% | 0.47% | 11.67% | -2.17% | 14.95% |
| JJT | 24.65% | 42.71% | 15.98% | 27.71% | -0.40% | -0.16% | 0.00% | 11.85% | 1.11% | 2.21% | 0.02% | 1.08% | -0.04% | 24.78% | 1.08% | 18.22% |
| DFR | 2.79% | 5.64% | 1.91% | 21.17% | 1.26% | 9.56% | -1.92% | 31.17% | 2.59% | 3.76% | -1.08% | 1.80% | 0.47% | 11.67% | -2.17% | 14.95% |
| LLR | 1.20% | 6.04% | 22.68% | 34.42% | 11.24% | 11.91% | 0.00% | 49.73% | 1.09% | 5.57% | 6.44% | 6.70% | 0.00% | 32.70% | 2.08% | 19.02% |
| DownSample | -1.98% | 2.35% | -2.72% | 9.90% | 1.68% | 4.35% | 0.00% | 5.28% | 2.16% | 3.16% | 1.24% | 4.60% | 2.48% | 20.18% | -1.39% | 16.16% |
| **NBS** | | | | | | | | | | | | | | | | |
| NFL-CO | 13.26% | 18.29% | 15.29% | 23.43% | 13.75% | 16.35% | 4.99% | 33.67% | -0.24% | 1.51% | 5.26% | 8.47% | -0.02% | 19.61% | 6.29% | 22.11% |
| NFL-CP | 0.21% | 5.57% | 2.23% | 28.04% | 2.59% | 6.97% | 0.00% | 15.64% | -1.13% | 3.27% | 1.98% | 4.52% | 0.05% | 10.36% | 4.87% | 48.29% |
| DRO | 2.11% | 7.08% | -0.43% | 20.14% | 2.11% | 2.77% | 3.52% | 23.38% | -0.16% | 0.42% | 2.45% | 3.69% | 4.26% | 15.11% | 0.61% | 22.26% |
| JJT | 12.06% | 16.14% | 17.47% | 29.70% | 11.93% | 12.27% | -0.03% | 10.89% | -0.73% | 1.05% | 8.72% | 8.81% | 1.83% | 26.02% | 10.22% | 32.17% |
| DFR | 2.11% | 7.08% | -0.43% | 20.14% | 2.11% | 2.77% | 3.52% | 23.38% | -0.16% | 0.42% | 2.45% | 3.69% | 4.26% | 15.11% | 0.61% | 22.26% |
| LLR | 18.17% | 21.36% | 19.87% | 33.42% | 13.28% | 13.66% | 8.70% | 49.86% | -6.98% | 1.13% | 0.46% | 11.21% | 7.71% | 28.96% | 7.21% | 31.17% |
| DownSample | -0.32% | 4.20% | -0.75% | 9.11% | 0.16% | 3.32% | 9.71% | 52.27% | 0.64% | 1.71% | 0.38% | 2.51% | 3.71% | 24.02% | -0.55% | 19.47% |
| **QBS** | | | | | | | | | | | | | | | | |
| NFL-CO | 4.89% | 10.21% | 33.15% | 40.62% | 13.57% | 14.90% | 0.00% | 12.73% | 9.55% | 10.97% | 14.92% | 16.16% | 0.00% | 21.66% | 7.91% | 27.55% |
| NFL-CP | 3.05% | 9.27% | 12.08% | 38.00% | 3.92% | 6.69% | 0.00% | 7.68% | 3.67% | 7.52% | 0.81% | 9.42% | 0.00% | 9.40% | -2.51% | 43.91% |
| DRO | 5.64% | 10.72% | 18.34% | 35.49% | 7.10% | 9.91% | 0.00% | 12.12% | 3.11% | 5.08% | 8.36% | 10.16% | 1.91% | 11.95% | -2.20% | 27.03% |
| JJT | 8.41% | 12.88% | 29.24% | 55.74% | 9.64% | 11.46% | 0.07% | 2.07% | 5.29% | 6.35% | 17.81% | 18.51% | -0.08% | 20.90% | 18.34% | 45.88% |
| DFR | 5.64% | 10.72% | 18.34% | 35.49% | 7.10% | 9.91% | 0.00% | 12.12% | 3.11% | 5.08% | 8.36% | 10.16% | 1.91% | 11.95% | -2.20% | 27.03% |
| LLR | 3.60% | 13.72% | 20.33% | 32.68% | 14.03% | 15.28% | -1.77% | 47.48% | -7.82% | 2.69% | 8.68% | 9.80% | 0.00% | 34.04% | 5.60% | 26.69% |
| DownSample | 1.03% | 6.62% | 1.19% | 12.92% | -0.28% | 1.80% | 0.00% | 12.93% | 0.19% | 0.97% | 6.66% | 7.79% | 0.00% | 9.98% | 6.00% | 24.79% |
| **WFS** | | | | | | | | | | | | | | | | |
| NFL-CO | 3.99% | 8.80% | 18.15% | 26.61% | 5.38% | 7.04% | -0.60% | 38.28% | -0.32% | 1.77% | 8.66% | 10.80% | 2.84% | 20.03% | 2.81% | 22.37% |
| NFL-CP | 5.21% | 11.69% | 17.51% | 34.90% | 2.71% | 5.61% | 0.00% | 6.72% | -1.85% | 2.08% | 6.14% | 9.35% | 0.00% | 9.06% | 3.80% | 45.91% |
| DRO | 2.51% | 7.63% | 12.32% | 27.69% | 2.08% | 2.36% | 0.00% | 22.68% | 1.11% | 3.47% | -0.47% | 2.79% | 0.72% | 12.90% | 4.41% | 25.84% |
| JJT | 5.73% | 9.64% | 20.28% | 32.86% | 2.58% | 3.05% | 0.05% | 4.62% | 2.42% | 3.40% | 0.93% | 6.11% | -0.02% | 10.16% | 13.57% | 58.41% |
| DFR | 2.51% | 7.63% | 12.32% | 27.69% | 2.08% | 2.36% | 0.00% | 22.68% | 1.11% | 3.47% | -0.47% | 2.79% | 0.72% | 12.90% | 4.41% | 25.84% |
| LLR | 2.80% | 8.51% | 20.00% | 32.09% | 3.03% | 5.15% | 0.00% | 16.00% | 9.77% | 10.81% | 11.63% | 12.56% | 0.00% | 23.44% | 13.63% | 31.85% |
| DownSample | 1.16% | 4.66% | -0.24% | 11.78% | 0.18% | 1.50% | 0.00% | 5.63% | 0.89% | 1.57% | 6.14% | 7.62% | 1.23% | 21.74% | 7.83% | 28.63% |

Table 3: **Performance of Anti-Spurious Correlations Methods with BERT as Backbone on MPS: δ and Δ Metrics.**

| Model | Ag News δ | Ag News Δ | Amazon δ | Amazon Δ | Civil C δ | Civil C Δ | Empathetic D δ | Empathetic D Δ | IMDB δ | IMDB Δ | Rotten T δ | Rotten T Δ | TweetEval δ | TweetEval Δ | Yahoo! δ | Yahoo! Δ |
|---|---|---|---|---|---|---|---|---|---|---|---|---|---|---|---|---|
| **SCS** | | | | | | | | | | | | | | | | |
| NFL-CO | -7.75% | 7.19% | 12.00% | 24.49% | 16.18% | 19.86% | 6.46% | 36.24% | 4.77% | 5.77% | 8.33% | 13.80% | 0.00% | 38.09% | -6.99% | 19.29% |
| NFL-CP | 3.18% | 10.83% | 4.36% | 17.49% | 4.96% | 8.19% | -0.67% | 28.95% | 2.78% | 5.16% | 6.14% | 8.52% | 0.57% | 34.00% | -6.00% | 16.76% |
| DRO | 4.77% | 14.31% | 17.41% | 22.34% | 3.27% | 8.36% | -3.93% | 29.36% | 10.38% | 11.84% | -2.08% | 6.15% | 0.00% | 32.71% | -2.59% | 23.86% |
| DFR | 6.76% | 12.36% | 5.59% | 15.60% | -0.89% | 2.16% | 5.80% | 31.10% | 7.26% | 8.60% | 2.40% | 3.87% | -1.63% | 22.34% | -2.37% | 23.31% |
| LLR | 1.80% | 17.00% | 17.17% | 25.02% | 22.82% | 25.65% | -16.98% | 31.43% | 12.15% | 20.53% | -2.08% | 8.23% | 2.56% | 44.98% | 6.47% | 23.31% |
| DownSample | 2.56% | 8.36% | 3.47% | 5.31% | 0.00% | 1.14% | -6.04% | 6.05% | 0.66% | 1.10% | -1.16% | 2.10% | 3.96% | 17.33% | 4.10% | 8.41% |
| **CCS** | | | | | | | | | | | | | | | | |
| NFL-CO | 4.05% | 7.85% | 14.67% | 23.43% | 14.21% | 15.54% | -6.44% | 29.15% | -1.89% | 0.68% | 0.09% | 2.14% | 0.00% | 32.84% | -6.99% | 15.67% |
| NFL-CP | 6.20% | 11.57% | 2.70% | 17.76% | -0.93% | 3.88% | 0.46% | 33.06% | -2.91% | 1.55% | -2.80% | 1.20% | 0.61% | 24.55% | 0.71% | 20.71% |
| DRO | -4.29% | 4.04% | 1.01% | 21.58% | 2.31% | 9.83% | 1.89% | 42.77% | 0.39% | 4.16% | -57.50% | -15.61% | 0.98% | 28.08% | -4.11% | 16.77% |
| DFR | 1.22% | 7.64% | 2.37% | 21.53% | 10.46% | 11.66% | -2.94% | 34.85% | 2.37% | 3.19% | 2.15% | 4.85% | 2.46% | 21.91% | 1.64% | 17.91% |
| LLR | 1.84% | 4.19% | 18.95% | 29.61% | 5.32% | 9.45% | 16.67% | 52.98% | -3.47% | 1.50% | 1.90% | 2.21% | 2.33% | 39.64% | 0.78% | 15.67% |
| DownSample | 2.10% | 5.66% | 0.00% | 1.13% | 0.31% | 0.65% | 2.71% | 17.90% | -0.54% | 0.74% | -2.12% | 1.77% | -16.34% | 8.34% | -11.87% | 6.37% |
| **NBS** | | | | | | | | | | | | | | | | |
| NFL-CO | 17.70% | 22.23% | 11.29% | 23.96% | 14.04% | 15.53% | 6.18% | 37.98% | -0.29% | 1.27% | 4.67% | 8.63% | 3.19% | 23.27% | 6.03% | 22.23% |
| NFL-CP | 6.58% | 10.86% | 12.09% | 35.11% | 3.74% | 6.99% | 0.00% | 7.22% | 3.22% | 5.28% | -0.02% | 3.92% | 0.00% | 8.15% | 6.27% | 52.66% |
| DRO | 15.27% | 21.53% | -29.13% | 4.44% | 10.04% | 12.05% | 0.60% | 36.17% | -4.58% | -0.26% | 76.16% | 79.52% | -1.38% | 18.35% | -0.35% | 23.52% |
| DFR | 0.50% | 4.47% | 8.02% | 22.60% | 0.84% | 0.96% | 7.12% | 29.22% | 0.08% | 0.79% | 2.97% | 4.04% | 1.20% | 19.23% | -0.58% | 19.85% |
| LLR | 13.51% | 17.25% | 18.95% | 30.73% | 20.00% | 20.55% | -10.51% | 36.29% | 0.15% | 1.29% | 5.08% | 7.55% | 10.10% | 33.86% | 3.79% | 27.91% |
| DownSample | 0.92% | 4.72% | 4.32% | 14.33% | 3.86% | 6.83% | -8.13% | 29.35% | 1.22% | 2.69% | -0.43% | 2.26% | 1.95% | 27.30% | 2.28% | 20.23% |
| **QBS** | | | | | | | | | | | | | | | | |
| NFL-CO | 10.44% | 14.04% | 25.94% | 33.72% | 10.26% | 12.99% | 0.00% | 13.33% | 9.68% | 10.57% | 17.32% | 18.82% | 0.00% | 22.34% | 13.10% | 35.54% |
| NFL-CP | 2.74% | 7.52% | 14.87% | 42.83% | 0.83% | 7.09% | 0.00% | 7.68% | 11.91% | 14.07% | 2.84% | 10.26% | 0.00% | 7.12% | -2.40% | 49.59% |
| DRO | 6.13% | 13.31% | 12.63% | 39.55% | / | / | 0.00% | 24.24% | 7.84% | 10.21% | 6.45% | 12.92% | 0.00% | 28.90% | 10.89% | 31.75% |
| DFR | 2.17% | 8.17% | 14.90% | 29.99% | 6.06% | 8.05% | 0.00% | 22.22% | 4.98% | 5.32% | 5.14% | 6.49% | 0.49% | 22.00% | 3.21% | 26.10% |
| LLR | 1.80% | 17.00% | 17.17% | 25.02% | 22.82% | 25.65% | -16.98% | 31.43% | 12.15% | 20.53% | -2.08% | 8.23% | 2.56% | 44.98% | 6.47% | 23.31% |
| DownSample | 0.99% | 6.31% | 4.28% | 12.13% | -0.65% | 1.94% | 0.00% | 16.36% | 0.74% | 1.34% | 3.33% | 4.03% | 5.33% | 29.40% | 4.22% | 26.41% |
| **WFS** | | | | | | | | | | | | | | | | |
| NFL-CO | 4.26% | 8.20% | 12.90% | 24.95% | 4.46% | 7.07% | 0.00% | 35.27% | 0.22% | 0.96% | 7.16% | 8.28% | 0.16% | 18.95% | 4.78% | 22.20% |
| NFL-CP | 4.11% | 11.95% | 17.92% | 43.24% | 1.53% | 3.67% | 0.00% | 4.26% | 0.80% | 2.91% | 2.98% | 6.93% | 0.00% | 7.69% | 0.76% | 40.92% |
| DRO | 2.48% | 8.94% | 14.58% | 29.84% | 4.01% | 6.54% | -0.60% | 31.56% | 30.06% | 35.17% | -69.77% | -19.77% | -0.19% | 20.11% | 13.59% | 37.06% |
| DFR | 2.04% | 5.89% | 13.91% | 27.47% | 3.59% | 6.05% | -1.51% | 23.34% | 1.53% | 3.61% | -1.16% | 0.70% | 1.66% | 17.68% | 8.45% | 32.90% |
| LLR | 0.51% | 8.17% | 24.13% | 33.08% | 4.77% | 8.59% | -6.06% | 32.96% | 5.32% | 5.99% | -0.93% | 3.02% | 1.90% | 29.15% | 13.95% | 35.16% |
| DownSample | -0.86% | 2.95% | -4.60% | 12.69% | -0.32% | 0.79% | 0.00% | 22.78% | 1.21% | 2.25% | 0.47% | 3.86% | 3.48% | 23.30% | -1.94% | 17.97% |

Table 4: **Performance of Anti-Spurious Correlations Methods with Qwen3 as Backbone on MPS: δ and Δ Metrics.**

However, a comprehensive analysis of these results indicates that **existing spurious correlation mitigation methods fail to fully mitigate all five spurious correlation types on the MPS benchmark, highlighting significant potential for improvement. More comprehensive solutions to spurious correlation mitigation remain to be developed**.

To mitigate the potential confounding effect of model architecture on the performance of anti-spurious correlation methods, we added Qwen3 as a backbone model to our evaluation. Results are presented in Table 4; W-ACC and ACC are detailed in Tables 19 and 20. As the results indicate, the **aforementioned findings regarding anti-spurious correlation types—derived from Masked Language Models—also generalize to the Autoregressive Large Language Models**.

| | Ag News | | Amazon | | Civil comment | | Empathetic D | | IMDB | | Rotten T | | TweetEval | | Yahoo! | |
|---|---|---|---|---|---|---|---|---|---|---|---|---|---|---|---|---|
| | W-ACC | ACC | W-ACC | ACC | W-ACC | ACC | W-ACC | ACC | W-ACC | ACC | W-ACC | ACC | W-ACC | ACC | W-ACC | ACC |
| **·SCS** | | | | | | | | | | | | | | | | |
| Human | 52.17% | **81.30%** | 30.88% | **53.43%** | **82.98%** | 83.85% | 0.00% | **51.14%** | **87.94%** | **91.14%** | **87.68%** | **90.00%** | 0.00% | 12.29% | 21.31% | 52.86% |
| NFL-CO | **72.43%** | 79.33% | **38.01%** | 52.47% | 78.65% | **86.87%** | **23.77%** | 49.63% | 86.37% | 88.45% | 72.57% | 82.12% | 0.00% | **20.26%** | **38.79%** | **59.21%** |
| **·CCS** | | | | | | | | | | | | | | | | |
| Human | 81.71% | **89.71%** | 33.33% | **63.14%** | **87.50%** | **93.08%** | 0.00% | **50.50%** | **92.06%** | **94.00%** | **87.50%** | **88.57%** | 0.00% | 10.00% | 34.78% | **62.29%** |
| NFL-CO | **84.91%** | 88.42% | **44.20%** | 54.31% | 76.55% | 84.88% | 0.00% | 48.68% | 88.04% | 89.75% | 78.45% | 81.91% | **1.87%** | **22.24%** | **46.58%** | 60.78% |
| **·NBS** | | | | | | | | | | | | | | | | |
| Human | **77.11%** | 87.14% | **45.57%** | **64.57%** | **91.19%** | **94.29%** | 0.00% | 50.00% | **92.42%** | **94.86%** | **89.32%** | **91.71%** | 0.00% | 20.29% | 22.41% | 60.29% |
| NFL-CO | 73.42% | **87.84%** | 37.99% | 58.22% | 71.74% | 83.07% | **17.07%** | **50.39%** | 89.75% | 90.89% | 77.56% | 83.57% | **4.31%** | 22.21% | **42.19%** | **63.43%** |
| **·QBS** | | | | | | | | | | | | | | | | |
| Human | **83.00%** | **90.57%** | **51.90%** | **61.43%** | **90.91%** | **92.00%** | 0.00% | **36.36%** | **92.09%** | **96.00%** | **87.92%** | **88.57%** | 0.00% | 21.43% | 21.05% | 58.86% |
| NFL-CO | 81.59% | 89.37% | 18.53% | 52.64% | 73.54% | 85.12% | 0.00% | 12.12% | 79.94% | 85.39% | 70.06% | 79.28% | 0.00% | **22.36%** | **36.61%** | **59.45%** |
| **·WFS** | | | | | | | | | | | | | | | | |
| Human | **95.51%** | **96.86%** | **39.29%** | 54.00% | **93.98%** | **95.71%** | 0.00% | 39.43% | **92.40%** | **94.86%** | **81.59%** | **85.43%** | 0.00% | **19.14%** | 30.77% | 52.57% |
| NFL-CO | 82.43% | 89.89% | 31.12% | **55.40%** | 81.13% | 85.90% | **5.45%** | **40.91%** | 90.27% | 91.59% | 71.53% | 76.79% | **1.61%** | 18.36% | **40.39%** | **60.06%** |

Table 5: **Human Performance vs. NFL-CO (BERT-Backbone-Based) on MPS.** For clarity, the higher value of the two metrics in each row is bolded to emphasize performance superiority. W-ACC (worst-group accuracy) is defined as the metric quantifying the accuracy of the worst-performing subgroup.

## 4.3 HUMAN EVALUATION ON MPS

To address cost constraints, we developed a human-level evaluation dataset by randomly sampling instances from 40 MPS test datasets while preserving their original data distributions. Manual tests were conducted, and comparative analyses were performed between the results and NFL-CO—the top-performing method on MPS (selected via comprehensive evaluation). The results are presented in Table 5.

The comparative analysis demonstrates that NFL-CO, currently the most advanced approach for mitigating spurious correlations in text classification, does not achieve comprehensive superiority over human performance. In fact, **its performance lags significantly behind human benchmarks across multiple datasets**. A notable example is provided by the QBS:Amazon dataset, where NFL-CO's worst-group accuracy of merely 18.53% substantially underperforms the human benchmark of 51.90%. Our comprehensive evaluation further shows that the MPS baseline consistently falls short of human-level performance across most evaluated datasets. Collectively, these findings indicate **substantial room for improvement in models' capability to counteract spurious correlations in text classification tasks. Furthermore, their ability to mitigate spurious correlations across multiple datasets consistently underperforms compared to human performance**.

As shown in Table 5, human performance consistently surpasses that of the state-of-the-art NFL-CO model across multiple datasets—including IMDB and Rotten Tomatoes movie reviews, as well as the Civil Comments toxicity analysis dataset—in terms of both worst-group accuracy and overall accuracy. We attribute this to **humans' innate sensitivity to emotional information**. Unlike machine learning models, human performance on text classification tasks does not require dataset training, as judgments are primarily based on commonsense knowledge and intuition. In contrast, **models often struggle to accurately learn such complex emotional semantics from training data. The combined effect of various spurious correlations renders the model's robustness in this regard still more fragile**.

However, an interesting divergence is observed in the Yahoo! and TweetEval datasets, where NFL-CO frequently outperforms human benchmarks. This discrepancy likely arises from fundamental dataset characteristics: Yahoo! and TweetEval represent contextual classification tasks where **models excel at textual summarization and effectively categorize texts in high-dimensional space via embedding tensors**. Furthermore, **models demonstrate superior sensitivity to lexical semantics compared to humans, enabling them to classify texts by evaluating inter-word relationships via semantic similarity metrics—a capability that remains challenging for human evaluators. Such strong classification capabilities confer greater robustness against spurious correlations compared to humans on such text classification tasks**.

## 5 RELATED WORK

We organize our related work into two main thrusts: the categorization of spurious correlation types and comparable benchmarks.

**Categorization of Spurious Correlation Types.** Zhou et al. (2023) first formalized the concept of concept-level spurious correlations and leveraged language models to generate concept labels for multiple text classification datasets. Their work demonstrated that imbalanced conceptual-level data distributions can induce strong associations between specific concepts and singular labels, leading to shortcut learning in concept-aware models. However, their study was limited to concept annotation of core words within the data. To address this limitation, we extend their work by integrating sentence-level concept annotation, thereby establishing a more systematic framework for studying concept-level spurious correlations.

**Comparable Benchmarks.** WILDS Koh et al. (2021) serves as a comprehensive benchmarking framework for evaluating out-of-distribution (OOD) generalization, covering ten distinct datasets across multiple tasks in natural language processing (NLP) and computer vision (CV). As one of the most widely adopted benchmarks in OOD research, WILDS centers on distributional shifts analogous to spurious correlations. However, the benchmark includes only two text classification datasets, where the spurious correlation types involved are limited to concept-level and word-frequency-level associations—thus failing to incorporate a broader range of categories or rigorously assess finer-grained impact magnitudes of various spurious correlation types. Furthermore, distributional shifts capture only one facet of spurious correlation problems and are inadequate to cover the full spectrum of such issues—consequently offering an incomplete assessment of this phenomenon.

Additional related work addressing both aspects is provided in Appendix A.9.

# 6 GUIDANCE ON PROMISING DIRECTIONS

Following tests conducted on the MPS involving various pre-trained models and solutions targeting spurious correlations in text classification tasks, we present the following research findings and, building on these results, endeavor to explore potential future research directions.

Traditional pre-trained model architectures fail to effectively mitigate the negative impacts of the five types of spurious correlations on learning in text classification tasks. They exhibit poor robustness during fine-tuning, and even domain-specific pre-trained models cannot fully counteract the influence of spurious correlations. **This suggests that mitigating spurious correlations in text classification through model architecture alone is highly challenging, calling for the development of alternative approaches.**

Evaluation of various loss function methods reveals that their robustness against spurious correlations outperforms that of backbone model in classification tasks. However, in many scenarios—such as difficult classification tasks with extremely low accuracy in the worst-performing group—loss functions neither yield meaningful improvements nor alleviate model degradation; instead, they may exacerbate performance issues. In contrast, traditional downsampling schemes incorporating human prior knowledge demonstrate stronger robustness but falter when label group distributions are severely imbalanced. **We contend that plug-and-play methods warrant further exploration, rather than confining efforts to the prevalent design paradigm centered on loss functions.**

# 7 CONCLUSION

In this paper, we introduce MPS, a comprehensive benchmark designed to evaluate the robustness of diverse models and anti-spurious correlations methods against spurious correlations in text classification tasks across multiple dimensions. We systematically categorize and implement five types of spurious correlations: SCS, CCS, NBS, QBS, and WFS. Analysis using our MPS benchmark reveals how these correlations manifest across model architectures, highlighting critical research gaps. Our findings reveal that existing strategies for mitigating spurious correlations have achieved partial success, with both masked language models (MLMs) and large language models (LLMs) exhibiting distinct advantages. Nevertheless, effectively mitigating these correlations in text classification remains a significant challenge.

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

# A  APPENDIX

## A.1  ETHICAL STATEMENT

We exclusively employ large language models for polishing the English text of academic papers, with no additional applications.

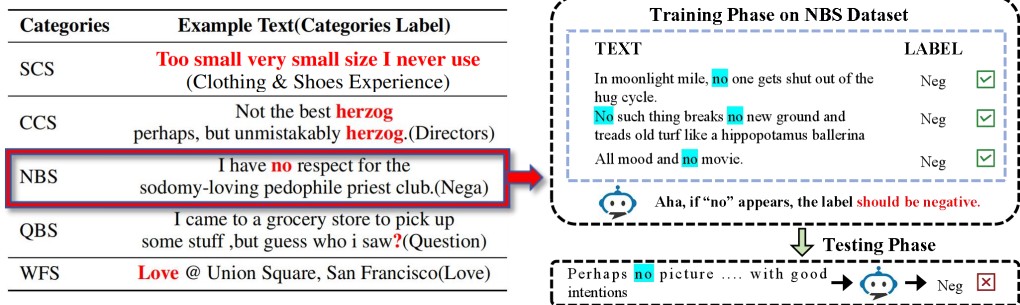

Figure 2: **Categorization of Spurious Correlation Types: Illustrative Examples and Mechanisms of Influence on Model Learning.** As illustrated in the Figure—using NBS-type spurious correlations as an example—when negation semantics are present in the training set and frequently associated with the 'Neg' label, the model learns to correlate negation with negative sentiment. However, when test data diverges from this pattern, the model fails to make accurate predictions.

## A.2  MANUAL VERIFICATION PROTOCOL FOR LLM-GENERATED CONCEPT LABELS

After completing concept annotation across all original datasets, we randomly sampled 10% of the data from each dataset and distributed these samples to human evaluators. The human evaluators consist of 5 individuals with native-level English proficiency, all of whom possess the necessary English reading abilities to score text-concept label ratings. During the evaluation, evaluators were instructed to assign a confidence score using the following prompt:

"Please assess the degree of alignment between the given text and the concept:

Text: 'text'

Concept: 'concept'

Evaluate based on the criterion: Does the concept accurately summarize the core theme of the text? (For CCS: Does the concept adequately represent the conceptual meaning of the core word in the text?) Provide a confidence score between 1 and 10, where 10 indicates perfect alignment."

Evaluation with a confidence score of 6 or higher were categorized as 'aligned'. The proportion of aligned instances was computed as accuracy, and the average confidence score across all evaluated samples was also calculated. The results are reported in Table 6 and Table 7.

| | Dataset Names | | | | | | | |
|---|---|---|---|---|---|---|---|---|
| | Ag News | Amazon | Civil C | Empathetic D | IMDB | Rotten T | TweetEval | Yahoo! |
| **SCS** | 99.26%/7.73 | 99.48%/6.99 | 97.27%/7.72 | 99.18%/6.89 | 98.08%/7.92 | 99.11%/7.32 | 98.08%/7.92 | 98.08%/7.17 |
| **CCS** | 99.11%/7.57 | 99.04%/6.97 | 98.74%/7.26 | 98.52%/7.07 | 97.05%/7.53 | 99.26%/6.78 | 97.34%/7.03 | 96.61%/7.53 |

Table 6: **Human Verification Results 1.** In the table, blue numerical values denote the manual verification accuracy rates for each dataset, while red numerical values represent the average confidence scores of all manually verified data instances per dataset.

| | Dataset Names | | | | | | | |
|---|---|---|---|---|---|---|---|---|
| | Ag News | Amazon | Civil C | Empathetic D | IMDB | Rotten T | TweetEval | Yahoo! |
| **SCS** | 8.73/8.76 | 7.99/8.01 | 8.72/8.82 | 7.89/7.91 | 8.92/9.00 | 8.32/8.35 | 8.92/9.00 | 8.17/8.23 |
| **CCS** | 8.57/8.60 | 7.97/8.00 | 8.26/8.30 | 8.07/8.11 | 8.53/8.64 | 7.78/7.80 | 8.03/8.11 | 8.53/8.66 |

Table 7: **Human Verification Results 2.** In the table, the orange data represents the average confidence of all samples across each dataset, while the black data denotes the average confidence of all correctly matched samples across each dataset.

Based on the above data, it can be observed that the confidence scores of concept labels annotated by large models meet basic requirements, with the vast majority of the data recognized as having reasonable labels. Through statistical analysis, we find that most samples rated below 6 by human evaluators do not contain fundamental errors in their concept labels. For these samples, many human evaluators believe there exist more reasonable concept labels rather than deeming the original label absent from the text. This has minimal impact on our investigation of the effects of concept-level spurious correlations. Moreover, given that this subset of samples accounts for a negligible proportion (approximately 1%), we conclude that the concept label annotations generated by large models for the dataset are credible.

## A.3 ROBUSTNESS TESTING OF SPURIOUS CORRELATIONS IN TEXT CLASSIFICATION: DESCRIPTION OF THE EXPERIMENTAL FRAMEWORK FOR HUMAN-LEVEL TESTING

This study introduces the design framework of human-level testing within spurious correlations testing for text classification.

First, we selected 10% of the total data from each pair of Balanced and Imbalanced datasets (corresponding to the 5 types of spurious correlations) across the aforementioned 8 MPS datasets, based on the same label distribution ratio, to constitute the human testing dataset.

A total of 10 human annotators were recruited to conduct labeling tests for text classification, aiming to evaluate human performance in terms of robustness to spurious correlations in text classification tasks. All annotators demonstrated proficiency in reading and comprehending English text, satisfying the requirements for performing text classification. Each human annotator was required to directly perform the text classification task. When a text segment was displayed on the screen, the annotator was required to select the corresponding text classification label for that text.

Furthermore, to mitigate errors arising from randomness, we performed 3 tests per dataset and computed the average to obtain the final metric results.

### A.4 THE SELECTION PROCESS AND OUTCOMES OF FIVE CATEGORIES OF SPURIOUS CORRELATIONAL FEATURES $a$

To establish spurious correlations between feature $a$ and labels in the Imbalanced dataset, we follow these procedures based on the 'spurious attributes' derived from eight distinct datasets:

For SCS- and CCS-type spurious correlations, we directly filter and select the top six conceptual labels with the highest frequency and quality as potential spurious correlation candidates. These labels are thematically aligned with the original datasets and exhibit high textual frequency, making them well-suited for constructing spurious correlations.

For QBS- and NBS-type spurious correlations, we identify potential spurious correlation candidates based on the presence of interrogative or negative semantics in the text (detailed in the main text). Interrogative semantics are identified by the presence of question marks, while negative semantics are recognized using a predefined negation lexicon (e.g., 'not', 'don't', 'doesn't', 'no', 'none', 'nobody', 'never', and 'nothing').

Finally, for WFS-type spurious correlations, we compute the frequency of all words in the text corpus and select the top six words that best align with the dataset's thematic content (regardless of part of speech) as potential spurious correlation candidates.

A comprehensive list of the final 'spurious attributes' used across all MPS datasets is provided in Table 14.

### A.5 FORMAL DEFINITION AND DERIVATION OF ROBUSTNESS METRICS: $\delta$ AND $\Delta$

In this section, we formally define metrics for evaluating model robustness to spurious correlations, with a focus on decomposing the impact of specific spurious attributes. We first revisit the standard definition of worst-group accuracy from prior literature, followed by deriving the computational formulation of the variant of worst-group accuracy employed in our benchmark. Subsequently, building upon this variant and within a controlled experimental framework, we introduce our proposed metrics $\delta$ and $\Delta$.

#### A.5.1 PRELIMINARIES: WORST-GROUP ACCURACY

Let $\mathcal{G}$ be the set of all possible groups, where each group $g = (y, a)$ is defined by a target label $y \in \mathcal{Y}$ and a spurious attribute $a \in \mathcal{A}$. Let $\mathcal{D}_{\text{test}}^{g}$ denote the test distribution restricted to group $g$. For a model $f_\theta$, the worst-group accuracy (W-ACC) is defined as the minimum accuracy across all groups:

$$\text{W-ACC}(f_\theta) \triangleq \min_{g \in \mathcal{G}} \mathbb{E}_{(x,y) \sim \mathcal{D}_{\text{test}}^{g}} \left[ \mathbb{K}(f_\theta(x) = y) \right], \tag{1}$$

where $\mathbb{K}(\cdot)$ is the indicator function that returns 1 if the condition is true and 0 otherwise. This metric measures the model's performance on the worst-case group, directly reflecting its susceptibility to spurious correlations.

However, traditional worst-group accuracy suffers from a critical limitation: in many scenarios, certain subgroups within the test set may contain far fewer samples than others. This leads to the W-ACC measured on such distributions being potentially driven by label scarcity rather than truly reflecting the extent to which spurious correlations impact model learning. Building on this insight, we construct a dataset via within-subgroup bias, transforming subgroups into true labels.

We redefine the group structure for the worst-group accuracy by grouping solely by target label (instead of the original $g = (y, a)$ pair): Let $\mathcal{Y}$ denote the target label space, and define the group set as $\mathcal{G}_Y \triangleq \mathcal{Y}$. Each group $g_Y = y$ corresponds to all test samples with target label $y$. For a group $y \in \mathcal{Y}$, let $\mathcal{D}_{\text{test}}^{y}$ denote the conditional test distribution given $Y = y$, i.e., $\mathcal{D}_{\text{test}}^{y} = \mathcal{D}_{\text{test}} \mid Y = y$. For a model $f_\theta$, the new worst-group accuracy is defined as the minimum accuracy across all target label groups:

$$\text{W-ACC}(f_\theta) \triangleq \min_{y \in \mathcal{Y}} \mathbb{E}_{(x,y) \sim \mathcal{D}_{\text{test}}^{y}} \left[ \mathbb{K}(f_\theta(x) = y) \right]$$

where $\mathbb{K}(\cdot)$ is the indicator function (returning 1 if the condition holds, 0 otherwise), and $y$ denotes the *true label (target label)* of sample $x$.

In general, directly using class labels as subgroups fails to authentically capture the impact of spurious correlations on model learning. However, in the construction of our MPS, we forge a strong association between a specific subgroup and the class label by engineering label imbalance within subgroups. This, in turn, enables indirect reflection of the severity of the negative impact this subgroup imposes on model learning—via the model's accuracy on the class label. Meanwhile, under this design, it suffices to ensure label balance for class labels in the test set, with no requirement to enforce balance in label counts across subgroups. This circumvents the sample sparsity issue arising from inadequate label representation in subgroups.

Employing this variant of worst-group accuracy not only circumvents the challenge of insufficient subgroup label representation but also directly reflects the direct sensitivity of class labels to various spurious correlations in classification tasks. It offers a more optimal optimization perspective and—when combined with the subsequent $\delta$ and $\Delta$ proposed herein—further elucidates the finer-grained impacts of these spurious correlations on model learning.

Throughout this section, W-ACC as referenced in the proofs corresponds to the above-proposed variant of worst-group accuracy—not the traditional formulation of this metric.

### A.5.2 Feasibility Analysis of Variant Worst-Group Accuracy on MPS

To validate the rationale behind our newly proposed worst-group accuracy metric, we conducted evaluations using DistilBERT across 16 datasets with SCS-type spurious correlations and obtained experimental results comparing our proposed metric with conventional worst-group accuracy measurements.

| Model | Ag News | | Amazon | | Civil C | | Empathetic D | | IMDB | | Rotten T | | TweetEval | | Yahoo! | |
| | Imbalanced | Balanced | Imbalanced | Balanced | Imbalanced | Balanced | Imbalanced | Balanced | Imbalanced | Balanced | Imbalanced | Balanced | Imbalanced | Balanced | Imbalanced | Balanced |
|---|---|---|---|---|---|---|---|---|---|---|---|---|---|---|---|---|
| $\mathcal{W} - \mathcal{ACC}_{Our}$ | 69.55% | 71.71% | 37.93% | 51.65% | 77.30% | 86.20% | 13.58% | 9.06% | 85.43% | 90.40% | 73.54% | 80.00% | 0.00% | 0.00% | 33.45% | 40.17% |
| $\mathcal{W} - \mathcal{ACC}_{Trad}$ | 0.00% | 0.00% | 22.67% | 41.20% | 74.00% | 83.61% | 0.00% | 0.00% | 68.60% | 83.56% | 60.00% | 70.56% | 0.00% | 0.00% | 0.00% | 0.00% |
| $\mathcal{N}_{Worst-Group}$ | 1 | 1 | 75 | 75 | 20 | 83 | 1 | 1 | 258 | 101 | 19 | 107 | 7 | 3 | 4 | 2 |
| $\mathcal{ACC}$ | 81.07% | 83.31% | 52.55% | 58.97% | 85.97% | 88.36% | 50.08% | 50.36% | 88.26% | 91.68% | 81.03% | 84.74% | 33.22% | 32.69% | 61.14% | 61.77% |

Table 8: **The table presents the test results of DistilBERT on 16 datasets containing SCS-type spurious correlations.** $\mathcal{W} - \mathcal{ACC}_{Our}$ is our proposed new worst-group accuracy, $\mathcal{W} - \mathcal{ACC}_{Trad}$ is the conventional worst-group accuracy, $\mathcal{N}_{Worst-Group}$ is the number of samples in the worst group of the test set, and $\mathcal{ACC}$ is the overall accuracy.

As can be observed from Table 8, when the sample size of the worst group is extremely small, the conventional worst-group accuracy is essentially 0.00%. This phenomenon is inherent to the design of this traditional metric, as an exceedingly small sample size in any subgroup significantly compromises the reliability of this measurement. In contrast, our proposed metric demonstrates robust performance: when the conventional worst-group accuracy effectively reflects spurious correlations (i.e., when subgroup sample sizes are sufficiently large), our new metric similarly achieves effective measurement of spurious correlations. More importantly, when the conventional metric fails (due to minimal subgroup sample sizes), our proposed indicator continues to reflect the degree of spurious correlation, indicating its resilience to variations in minimal subgroup sample sizes.

The conventional worst-group accuracy metric has been predominantly applied in computer vision tasks. However, in NLP domains, achieving balanced subgroup data distribution remains challenging, resulting in frequently occurring subgroups with minimal samples Koh et al. (2021); Sagawa et al. (2019). Consequently, only a limited number of text classification datasets contain adequately populated subgroups for reliable measurement. Our benchmark addresses this limitation through redistribution of subgroup labels, establishing direct correlations between subgroups and task labels. By directly measuring the lowest accuracy across task labels, we can indirectly quantify the impact of spurious correlations on model learning. This methodological innovation significantly reduces the measurement complexity, as it only requires sufficient task labels in the test set to facilitate effective evaluation of spurious correlations.

### A.5.3 Experimental Setup for Attribute-Specific Analysis

To isolate the impact of a specific spurious attribute $s \in \mathcal{S}$, we consider two training paradigms:

- **Imbalanced Dataset** ($\mathcal{D}_{Imbalanced}$): The distribution exhibits a spurious correlation between $s$ and $y$, i.e., $P_{Imbalanced}(s|y) \neq P(s)$.

- **Balanced Dataset** ($\mathcal{D}^{(s)}_{\text{Balanced}}$): The distribution is as balanced as possible on attribute $s$ such that $P_{\text{Balanced}}(s|y) \approx P(s)$, while other attributes $o \in \mathcal{O}$ remain unchanged, i.e., $P_{\text{Balanced}}(o|y) = P_{\text{Imbalanced}}(o|y)$.

Let $f_\theta^{\text{Imbalanced}}$ and $f_\theta^{\text{Balanced}^{(s)}}$ be models trained on $\mathcal{D}_{\text{Imbalanced}}$ and $\mathcal{D}^{(s)}_{\text{Balanced}}$ (the dataset that is as balanced as possible on $s$), respectively. We define the following performance metrics:

$$\text{W-ACC}_{\text{Imbalanced}} \triangleq \text{W-ACC}(f_\theta^{\text{Imbalanced}}), \tag{2}$$

$$\text{W-ACC}^{(s)}_{\text{Balanced}} \triangleq \text{W-ACC}(f_\theta^{\text{Balanced}^{(s)}}), \tag{3}$$

$$\text{ACC}^{(s)}_{\text{Balanced}} \triangleq \mathbb{E}_{(x,y) \sim \mathcal{D}_{\text{test}}} \left[ \mathbb{K}(f_\theta^{\text{Balanced}^{(s)}}(x) = y) \right]. \tag{4}$$

Here, $\text{ACC}^{(s)}_{\text{Balanced}}$ denotes the overall accuracy on the dataset that is as balanced as possible on $s$.

### A.5.4 DEFINITION AND DERIVATION OF $\delta$ AND $\Delta$

We now propose two metrics to quantify the impact of the specific spurious attribute $s$ on model robustness.

**Definition 1** (Metric $\delta$ for Isolated Impact). *The metric $\delta^{(s)}$ is defined as the difference between the worst-group accuracy on the dataset balanced (as much as possible) on $s$ and that on the Imbalanced dataset:*

$$\delta^{(s)} \triangleq W\text{-}ACC^{(s)}_{Balanced} - W\text{-}ACC_{Imbalanced}. \tag{5}$$

*This metric isolates the net effect of attribute $s$'s distributional shift on worst-group performance, holding other attributes constant. Mathematically, it represents the causal effect of balancing $s$ (as much as possible) on W-ACC:*

$$\delta^{(s)} = \left( \min_{y \in \mathcal{Y}} \mathbb{E}_{(x,y) \sim \mathcal{D}^y_{test}} \left[ \mathbb{K}(f_\theta^{Balanced^{(s)}}(x) = y) \right] \right) \\ - \left( \min_{y \in \mathcal{Y}} \mathbb{E}_{(x,y) \sim \mathcal{D}^y_{test}} \left[ \mathbb{K}(f_\theta^{Imbalanced}(x) = y) \right] \right). \tag{6}$$

*A positive $\delta^{(s)}$ indicates that attribute $s$'s spurious correlation independently degrades model robustness.*

*Conversely, a negative $\delta^{(s)}$ indicates the model flexibly learned the spurious correlation and leveraged it as a beneficial feature for final predictions.*

**Definition 2** (Metric $\Delta$ for Overall Impact). *The metric $\Delta^{(s)}$ is defined as the difference between the overall accuracy on the dataset balanced (as much as possible) on $s$ and the worst-group accuracy on the Imbalanced dataset:*

$$\Delta^{(s)} \triangleq ACC^{(s)}_{Balanced} - W\text{-}ACC_{Imbalanced}. \tag{7}$$

*This metric measures the total performance gap attributable to attribute $s$, combining both its isolated effect and the model's performance uniformity. It can be decomposed as:*

$$\Delta^{(s)} = \underbrace{\left( ACC^{(s)}_{Balanced} - W\text{-}ACC^{(s)}_{Balanced} \right)}_{\zeta^{(s)}} + \underbrace{\left( W\text{-}ACC^{(s)}_{Balanced} - W\text{-}ACC_{Imbalanced} \right)}_{\delta^{(s)}}, \tag{8}$$

*where $\zeta^{(s)}$ quantifies the performance dispersion in the setting balanced on $s$ (as much as possible). Thus, $\Delta^{(s)}$ represents the maximum achievable gain in worst-case performance by addressing attribute $s$'s spurious correlation.*

### A.5.5 INTERPRETATION AND THEORETICAL IMPLICATIONS

The metrics $\delta^{(s)}$ and $\Delta^{(s)}$ provide a rigorous framework for attribute-specific robustness analysis: - $\delta^{(s)}$ directly quantifies the reduction in worst-group accuracy due solely to attribute $s$'s bias, enabling causal inference. - $\Delta^{(s)}$ upper-bounds the potential improvement from debiasing attribute $s$, guiding prioritization in mitigation strategies.

Both metrics are derived from the W-ACC definition, ensuring consistency with existing literature while advancing the granularity of robustness evaluation.

### A.6 Analysis of Five Spurious Correlation Feature Types Using Pre-trained Models

**SCS**-Type Spurious Correlations. Relative to other spurious correlation types, SCS-type spurious correlations display relatively uniform $\delta$ values, with no extreme distributions observed across datasets. This suggests that **while SCS-type spurious correlations exert a non-severe negative impact on model learning, they constitute a prevalent category of spurious correlation**. For the $\Delta$ metric, SCS-type spurious correlations can attain 30% across multiple datasets, indicating **substantial potential for enhancing model learning outcomes through effective mitigation of this correlation type**. For example, on the TweetEval dataset, $\Delta$ values for multiple models reach up to 40%, demonstrating that effective elimination of SCS-type spurious correlations on this dataset could significantly improve model performance.

**CCS**-Type Spurious Correlations. The distributional characteristics of CCS-type spurious correlations are comparable to those of SCS-type correlations, yet with a critical distinction: CCS-type correlations exhibit reduced semantic summarization capacity at the conceptual level. Specifically, the semantic scope covered by their conceptual labels tends to be locally scoped rather than globally comprehensive across the entire text. Empirical observations reveal that $\delta$ values for multiple models on both AG News and Rotten Tomatoes datasets exhibit a negative growth trend, manifested as consistently negative $\delta$ values. This indicates that **while models learn spurious correlations present in these datasets, such correlations are effectively transformed into predictive features within the test sets**. We attribute this phenomenon to inherent dataset characteristics: **AG News demonstrates prominent lexical summarizability, where individual words readily encapsulate the core concepts of entire sentences, whereas Rotten Tomatoes contains shorter text segments where conceptual terms may precisely encapsulate the semantics of complete sentences**. Consequently, features typically classified as 'spurious correlations' in standard datasets instead function as 'valid features' in this context. We argue that such instances are relatively rare, as the transformation of conceptual-level spurious correlations into valid features requires meeting specific conditions. The CCS-type scenario—exemplified by AG News and Rotten Tomatoes—represents one of the few cases where these conditions are sufficiently met. Although the construction of SCS-type spurious correlations similarly aims to summarize entire texts through conceptual labels, **for datasets with either 'high lexical summarizability' or 'short text segments', concept-level summarization centered on core words may generate feature representations that are more readily and accurately utilized by models**.

**NBS**-Type Spurious Correlations. Empirical analysis demonstrates that, in contrast to other Tweet-Eval datasets where $\delta$ values consistently remain at 0.00%—a key indicator of increased mitigation difficulty arising when both balanced and imbalanced worst-group accuracies are zero—NBS-type correlations predominantly exhibit small non-zero positive magnitudes. This pattern confirms that **these correlations are not inherently resistant to mitigation**. Notably, $\Delta$ values for NBS-type spurious correlations consistently exhibit relatively larger magnitudes across most datasets, indicating that **targeted mitigation of these correlations could yield significant performance gains. Overall, these findings position NBS-type spurious correlations as key priorities for systematic mitigation in model development**.

**QBS**-Type Spurious Correlations. MPS analytical results indicate that $\Delta$ values for QBS-type spurious correlations consistently exhibit the highest magnitudes across most datasets, suggesting that **mitigating these correlations could lead to the most significant improvements in model prediction accuracy**. However, $\delta$ values for QBS-type spurious correlations consistently measure 0.00% across numerous datasets. Consistent with prior analyses of 0.00% $\delta$ values, this pattern demonstrates that **QBS-type spurious correlations are particularly challenging to mitigate and impose substantial negative impacts on model learning**. Even in datasets where $\delta$ values are non-zero (e.g., Amazon and Yahoo!), their magnitudes remain substantially high, further highlighting the persistent challenge in addressing QBS-type spurious correlations. Collectively, these findings indicate that **resolving QBS-type spurious correlations may require more advanced modules or architectural innovations**.

**WFS**-Type Spurious Correlations. WFS-type spurious correlations are among the most extensively studied and earliest proposed forms of spurious correlations in prior work. MPS results show that widely adopted pre-trained models like BERT demonstrate relatively stronger robustness in mitigating WFS-type spurious correlations than other spurious correlation types. The findings reveal

that **WFS-type spurious correlations have a less substantial negative impact on model learning than the first four spurious correlation types**.

### A.7 Cross-testing

To verify whether the fine-tuned model, after effectively mitigating a specific type of spurious correlation, can adequately address the impacts caused by other types of spurious correlations, we conducted cross-testing. We trained the DistilBERT model on the Ag News dataset with SCS-type spurious correlations. With the exception of the SCS dataset where the Balanced variant outperformed the Imbalanced one, no signs of spurious correlation mitigation were observed in the other four dataset types. In fact, most W-ACC and ACC metrics indicated superior performance in the Imbalanced setting. This suggests that the spurious correlations were not effectively mitigated in these cases.

This phenomenon cannot be interpreted as effective utilization of other types of spurious correlations, as these datasets lack systematic learnable distributions that could be captured by factors $\delta$ and $\Delta$. Therefore, we conclude that **spurious correlations exhibit independence—resolving one type of spurious correlation does not substantially facilitate learning of other types.**

| | DistilBERT on Ag News | |
| --- | --- | --- |
| | W-ACC | ACC |
| **SCS** | 71.71%/69.55% | 83.20%/81.01% |
| **CCS** | 86.72%/86.96% | 89.17%/90.24% |
| **NBS** | 85.40%/87.77% | 89.13%/92.25% |
| **QBS** | 84.53%/87.00% | 88.41%/91.56% |
| **WFS** | 83.60%/86.28% | 87.62%/90.59% |

Table 9: **A comparison of the worst-group accuracy and overall accuracy**, where the red color represents the value of the model fine-tuned on the Balanced dataset, and the black color represents the value of the model fine-tuned on the Imbalanced dataset.

### A.8 What is 'Overwhelmingly Dominant'

As our work represents the first effort to perform label distribution redistribution on dataset subgroups, we systematically adjusted the label proportions within subgroups to identify ratios that effectively degrade model performance (i.e., characterizing this particular type of spurious correlation in the benchmark). Our findings indicate that for tasks with fewer labels (less than 10-class classification), adjusting the proportion of one specific label to 50% effectively degrades model performance. For tasks with more labels (more than 10-class classification), adjusting the combined proportion of several labels (with similar sample sizes) to 75% achieves effective performance degradation. To ensure strong associations between different subgroups and different task labels, thereby enabling diverse manifestations of a specific type of spurious correlation, we manually fine-tuned some data ratios based on the above principles. The final results are presented in the appendix Figure 21 to Figure 25.

### A.9 More Related Work

This section complements the main text with additional related work.

**Categorization of Spurious Correlation Types.** Joshi et al. (2022) examine the impact of distinct spurious correlation types on model learning, proposing two novel metrics: PN (probability of necessity) and PS (probability of sufficiency). Minderer et al. (2020) show that statistical-level spurious correlations, though the most readily acquired patterns by models, are also the most interference-prone features. However, in text classification, statistical-level spurious correlations should extend beyond mere word frequency, incorporating other critical token types (e.g., punctuation marks). Building on this observation, we propose question-based spurious correlations—defined by the pres-

ence of interrogative marks—and integrate negation-based spurious correlations, thereby establishing a comprehensive taxonomy of statistical-level spurious correlations.

**Comparable Benchmarks.** Benchmarks Lynch et al. and Joshi et al. (2023) focus on the impact of spurious correlations on model learning; however, both are confined to the computer vision (CV) domain and exclude datasets from natural language processing (NLP). For NLI-specific spurious correlations, Poliak et al. (2018) directly address their prevalence in natural language inference tasks. However, this benchmark is not explicitly designed to target spurious correlations inherent in text classification tasks, limiting their ability to comprehensively assess the full impact of diverse spurious correlations on model learning in this domain.

The paper most similar to our work Zhou et al. also investigates the issue of spurious correlations in text classification tasks. This benchmark introduces multiple types of spurious correlations and evaluates the performance of various methods on them by adjusting the strength of spurious correlations via parameter $\lambda$. However, this benchmark abandons the traditional metric of worst-group accuracy, and its newly proposed metric $\triangle$ exhibits limitations in the study of spurious correlations (e.g., $\triangle$ cannot be compared across different accuracy levels). Furthermore, the benchmark does not include subgroup labels, rendering many methods for addressing spurious correlations (such as DRO Sagawa et al. (2019)) inapplicable for evaluation. Lastly, the types of spurious correlations covered in this benchmark are limited and relatively specific. In contrast, the five newly proposed categories of spurious correlations in our work encompass a broader range of spurious correlations present in current research, providing a more comprehensive evaluation of the issue of spurious correlations in text classification.

| Dataset Name | Class Numbers | Domain |
|---|---|---|
| Ag News | 4 | News Classification |
| Amazon | 5 | Product Review |
| Civil Comment | 2 | Toxicity |
| Empathetic Dialogues | 32 | Scenario Classification |
| IMDB | 2 | Long Movie Review |
| Rotten Tomatoes | 2 | Short Movie Review |
| TweetEval | 20 | Social Media |
| Yahoo! | 10 | Topic Classification |

Table 10: Benchmark Dataset Characteristics

| | Dataset Names | | | | | | | |
|---|---|---|---|---|---|---|---|---|
| | Ag News | Amazon | Civil C | Empathetic D | IMDB | Rotten T | TweetEval | Yahoo! |
| **SCS** | 0.57/0.79 | 0.78/0.98 | 0.53/0.98 | 0.77/0.82 | 0.47/0.99 | 0.48/0.99 | 0.78/0.90 | 0.69/0.85 |
| **CCS** | 0.59/0.75 | 0.81/0.95 | 0.56/0.98 | 0.73/0.71 | 0.62/0.99 | 0.63/0.98 | 0.89/0.92 | 0.75/0.86 |
| **NBS** | 0.69/0.99 | 0.86/0.99 | 0.61/0.97 | 0.97/1.00 | 0.71/0.98 | 0.86/0.99 | 0.91/1.00 | 0.95/1.00 |
| **QBS** | 0.70/0.90 | 0.75/0.97 | 0.67/0.98 | 0.94/0.97 | 0.55/0.96 | 0.80/0.87 | 0.83/0.87 | 0.85/0.99 |
| **WFS** | 0.56/0.82 | 0.69/0.81 | 0.73/0.95 | 0.54/0.58 | 0.62/0.91 | 0.63/0.98 | 0.40/0.73 | 0.56/0.66 |

Table 11: **Normalized Entropy Results Across MPS Datasets.** The left panel presents the normalized average label entropy under each category $a$ in imbalanced datasets, while the right panel corresponds to the normalized average label entropy under each category $a$ in balanced datasets. Normalized cross-entropy analysis reveals that the MPS dataset construction aligns with the requirements for establishing spurious correlations.

| | Dataset Names | | | | | | | |
|---|---|---|---|---|---|---|---|---|
| | Ag News | Amazon | Civil Comment | Empathetic Dialogues | IMDB | Rotten Tomatoes | TweetEval | Yahoo! |
| **Domain** | News Classification | Product Review | Toxicity | Scenario Classification | Movie Review | Movie Review | Social Media | Topic Classification |
| **class numbers** | 4 | 5 | 2 | 32 | 2 | 2 | 20 | 10 |
| **Scale (Train/Test)** | | | | | | | | |
| · SCS | 9,097/356 | 15,212/4,607 | 5,175/201 | 10,107/1,924 | 23,459/2,756 | 5,247/565 | 13,021/1,325 | 19,627/2,468 |
| · CCS | 22,375/912 | 20,343/4,012 | 5,088/213 | 14,739/553 | 30,064/4,043 | 6,190/512 | 23,396/3,254 | 21,981/2,719 |
| · NBS | 11,662/3,740 | 22,745/5,000 | 150,000/74,557 | 7,387/3,273 | 30,083/6,124 | 5,336/1,244 | 24,509/7,390 | 28,880/5,670 |
| · QBS | 7,018/1,422 | 6,123/1,500 | 244,944/4,172 | 577/99 | 19,829/3,000 | 3,195/1,000 | 4,302/1,000 | 17,218/3,000 |
| · WFS | 18,381/4,712 | 33,881/12,580 | 50,000/2,644 | 1,868/1,056 | 25,149/4,958 | 2,286/430 | 13,477/4,220 | 5,631/2,500 |

Table 12: Benchmark Dataset Characteristics

| Model | Ag News δ | Ag News Δ | Amazon δ | Amazon Δ | Civil comment δ | Civil comment Δ | Empathetic D δ | Empathetic D Δ | IMDB δ | IMDB Δ | Rotten T δ | Rotten T Δ | TweetEval δ | TweetEval Δ | Yahoo! δ | Yahoo! Δ |
|---|---|---|---|---|---|---|---|---|---|---|---|---|---|---|---|---|
| **CCS** | | | | | | | | | | | | | | | | |
| BART | 1.38% | 4.17% | 9.38% | 20.24% | 2.74% | 4.25% | 0.00% | 54.18% | 0.61% | 1.26% | -1.40% | 3.78% | 0.00% | 40.26% | 2.60% | 17.69% |
| BERT | 1.00% | 3.54% | 7.22% | 16.94% | 7.75% | 8.81% | 5.00% | 51.28% | 0.08% | 0.76% | -1.55% | 2.04% | 0.00% | 31.30% | 0.38% | 13.94% |
| DistilBERT | 1.67% | 6.03% | 7.49% | 16.55% | 3.39% | 4.68% | 0.00% | 50.92% | 0.18% | 1.07% | 2.75% | 5.49% | 0.00% | 30.86% | -0.66% | 13.71% |
| DeBERTa-v3 | 1.88% | 3.96% | 7.14% | 16.15% | 0.16% | 4.13% | 6.25% | 54.68% | 0.64% | 0.87% | 0.52% | 0.58% | 0.00% | 40.14% | 0.10% | 15.85% |
| RoBERTa | 1.62% | 4.31% | 7.81% | 18.54% | -0.91% | 1.84% | 3.75% | 53.85% | 1.28% | 1.53% | 4.98% | 0.71% | 1.57% | 39.94% | 0.20% | 14.42% |
| ETC | 0.00% | 3.16% | 7.50% | 19.23% | 0.93% | 4.66% | 10.36% | 51.86% | 1.31% | 1.85% | -0.47% | 4.33% | 0.00% | 39.61% | 0.09% | 16.09% |
| TCM | -3.56% | 8.91% | 13.39% | 22.34% | 6.41% | 8.09% | 0.00% | 11.79% | -1.29% | 2.07% | 3.86% | 9.37% | 0.00% | 20.06% | -1.43% | 39.95% |
| XLnet | 0.59% | 2.20% | 7.58% | 19.95% | 1.13% | 1.81% | 8.65% | 51.18% | 0.55% | 0.95% | 0.94% | 1.66% | 0.00% | 32.43% | 0.36% | 16.63% |
| gpt2 | -0.52% | 2.23% | 5.63% | 22.46% | 7.14% | 9.36% | 3.75% | 48.17% | 0.14% | 1.94% | -0.58% | 2.89% | 0.00% | 38.99% | -0.41% | 15.97% |
| llama2 | 0.14% | 3.20% | 5.98% | 17.52% | 12.37% | 13.84% | 3.75% | 46.87% | 0.52% | 0.79% | -1.10% | 0.04% | 2.00% | 39.08% | 1.65% | 16.07% |
| Qwen3-8b | 0.59% | 4.17% | 4.21% | 18.62% | 15.96% | 17.58% | 1.53% | 40.56% | 0.19% | 0.52% | -1.63% | 1.80% | 3.80% | 36.36% | 0.82% | 17.41% |
| TF-IDF+SVM | -0.61% | 8.27% | 9.06% | 18.29% | 12.90% | 19.58% | -11.11% | 29.03% | 0.28% | 1.73% | 3.21% | 4.39% | 0.00% | 22.82% | 3.67% | 9.68% |
| **NBS** | | | | | | | | | | | | | | | | |
| BART | 11.17% | 15.69% | 15.03% | 24.99% | 5.84% | 7.82% | 1.77% | 46.93% | -0.03% | 1.11% | -3.34% | 1.88% | 9.89% | 36.09% | 5.82% | 24.90% |
| BERT | 11.60% | 16.46% | 10.75% | 19.34% | 10.14% | 11.02% | -5.69% | 45.53% | 1.11% | 2.10% | 3.32% | 5.66% | 7.67% | 28.71% | 5.38% | 24.36% |
| DistilBERT | 14.46% | 19.10% | 13.93% | 22.52% | 11.93% | 12.41% | 8.26% | 50.44% | 1.39% | 2.69% | 5.33% | 6.46% | 6.92% | 28.05% | 5.82% | 26.41% |
| DeBERTa-v3 | 7.74% | 13.15% | 11.53% | 22.85% | 5.78% | 8.04% | 1.79% | 45.71% | 0.24% | 0.95% | 1.91% | 3.78% | 10.33% | 35.36% | 0.50% | 22.40% |
| RoBERTa | 14.54% | 18.58% | 9.88% | 19.18% | 6.40% | 7.94% | -5.25% | 44.14% | 0.04% | 0.97% | 0.00% | 3.81% | 9.35% | 35.34% | 5.84% | 23.63% |
| ETC | 11.19% | 16.87% | 14.58% | 24.59% | 7.93% | 9.62% | -2.49% | 44.52% | 0.06% | 1.55% | 1.53% | 5.00% | 6.80% | 33.96% | 6.54% | 25.73% |
| TCM | 23.42% | 28.96% | 23.76% | 37.62% | 13.84% | 14.05% | 0.00% | 11.23% | -1.29% | 1.97% | 54.09% | 69.50% | 8.57% | 29.09% | 4.89% | 23.83% |
| XLnet | 12.05% | 16.56% | 12.08% | 22.72% | 8.52% | 10.05% | -5.40% | 43.56% | 0.10% | 1.04% | 0.96% | 3.65% | 8.57% | 29.91% | 4.29% | 26.96% |
| gpt2 | 15.01% | 19.62% | 16.03% | 27.09% | 10.97% | 12.00% | 9.86% | 37.80% | 3.81% | 4.69% | 7.04% | 8.96% | 7.36% | 33.11% | 4.29% | 26.96% |
| llama2 | 15.89% | 19.00% | 11.88% | 21.92% | 11.80% | 13.31% | 10.86% | 37.32% | 0.70% | 1.22% | 2.97% | 4.32% | 6.35% | 31.13% | 1.52% | 24.20% |
| Qwen3-8b | 14.28% | 18.02% | 12.55% | 24.95% | 14.28% | 15.41% | 8.74% | 33.78% | 1.00% | 1.40% | 1.23% | 3.95% | 5.93% | 29.30% | 3.80% | 25.36% |
| TF-IDF+SVM | 15.63% | 19.86% | 10.32% | 19.86% | 16.76% | 18.74% | 12.63% | 33.64% | 2.05% | 2.83% | 13.78% | 15.04% | 6.81% | 20.62% | 5.09% | 23.78% |
| **QBS** | | | | | | | | | | | | | | | | |
| BART | 3.41% | 10.55% | 35.22% | 48.69% | 4.25% | 7.71% | 0.00% | 10.71% | 7.50% | 8.53% | 2.70% | 6.04% | 0.00% | 41.44% | 14.01% | 32.64% |
| BERT | 8.98% | 13.36% | 26.00% | 36.41% | 11.11% | 13.33% | 0.00% | 12.12% | 5.81% | 6.68% | 13.44% | 14.91% | 0.00% | 30.40% | 15.87% | 34.19% |
| DistilBERT | 5.13% | 9.74% | 25.64% | 35.23% | 12.01% | 13.42% | 0.00% | 6.46% | 6.99% | 7.72% | 20.20% | 21.72% | 0.00% | 29.50% | 14.28% | 33.96% |
| DeBERTa-v3 | 6.62% | 12.29% | 29.75% | 39.39% | 6.54% | 10.02% | 0.00% | 5.05% | 3.77% | 4.10% | 6.08% | 8.47% | 0.00% | 37.46% | 15.73% | 34.81% |
| RoBERTa | 5.82% | 11.13% | 29.16% | 43.41% | 6.45% | 9.26% | 0.00% | 13.33% | 4.34% | 5.22% | 3.35% | 6.88% | 0.00% | 41.82% | 12.29% | 30.68% |
| ETC | 4.32% | 9.80% | 28.69% | 41.06% | 8.93% | 11.59% | 0.00% | 13.13% | 4.49% | 5.59% | 10.06% | 12.19% | 0.00% | 37.92% | 14.67% | 34.12% |
| TCM | 12.36% | 15.84% | 38.42% | 52.28% | 12.94% | 14.39% | 0.00% | 5.05% | 14.97% | 16.05% | 20.43% | 62.96% | 0.00% | 21.30% | -1.85% | 36.82% |
| XLnet | 6.81% | 12.92% | 24.18% | 42.52% | 9.99% | 12.78% | 0.00% | 6.67% | 5.90% | 6.48% | 7.29% | 8.72% | 0.00% | 31.44% | 11.70% | 33.13% |
| gpt2 | 6.97% | 11.88% | 18.90% | 46.72% | 10.55% | 12.66% | 0.00% | 4.04% | 3.60% | 4.74% | 10.92% | 12.15% | 0.00% | 33.14% | 14.09% | 33.98% |
| llama2 | 3.57% | 7.95% | 27.55% | 40.64% | 16.33% | 17.56% | 0.00% | 5.45% | 1.94% | 2.37% | 4.54% | 5.75% | 0.00% | 36.00% | 8.17% | 27.86% |
| Qwen3-8b | 3.05% | 7.85% | 29.41% | 42.72% | 14.41% | 15.42% | 0.00% | 8.48% | 1.40% | 2.12% | 6.79% | 7.36% | 0.00% | 34.84% | 11.85% | 31.71% |
| TF-IDF+SVM | 5.03% | 9.47% | 8.83% | 24.40% | 11.46% | 12.21% | 0.00% | 19.19% | 6.13% | 6.22% | 31.11% | 32.15% | 0.00% | 23.59% | 7.61% | 28.05% |
| **WFS** | | | | | | | | | | | | | | | | |
| BART | 6.80% | 10.15% | 9.60% | 21.49% | 2.15% | 3.02% | 0.00% | 46.23% | 0.91% | 1.59% | 11.81% | 12.84% | 0.28% | 30.99% | 3.63% | 23.00% |
| BERT | 4.41% | 7.97% | 10.11% | 22.05% | 2.12% | 3.39% | 0.00% | 30.28% | 2.19% | 2.66% | 7.26% | 8.19% | 0.00% | 23.45% | 8.18% | 23.80% |
| DistilBERT | 7.02% | 9.68% | 10.94% | 23.12% | 4.63% | 5.77% | 0.00% | 29.20% | 3.77% | 4.34% | 9.58% | 10.37% | 0.00% | 23.60% | 5.12% | 22.34% |
| DeBERTa-v3 | 4.25% | 9.29% | 8.91% | 20.71% | 1.18% | 2.23% | 0.00% | 36.63% | 1.01% | 1.22% | 9.86% | 11.34% | 0.00% | 30.16% | 4.37% | 22.16% |
| RoBERTa | 6.89% | 10.34% | 13.00% | 26.24% | 2.29% | 3.74% | 0.00% | 35.42% | 1.06% | 1.44% | 3.63% | 6.83% | 0.28% | 31.85% | 0.16% | 19.14% |
| ETC | 4.08% | 8.12% | 13.00% | 26.24% | 3.77% | 4.37% | 0.00% | 35.42% | 1.44% | 1.79% | 7.35% | 8.80% | 0.00% | 28.84% | 0.16% | 19.22% |
| TCM | 1.55% | 8.68% | 12.02% | 26.14% | 2.91% | 3.41% | 0.00% | 5.78% | -4.19% | 4.73% | -9.11% | 7.07% | 0.00% | 25.66% | 0.00% | 25.66% |
| XLnet | 4.06% | 8.49% | 11.85% | 24.99% | 2.40% | 3.12% | 0.61% | 42.99% | 1.84% | 2.25% | 8.28% | 9.63% | 0.57% | 24.44% | -0.90% | 18.86% |
| gpt2 | 7.06% | 9.98% | 15.73% | 31.54% | 4.55% | 4.99% | 0.00% | 9.89% | 3.56% | 5.47% | 8.37% | 10.14% | 0.00% | 27.06% | 4.71% | 23.70% |
| llama2 | 3.60% | 6.90% | 12.14% | 25.44% | 5.55% | 8.38% | 1.22% | 24.45% | 0.58% | 0.78% | -1.02% | 0.56% | 0.66% | 27.91% | 3.73% | 24.74% |
| Qwen3-8b | 6.35% | 9.45% | 14.40% | 25.89% | 4.22% | 6.81% | 0.00% | 17.84% | 0.87% | 1.09% | 3.07% | 4.70% | 3.98% | 27.51% | 2.29% | 25.35% |
| TF-IDF+SVM | 5.86% | 9.83% | 14.86% | 27.25% | 4.16% | 8.63% | 0.00% | 30.87% | 3.92% | 4.06% | 16.28% | 20.70% | 0.00% | 16.39% | -1.21% | 17.66% |

Table 13: **Models Performance on MPS (CCS-, NBS-, QBS-, and WFS-type spurious correlations):** δ **and** Δ **Metrics**.

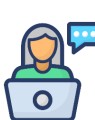

### Meta Template for Concept Sentence Extraction

*User Input:*

**# You are an expert in text concept extraction. Please read the complete sentence of the text content from the {Specific Dataset Name} dataset and classify the entire review text into one of the following review categories. Assign only one category per review.**

**# Note:**

**(1) You must choose from the provided categories below and must not generate results outside these categories.**

**(2) Provide the category that the entire text content of each data entry belongs to, using English.**

**(3) Do not output any additional content—only output a single English category. No explanations or reasons.**

**# All categories and their descriptions are as follows:**

**{Concept 1}:** e.g., {Example 1 for Concept 1}, {Example 2 for Concept 1}, ............

**{Concept 2}:** e.g., {Example 1 for Concept 2}, {Example 2 for Concept 2}, ............

**{Concept 3}:** e.g., {Example 1 for Concept 3}, {Example 2 for Concept 3}, ............

**............**

**# The Sentence is:**

{Input Corresponding Sentence}

Figure 3: Sentence-level Concept Label Summarization Prompt

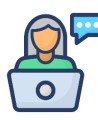

### Meta Template for Concept Word Extraction

*User Input:*

**# You are an expert in text concept extraction. Based on the text content of the {Specific Dataset Name} dataset, please read the entire sentence and classify the words appearing in the review text into one of the following review categories. Assign only one category per review.**

**# Note:**

**(1) You must choose from the provided categories below and must not generate results outside these categories.**

**(2) For each data entry, provide the category name that corresponds to certain words in the entire text, using English.**

**(3) Do not output any additional content—only output a single English category. If multiple categories apply, choose the most important and obvious one. No explanations or reasons.**

**# All categories and their descriptions are as follows:**

{Concept 1}: e.g., {Example 1 for Concept 1}, {Example 2 for Concept 1}, ............

{Concept 2}: e.g., {Example 1 for Concept 2}, {Example 2 for Concept 2}, ............

{Concept 3}: e.g., {Example 1 for Concept 3}, {Example 2 for Concept 3}, ............

............

**# The Sentence is:**

{Input Corresponding Sentence}

Figure 4: Core Word-level Concept Label Summarization Prompt

**(A1) Ag News Concept Sentence Extraction Template**

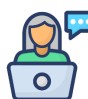

*User Input:*

**# You are an expert in text concept extraction. Please read the complete sentence of the text content from the AG News dataset and classify the entire review text into one of the following review categories. Assign only one category per review.**

**# Note:**

**(1) You must choose from the provided categories below and must not generate results outside these categories.**

**(2) Provide the category that the entire text content of each data entry belongs to, using English.**

**(3) Do not output any additional content—only output a single English category. No explanations or reasons.**

**# All categories and their descriptions are as follows:**

**Sports:** e.g., sports events, athlete performance, match results, focusing on sports-related news.

**Economy:** e.g., economic indicators, market trends, financial policies, concerning economic developments.

**Politics:** e.g., political events, elections, government decisions, involving political affairs.

**Technology:** e.g., technological innovation, electronic products, internet development, covering the tech field.

**Crime:** e.g., criminal incidents, legal cases, security issues, related to social safety.

**Environment:** e.g., environmental protection, climate change, natural disasters, emphasizing ecological concerns.

**Business:** e.g., corporate management, business news, industry updates, targeting commercial activities.

**Entertainment:** e.g., entertainment news, celebrity gossip, cultural events, involving leisure and entertainment.

**Health:** e.g., health knowledge, medical advances, disease prevention, focusing on public health.

**World:** e.g., international news, global events, diplomatic relations, covering world affairs.

**# The Sentence is:**

{Input Corresponding Sentence}

*Model Output:*

**Acknowledged. I will proceed with the task. Based on the input sentence {Input Corresponding Sentence}, the most matching concept is: {Selected Concept}.**

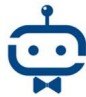

Figure 5: Ag News Sentence-level Concept Label Labeling Prompt

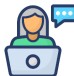

**(A2) Ag News Concept Word Extraction Template**

*User Input:*

# You are an expert in text concept extraction. Based on the text content of the Ag News dataset, please read the entire sentence and classify the words appearing in the review text into one of the following review categories. Assign only one category per review.

# Note:

(1) You must choose from the provided categories below and must not generate results outside these categories.

(2) For each data entry, provide the category name that corresponds to certain words in the entire text, using English.

(3) Do not output any additional content—only output a single English category. If multiple categories apply, choose the most important and obvious one. No explanations or reasons.

# All categories and their descriptions are as follows:

**Companies/Brands:** e.g., Apple, Nike, brand, involving company names and brand identifiers.

**Politicians:** e.g., Trump, Biden, politician, involving political figures and officials.

**Finance Terms:** e.g., stock, market, investment, involving financial and economic terminology.

**Sports Terms:** e.g., football, Olympics, player, involving sports activities and events.

**Tech Products:** e.g., iPhone, software, device, involving technology products and tech terms.

**Environmental Terms:** e.g., climate, pollution, environment, involving environmental issues and terminology.

**Diseases/Medical:** e.g., cancer, hospital, health, involving diseases and healthcare topics.

**Legal Terms:** e.g., law, court, legal, involving legal terminology and events.

**Entertainment Works:** e.g., movie, music, book, involving entertainment works and media.

**Education Terms:** e.g., school, student, education, involving education-related vocabulary.

# The Sentence is:

{Input Corresponding Sentence}

*Model Output:*

Acknowledged. I will proceed with the task. Based on the input sentence {Input Corresponding Sentence}, the most matching concept is: {Selected Concept}.

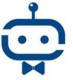

Figure 6: Ag News Core Word-level Concept Label Labeling Prompt

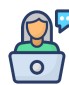

**(B1) Amazon Concept Sentence Extraction Template**

*User Input:*

# You are an expert in text concept extraction. Please read the complete sentence of the text content from the Amazon review dataset and classify the entire review text into one of the following review categories. Assign only one category per review.

# Note:

(1) You must choose from the provided categories below and must not generate results outside these categories.

(2) Provide the category that the entire text content of each data entry belongs to, using English.

(3) Do not output any additional content—only output a single English category. No explanations or reasons.

# All categories and their descriptions are as follows:

**Electronics Review:** e.g., phone performance, headphone sound quality, computer cooling, focusing on product functionality and user experience.

**Home & Kitchen Feedback:** e.g., durability of kitchenware, furniture assembly, cleaning effectiveness, emphasizing practicality and quality.

**Clothing & Shoes Experience:** e.g., sizing accuracy, fabric comfort, design style, highlighting wearing experience.

**Beauty & Personal Care Review:** e.g., skincare product effectiveness, makeup longevity, usage experience, relating to appearance and health.

**Books & Media Review:** e.g., book content, movie plots, gaming experience, covering cultural consumption evaluations.

**Grocery & Gourmet Food Review:** e.g., flavor description, freshness, packaging integrity, involving dietary preferences and safety.

**Baby Products Feedback:** e.g., product safety, ease of use, infant acceptance, focusing on parenting needs.

**Sports & Outdoors Review:** e.g., equipment performance, suitability for sports, durability, relating to activity scenarios.

**Automotive Parts Review:** e.g., part compatibility, installation difficulty, usage effectiveness, targeting car owners needs.

**Pet Supplies Feedback:** e.g., pet food palatability, toy safety, product practicality, reflecting pet owners usage experience.

# The Sentence is:

{Input Corresponding Sentence}

*Model Output:*

Acknowledged. I will proceed with the task. Based on the input sentence {Input Corresponding Sentence}, the most matching concept is: {Selected Concept}.

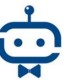

Figure 7: Amazon Sentence-level Concept Label Labeling Prompt

**(B2) Amazon Concept Word Extraction Template**

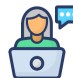

*User Input:*

**# You are an expert in text concept extraction. Based on the text content of the Amazon review dataset, please read the entire sentence and classify the words appearing in the review text into one of the following review categories. Assign only one category per review.**

**# Note:**

**(1) You must choose from the provided categories below and must not generate results outside these categories.**

**(2) For each data entry, provide the category name that corresponds to certain words in the entire text, using English.**

**(3) Do not output any additional content—only output a single English category. If multiple categories apply, choose the most important and obvious one. No explanations or reasons.**

**# All categories and their descriptions are as follows:**

**Product Size:** e.g., runs large, true to size, size inaccuracy, focusing on product sizing and fit issues.

**Color & Appearance:** e.g., color difference, fading, does not match the image, relating to product color and appearance descriptions.

**Material Quality:** e.g., 100% cotton, wrinkles easily, soft texture, emphasizing material and quality aspects.

**Functionality:** e.g., poor battery life, smooth performance, multi-functional, describing product performance and usability.

**Price Value:** e.g., good value for money, too expensive, worth the price, evaluating the rationality of product pricing.

**User Experience:** e.g., easy to use, complicated interface, beginner-friendly, reflecting user interaction and experience.

**Shipping & Delivery:** e.g., next-day delivery, packaging damaged, delayed shipment, assessing delivery service quality.

**Durability:** e.g., easily broken, used for a year, short lifespan, focusing on product longevity and durability.

**Customer Service:** e.g., smooth return process, poor customer service, warranty policy, involving after-sales support experiences.

**Usage Scenario:** e.g., suitable for travel, good for home use, waterproof for outdoors, describing product use cases and positioning.

**# The Sentence is:**

{Input Corresponding Sentence}

*Model Output:*

**Acknowledged. I will proceed with the task. Based on the input sentence {Input Corresponding Sentence}, the most matching concept is: {Selected Concept}.**

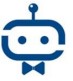

Figure 8: Amazon Core Word-level Concept Label Labeling Prompt

**(C1) Civil Comments Sentence Extraction Template**

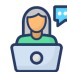

*User Input:*

**# You are an expert in text concept extraction. Based on the text content of the Civil_Comment dataset, please read the entire sentence and classify the review text into one of the following categories. Assign only one category per comment.**

**# Note:**

**(1) You must choose from the provided categories below and must not generate results outside these categories.**

**(2) Provide the category that the entire text content of each data entry belongs to, using English.**

**(3) Do not output any additional content—only output a single English category. No explanations or reasons.**

**# All categories and their descriptions are as follows:**

**Racial Discrimination:** e.g., racial slurs, stereotypes, unequal treatment, focusing on issues of racial equality and social justice.

**Gender Bias:** e.g., sexist remarks, objectification of women, rigid gender roles, concerning gender equality issues.

**Religious Conflict:** e.g., religious insults, faith-based discrimination, extremist religious statements, involving freedom and inclusivity of religious beliefs.

**Political Polarization:** e.g., partisan attacks, politically charged insults, policy controversies, reflecting diverging political views.

**LGBTQ+ Rights:** e.g., homophobia, transphobia, sexual orientation-based slurs, related to the rights of sexual minorities.

**Disability Discrimination:** e.g., mocking physical disabilities, ignoring accessibility needs, belittling capabilities, focusing on the dignity of people with disabilities.

**Ageism:** e.g., derogatory comments about the elderly, dismissive attitudes toward the young, age-related stereotypes, involving intergenerational equality issues.

**Appearance Shaming:** e.g., body shaming, insulting looks, imposing beauty standards, reflecting biases based on appearance.

**Cyberbullying:** e.g., personal attacks, malicious rumors, doxing, focusing on harmful behaviors in online environments.

**Hate Speech:** e.g., extremism, incitement to violence, group-based hatred, involving issues of social safety and harmony.

**# The Sentence is:**

{Input Corresponding Sentence}

*Model Output:*

**Acknowledged. I will proceed with the task. Based on the input sentence {Input Corresponding Sentence}, the most matching concept is: {Selected Concept}.**

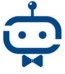

Figure 9: Civil Comment Sentence-level Concept Label Labeling Prompt

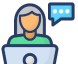

**(C2) Civil Comments Word Extraction Template**

*User Input:*

# You are an expert in text concept extraction. Based on the text content of the Civil_Comment dataset, please read the entire sentence and classify the words appearing in the comment text into one of the following categories. Assign only one category per comment.

# Note:

(1) You must choose from the provided categories below and must not generate results outside these categories.

(2) For each data entry, provide the category name that corresponds to certain words in the entire text, using English.

(3) Do not output any additional content—only output a single English category. If multiple categories apply, choose the most important and obvious one. No explanations or reasons.

# All categories and their descriptions are as follows:

**Racial Identifiers:** e.g., Black, Latino, Asian, involving references to and discussions of racial identity.

**Religious Terms:** e.g., Muslim, Christian, atheist, focusing on the naming and description of faith groups.

**Gender Pronouns:** e.g., she, he, they, reflecting language use related to gender identity.

**Political Terms:** e.g., liberal, conservative, socialist, involving expressions of political stance.

**Sexuality Terms:** e.g., gay, bisexual, queer, related to identity expressions of the LGBTQ+ community.

**Disability Terms:** e.g., wheelchair user, hearing impaired, autistic, focusing on terms referring to people with disabilities.

**Age Terms:** e.g., elderly, millennial, teenager, involving references to generational groups.

**Appearance Terms:** e.g., obese, bald, tattooed, focusing on comments about physical characteristics.

**Geographic Terms:** e.g., Mexican, Middle Eastern, African, related to expressions of regional identity.

**Occupational Terms:** e.g., police officer, teacher, migrant worker, reflecting profession-related identity references.

# The Sentence is:

{Input Corresponding Sentence}

---

*Model Output:*

Acknowledged. I will proceed with the task. Based on the input sentence {Input Corresponding Sentence}, the most matching concept is: {Selected Concept}.

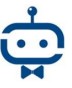

Figure 10: Civil Comment Core Word-level Concept Label Labeling Prompt

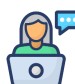

**(D1) Empathetic Dialogues Concept Sentence Extraction Template**

*User Input:*

# You are an expert in text concept extraction. Please read the complete sentence of the text content from the Empathetic Dialogue dataset and classify the entire review text into one of the following review categories. Assign only one category per review.

# Note:

(1) You must choose from the provided categories below and must not generate results outside these categories.

(2) Provide the category that the entire text content of each data entry belongs to, using English.

(3) Do not output any additional content—only output a single English category. No explanations or reasons.

# All categories and their descriptions are as follows:

**Daily Frustrations:** e.g., daily annoyances, minor setbacks, trivial troubles, focusing on common life issues.

**Joy Sharing:** e.g., happy moments, sharing successes, positive experiences, concerning positive emotions.

**Loss & Grief:** e.g., losing loved ones, sad experiences, mourning processes, involving emotional loss.

**Relationship Conflict:** e.g., interpersonal conflicts, arguments, emotional issues, related to relationship disputes.

**Career & Study:** e.g., work pressure, academic challenges, career development, targeting professional and educational matters.

**Accidents & Trauma:** e.g., unexpected events, traumatic experiences, emergencies, emphasizing sudden misfortunes.

**Life Transitions:** e.g., life changes, major decisions, transitional phases, covering life turning points.

**Health Crisis:** e.g., health crises, illness struggles, medical emergencies, focusing on health problems.

**Social Concerns:** e.g., social issues, public events, moral topics, involving social awareness.

**Moral Dilemmas:** e.g., ethical dilemmas, moral choices, value conflicts, related to ethical considerations.

# The Sentence is:

{Input Corresponding Sentence}

---

*Model Output:*

Acknowledged. I will proceed with the task. Based on the input sentence {Input Corresponding Sentence}, the most matching concept is: {Selected Concept}.

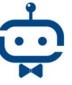

Figure 11: Empathetic Dialogues Sentence-level Concept Label Labeling Prompt

**(D2) Empathetic Dialogues Concept Word Extraction Template**

*User Input:*

# You are an expert in text concept extraction. Based on the text content of the Empathetic Dialogue dataset, please read the entire sentence and classify the words appearing in the review text into one of the following review categories. Assign only one category per review.

# Note:

(1) You must choose from the provided categories below and must not generate results outside these categories.

(2) For each data entry, provide the category name that corresponds to certain words in the entire text, using English.

(3) Do not output any additional content—only output a single English category. If multiple categories apply, choose the most important and obvious one. No explanations or reasons.

# All categories and their descriptions are as follows:

**Sadness:** e.g., sad, unhappy, grief, involving emotions of sadness.

**Hope:** e.g., hope, optimistic, wish, involving hope and optimism.

**Anger:** e.g., angry, mad, furious, involving emotions of anger.

**Fear:** e.g., fear, scared, afraid, involving emotions of fear.

**Love:** e.g., love, affection, care, involving love and emotional attachment.

**Shame:** e.g., shame, embarrassed, guilty, involving emotions of shame.

**Gratitude:** e.g., thankful, grateful, appreciation, involving expressions of gratitude.

**Pain:** e.g., pain, hurt, suffering, involving physical or emotional pain.

**Shock:** e.g., shock, surprised, amazed, involving reactions of shock.

**Nostalgia:** e.g., nostalgia, memory, past, involving nostalgic feelings.

# The Sentence is:

{Input Corresponding Sentence}

*Model Output:*

Acknowledged. I will proceed with the task. Based on the input sentence {Input Corresponding Sentence}, the most matching concept is: {Selected Concept}.

Figure 12: Empathetic Dialogues Core Word-level Concept Label Labeling Prompt

**(E1) IMDB Concept Sentence Extraction Template**

*User Input:*

# You are an expert in text concept extraction. Please read the complete sentence of the text content from the IMDB movie review dataset and classify the entire review text into one of the following review categories. Assign only one category per review.

# Note:

(1) You must choose from the provided categories below and must not generate results outside these categories.

(2) Provide the category that the entire text content of each data entry belongs to, using English.

(3) Do not output any additional content—only output a single English category. No explanations or reasons.

# All categories and their descriptions are as follows:

**Comedy:** e.g., humorous plots, funny dialogues, lighthearted atmosphere, focusing on entertainment.

**Horror:** e.g., horror elements, scary scenes, suspenseful ambiance, concerning fear experiences.

**Drama:** e.g., emotional conflicts, profound stories, character development, involving dramatic elements.

**Action:** e.g., action sequences, adventurous plots, intense fights, emphasizing dynamism.

**Romance:** e.g., love stories, emotional relationships, romantic moments, related to emotional expression.

**Biography:** e.g., real-life stories, historical events, life accounts, covering biographical content.

**Science Fiction:** e.g., sci-fi settings, futuristic technology, alien elements, focusing on imaginative worlds.

**Documentary:** e.g., real records, factual reporting, educational content, involving documentary style.

**Mystery:** e.g., puzzle-solving, detective stories, suspenseful plots, emphasizing reasoning processes.

**Animation:** e.g., animation effects, cartoon characters, family-friendly content, targeting animated media.

# The Sentence is:

{Input Corresponding Sentence}

*Model Output:*

Acknowledged. I will proceed with the task. Based on the input sentence {Input Corresponding Sentence}, the most matching concept is: {Selected Concept}.

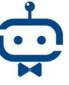

Figure 13: IMDB Sentence-level Concept Label Labeling Prompt

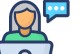

**(E2) IMDB Concept Word Extraction Template**

*User Input:*

# You are an expert in text concept extraction. Based on the text content of the IMDB dataset, please read the entire sentence and classify the words appearing in the review text into one of the following review categories. Assign only one category per review.

# Note:

(1) You must choose from the provided categories below and must not generate results outside these categories.

(2) For each data entry, provide the category name that corresponds to certain words in the entire text, using English.

(3) Do not output any additional content—only output a single English category. If multiple categories apply, choose the most important and obvious one. No explanations or reasons.

# All categories and their descriptions are as follows:

**Film Terms:** e.g., film, movie, cinema, involving movie terminology.

**Film Titles:** e.g., Avatar, Titanic, title, involving movie titles.

**Actors/Actresses:** e.g., actor, actress, star, involving performers.

**Directors:** e.g., director, filmmaker, auteur, involving directors.

**Genres:** e.g., comedy, drama, horror, involving movie genres.

**Cultural Impact Terms:** e.g., cultural, impact, influence, involving cultural impact.

**Awards:** e.g., Oscar, award, prize, involving awards and honors.

**Franchises/IPs:** e.g., franchise, IP, series, involving film series and intellectual property.

**Studios:** e.g., studio, production company, Hollywood, involving film studios.

**Iconic Characters:** e.g., character, hero, villain, involving iconic characters.

# The Sentence is:

{Input Corresponding Sentence}

*Model Output:*

Acknowledged. I will proceed with the task. Based on the input sentence {Input Corresponding Sentence}, the most matching concept is: {Selected Concept}.

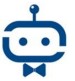

Figure 14: IMDB Core Word-level Concept Label Labeling Prompt

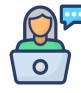

**(F1) Rotten Tomatoes Concept Sentence Extraction Template**

*User Input:*

# You are an expert in text concept extraction. Please read the complete sentence of the text content from the Rotten Tomatoes dataset and classify the entire review text into one of the following review categories. Assign only one category per review.

# Note:

(1) You must choose from the provided categories below and must not generate results outside these categories.

(2) Provide the category that the entire text content of each data entry belongs to, using English.

(3) Do not output any additional content—only output a single English category. No explanations or reasons.

# All categories and their descriptions are as follows:

**Comedy:** e.g., humorous elements, joke design, comedic effects, focusing on entertainment.

**Thriller:** e.g., tense atmosphere, suspenseful plots, thrilling moments, concerning suspense.

**Romance:** e.g., love themes, emotional depth, romantic expression, involving relationship portrayal.

**Documentary:** e.g., real stories, factual presentation, educational value, emphasizing documentary style.

**Drama:** e.g., emotional conflicts, character arcs, serious themes, covering dramatic elements.

**Action:** e.g., action sequences, adventurous experiences, physical conflicts, related to dynamic scenes.

**Fantasy:** e.g., fantasy worlds, magical elements, imaginative settings, focusing on the surreal.

**Horror:** e.g., horror imagery, fear induction, scare factors, involving horror experiences.

**Animation:** e.g., animation techniques, family entertainment, visual styles, targeting animated content.

**Crime:** e.g., crime stories, detective work, legal issues, emphasizing crime themes.\

# The Sentence is:

{Input Corresponding Sentence}

*Model Output:*

Acknowledged. I will proceed with the task. Based on the input sentence {Input Corresponding Sentence}, the most matching concept is: {Selected Concept}.

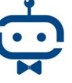

Figure 15: Rotten Tomato Sentence-level Concept Label Labeling Prompt

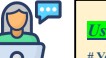

**(F2) Rotten Tomatoes Concept Word Extraction Template**

*User Input:*

\# You are an expert in text concept extraction. Based on the text content of the **Rotten Tomatoes** dataset, please read the entire sentence and classify the words appearing in the review text into one of the following review categories. Assign only one category per review.

\# Note:

(1) You must choose from the provided categories below and must not generate results outside these categories.

(2) For each data entry, provide the category name that corresponds to certain words in the entire text, using English.

(3) Do not output any additional content—only output a single English category. If multiple categories apply, choose the most important and obvious one. No explanations or reasons.

\# All categories and their descriptions are as follows:

**Film Terms:** e.g., film, movie, review, involving movie terminology.

**Actors/Actresses:** e.g., actor, performance, star, involving actors.

**Genres:** e.g., genre, comedy, drama, involving movie genres.

**Film Titles:** e.g., title, movie name, film, involving movie titles.

**Cultural Impact Terms:** e.g., cultural, social, impact, involving cultural impact.

**Directors:** e.g., director, filmmaker, involving directors.

**Awards:** e.g., award, Oscar, nomination, involving awards.

**Franchises/IPs:** e.g., franchise, series, IP, involving film series.

**Iconic Characters:** e.g., character, iconic, hero, involving iconic characters.

**Studios:** e.g., studio, production, company, involving studios.

\# The Sentence is:

{Input Corresponding Sentence}

*Model Output:*

Acknowledged. I will proceed with the task. Based on the input sentence {Input Corresponding Sentence}, the most matching concept is: {Selected Concept}.

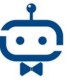

Figure 16: Rotten Tomato Core Word-level Concept Label Labeling Prompt

**(G1) TweetEval Concept Sentence Extraction Template**

*User Input:*

\# You are an expert in text concept extraction. Based on the text content of the Tweet dataset, please read the entire sentence and classify each complete text entry into one of the following categories. Assign only one category per tweet.

\# Note:

(1) You must choose from the provided categories below and must not generate results outside these categories.

(2) Provide the category that the entire text content of each data entry belongs to, using English.

(3) Do not output any additional content—only output a single English category. No explanations or reasons.

\# All categories and their descriptions are as follows:

**Social Trends:** e.g., vaccine controversy, climate strike, metaverse, focusing on real-time discussions and clashing viewpoints on current topics.

**Celebrity Culture:** e.g., celebrity scandals, influencer livestreams, fan wars, concerning public figures and the舆论 storms they generate.

**Political Battles:** e.g., election fraud, parliament brawls, diplomatic rhetoric, reflecting sharp ideological confrontations.

**Internet Memes:** e.g., meme flooding, emoji wars, viral challenges, representing the rapid spread of subculture phenomena.

**Brand Drama:** e.g., advertising failures, CEO gaffes, product recalls, related to public scrutiny of corporate actions.

**Sports Frenzy:** e.g., referee corruption, player transfers, esports showdowns, showcasing passionate fan engagements in sports.

**Disaster Updates:** e.g., earthquake live reports, terror attack scenes, pandemic data, involving real-time information flow about emergencies.

**Culture Wars:** e.g., gender pronouns, historical statues, censorship, triggering intense debates over values.

**Tech Debates:** e.g., algorithm bias, data leaks, AI threats, examining ethical boundaries of technological development.

**Personal Outbursts:** e.g., workplace rants, breakup announcements, midnight emo, presenting fragmented emotional outbursts.

\# The Sentence is:

{Input Corresponding Sentence}

*Model Output:*

Acknowledged. I will proceed with the task. Based on the input sentence {Input Corresponding Sentence}, the most matching concept is: {Selected Concept}.

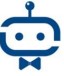

Figure 17: TweetEval Sentence-level Concept Label Labeling Prompt

**(G2) TweetEval Concept Word Extraction Template**

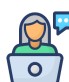

*User Input:*

\# You are an expert in text concept extraction. Based on the text content of the Tweet dataset, please read the entire sentence and classify the words appearing in the text into one of the following categories. Assign only one category per tweet.

\# Note:

(1) You must choose from the provided categories below and must not generate results outside these categories.

(2) For each data entry, provide the category name that corresponds to certain words in the entire text, using English.

(3) Do not output any additional content—only output a single English category. If multiple categories apply, choose the most important and obvious one. No explanations or reasons.

\# All categories and their descriptions are as follows:

**Celebrity Handles:** e.g., @elonmusk, @taylorswift13, focusing on mentions and interactions with public figures accounts.

**Sports Teams:** e.g., #Lakers, @ManUtd, concerning discussions about sports events and teams.

**Brand Accounts:** e.g., @Nike, @Tesla, involving discussions related to commercial brands.

**Politicians:** e.g., @POTUS, @narendramodi, reflecting discussions about political figures and policies.

**Trending Hashtags:** e.g., #COVID19, #BlackLivesMatter, tracking hot topics and social media movements.

**Internet Slang:** e.g., LOL, SMH, FOMO, representing characteristic expressions of internet culture.

**Abbreviations:** e.g., TBH, IMO, DM, demonstrating social media-specific shorthand.

**Emojis:** e.g., 😂, ❤️, 😅, conveying emotions and attitudes through visual symbols.

**Media Outlets:** e.g., @CNN, @BBC, relating to news organizations and media coverage.

**Location Tags:** e.g., #NYC, @Tokyo, marking geographical locations and regional events.

\# The Sentence is:

{Input Corresponding Sentence}

*Model Output:*

Acknowledged. I will proceed with the task. Based on the input sentence {Input Corresponding Sentence}, the most matching concept is: {Selected Concept}.

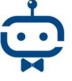

Figure 18: TweetEval Core Word-level Concept Label Labeling Prompt

**(H1) Yahoo! Answers  Concept Sentence Extraction Template**

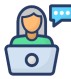

*User Input:*

\# You are an expert in text concept extraction. Based on the text content of the Yahoo comment dataset, please read the entire sentence and classify each complete text entry into one of the following categories. Assign only one category per comment.

\# Note:

(1) You must choose from the provided categories below and must not generate results outside these categories.

(2) Provide the category that the entire text content of each data entry belongs to, using English.

(3) Do not output any additional content—only output a single English category. No explanations or reasons.

\# All categories and their descriptions are as follows:

**Political Controversy:** e.g., election disputes, policy debates, international conflicts, focusing on ideological opposition and public policy discussions.

**Financial Trends:** e.g., stock market fluctuations, corporate earnings, economic indicators, concerning business development and investment decisions.

**Tech Updates:** e.g., product launches, data breaches, AI advancements, involving technological innovation and digital privacy.

**Celebrity Gossip:** e.g., celebrity scandals, award ceremonies, film reviews, tracking entertainment industry hotspots.

**Sports Events:** e.g., game results, player transfers, competition scandals, reflecting sports dynamics.

**Health Information:** e.g., medical breakthroughs, disease prevention, wellness tips, focusing on public health issues.

**Crime Reports:** e.g., violent cases, corruption investigations, judicial rulings, involving social security matters.

**Education Debates:** e.g., admission policies, campus incidents, academic controversies, relating to education system discussions.

**Environmental Issues:** e.g., climate change, energy policies, ecological conservation, exploring sustainable development.

**Life Advice:** e.g., interpersonal relationships, career planning, psychological adjustment, providing practical life guidance.

\# The Sentence is:

{Input Corresponding Sentence}

*Model Output:*

Acknowledged. I will proceed with the task. Based on the input sentence {Input Corresponding Sentence}, the most matching concept is: {Selected Concept}.

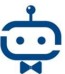

Figure 19: Yahoo! Sentence-level Concept Label Labeling Prompt

**(H2) Yahoo! Answers Concept Word Extraction Template**

*User Input:*

# You are an expert in text concept extraction. Based on the text content of the Yahoo dataset, please read the entire sentence and classify the words appearing in the text into one of the following categories. Assign only one category per data entry.

# Note:

(1) You must choose from the provided categories below and must not generate results outside these categories.

(2) For each data entry, provide the category name that corresponds to certain words in the entire text, using English.

(3) Do not output any additional content—only output a single English category. If multiple categories apply, choose the most important and obvious one. No explanations or reasons.

# All categories and their descriptions are as follows:

**Political Figures & Institutions:** e.g., Trump, Biden, White House, Congress, focusing on discussions of government officials and political entities.

**Corporate Brands:** e.g., Apple, Tesla, Google, Amazon, concerning developments and market impact of business giants.

**Sports Teams & Athletes:** e.g., Lakers, Messi, NBA, Premier League, reflecting discussions about sports events and star athletes.

**Celebrity Names:** e.g., Taylor Swift, Beyoncé, Marvel, Oscars, tracking entertainment figures and works.

**Financial Terms:** e.g., stock price, inflation, Federal Reserve, Bitcoin, involving economic indicators and investment topics.

**Tech Terminology:** e.g., AI, metaverse, 5G, data breach, focusing on technological innovation and digital issues.

**Global Hotspots:** e.g., Ukraine, Middle East, North Korea, EU, discussing geopolitics and international relations.

**Social Movements:** e.g., MeToo, BLM, environmental protection, LGBTQ, reflecting topics of social equality and rights.

**Health Keywords:** e.g., vaccine, COVID-19, depression, health insurance, focusing on medical and health issues.

**Internet Slang:** e.g., yyds, break defense, eating melons (online spectating), representing characteristic expressions of internet culture.

# The Sentence is:

{Input Corresponding Sentence}

*Model Output:*

Acknowledged. I will proceed with the task. Based on the input sentence {Input Corresponding Sentence}, the most matching concept is: {Selected Concept}.

Figure 20: Yahoo! Core Word-level Concept Label Labeling Prompt

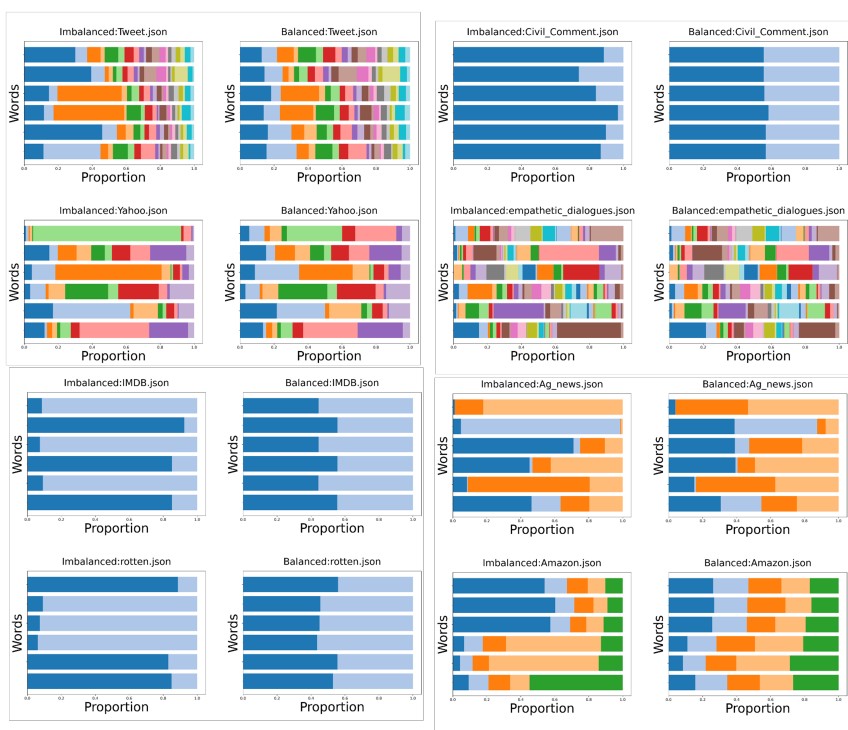

Figure 21: The Label Distribution within Balanced and Imbalanced Datasets of SCS-Type Spurious Correlations

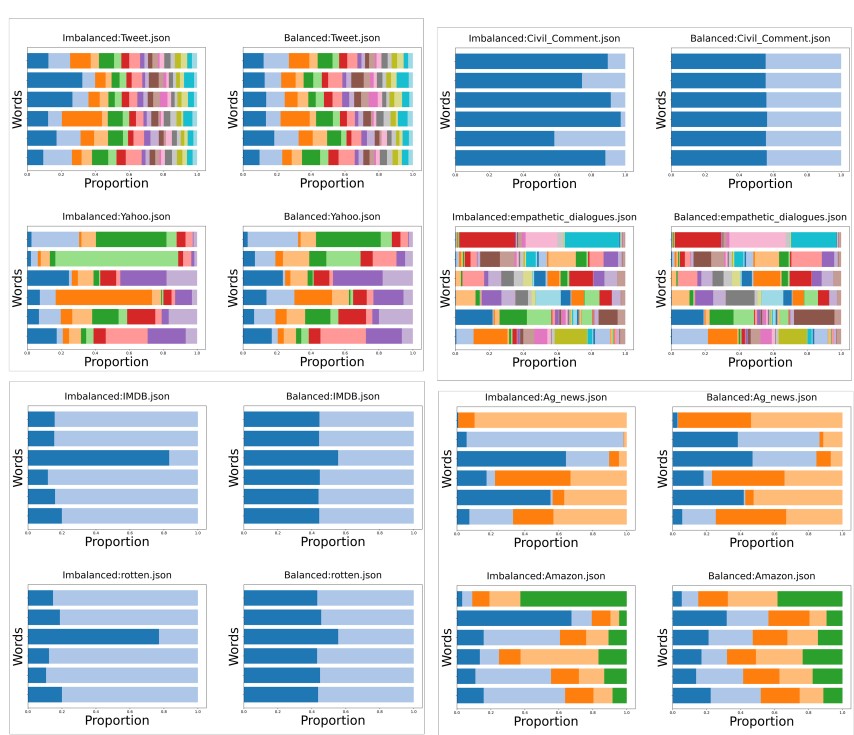

Figure 22: The Label Distribution within Balanced and Imbalanced Datasets of CCS-Type Spurious Correlations

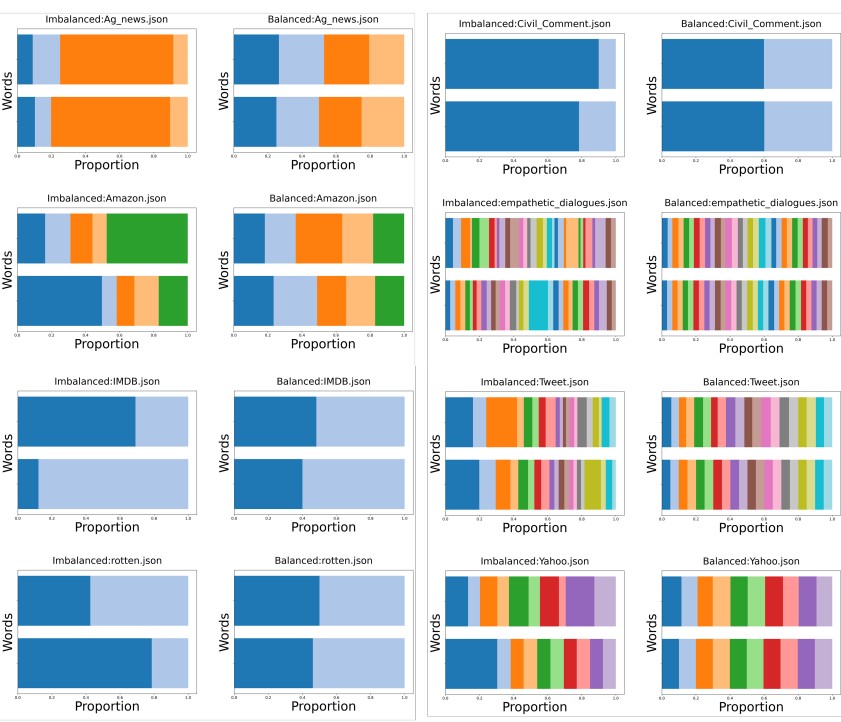

Figure 23: The Label Distribution within Balanced and Imbalanced Datasets of NBS-Type Spurious Correlations

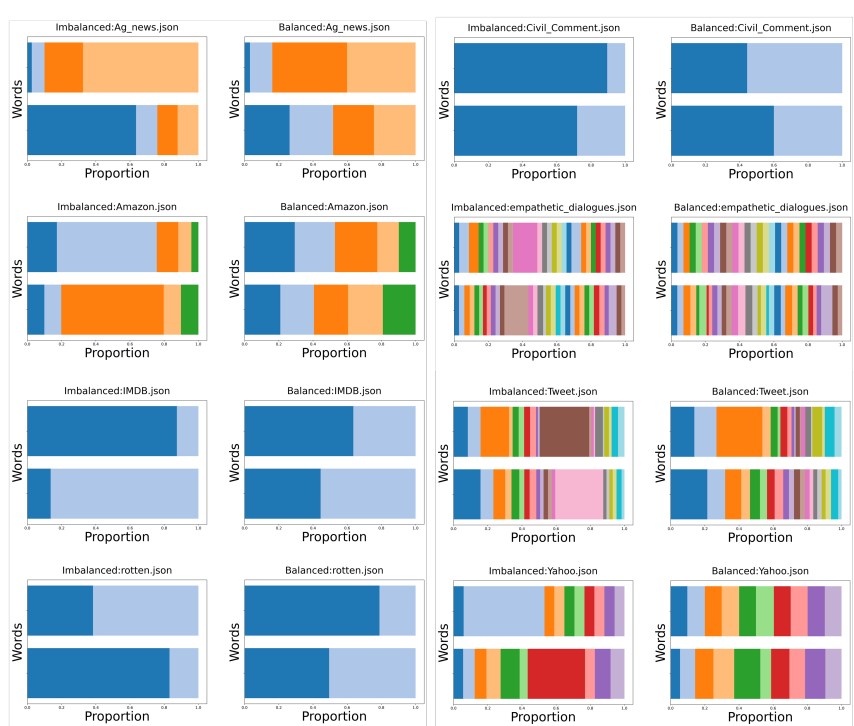

Figure 24: The Label Distribution within Balanced and Imbalanced Datasets of QBS-Type Spurious Correlations

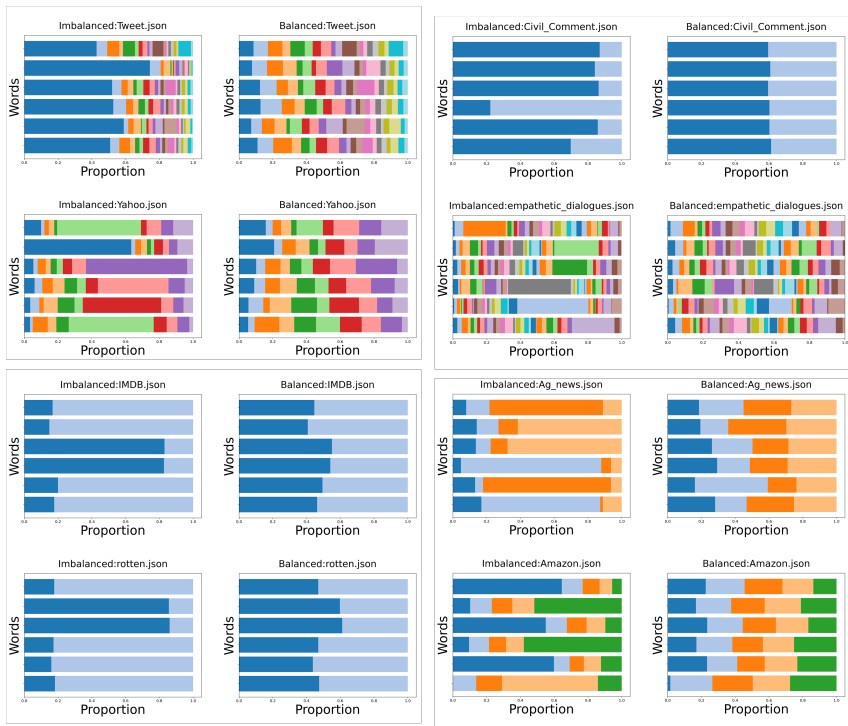

Figure 25: The Label Distribution within Balanced and Imbalanced Datasets of WFS-Type Spurious Correlations

| Datasets Name | | Spurious Correlation Label Name | | | | | |
|---|---|---|---|---|---|---|---|
| | | $a_1$ | $a_2$ | $a_3$ | $a_4$ | $a_5$ | $a_6$ |
| Ag News | SCS | Sports | Economy | Politics | Technology | Crime | Environment |
| | CCS | Companies Brands | Politicians | Finance Terms | Sports Terms | Tech Products | Environmental Terms |
| | NBS | $With_N$ | $Without_N$ | - | - | - | - |
| | QBS | $With_Q$ | $Without_Q$ | - | - | - | - |
| | WFS | ap | gt | quot | reuters | world | year |
| Amazon | SCS | Home Kitchen | Electronics Review | Clothing Shoes | Grocery Gourmet | Books Media | Beauty Personal Care |
| | CCS | Product Size | User Experience | Material Quality | Price Value | Shipping Delivery | Color Appearance |
| | NBS | $With_N$ | $Without_N$ | - | - | - | - |
| | QBS | $With_Q$ | $Without_Q$ | - | - | - | - |
| | WFS | product | quality | stars | time | use | work |
| Civil Comment | SCS | Hate Speech | Political Polarization | Gender Bias | Appearance Shaming | Disability Discrimination | Ageism |
| | CCS | Political Terms | Geographic Terms | Religious Terms | Age Terms | Occupational Terms | Appearance Terms |
| | NBS | $With_N$ | $Without_N$ | - | - | - | - |
| | QBS | $With_Q$ | $Without_Q$ | - | - | - | - |
| | WFS | government | people | state | time | trump | years |
| Empathetic Dialogues | SCS | Daily Frustrations | Joy Sharing | Loss Grief | Relationship Conflict | Career Study | Accidents Trauma |
| | CCS | Sadness | Hope | Anger | Fear | Love | Shame |
| | NBS | $With_N$ | $Without_N$ | - | - | - | - |
| | QBS | $With_Q$ | $Without_Q$ | - | - | - | - |
| | WFS | day | friend | going | time | week | work |
| IMDB | SCS | Comedy | Horror | Drama | Action | Romance | Biography |
| | CCS | Film Terms | Film Titles | Actors Actresses | Directors | Genres | Cultural Impact Terms |
| | NBS | $With_N$ | $Without_N$ | - | - | - | - |
| | QBS | $With_Q$ | $Without_Q$ | - | - | - | - |
| | WFS | film | good | like | movie | story | time |
| Rotten Tomato | SCS | Comedy | Thriller | Romance | Documentary | Drama | Action |
| | CCS | Film Terms | Actors Actresses | Genres | Film Titles | Cultural Impact Terms | Directors |
| | NBS | $With_N$ | $Without_N$ | - | - | - | - |
| | QBS | $With_Q$ | $Without_Q$ | - | - | - | - |
| | WFS | comedy | film | good | like | movie | story |
| Tweeteval | SCS | Celebrity Culture | Sports Frenzy | Personal Outbursts | Social Trends | Internet Memes | Brand Drama |
| | CCS | Location Tags | Internet Slang | Trending Hashtags | Media Outlets | Celebrity Handles | Brand Accounts |
| | NBS | $With_N$ | $Without_N$ | - | - | - | - |
| | QBS | $With_Q$ | $Without_Q$ | - | - | - | - |
| | WFS | amp | california | day | happy | love | user |
| Yahoo! | SCS | Life Advice | Financial Trends | Celebrity Gossip | Health Information | Sports Events | Environmental Issues |
| | CCS | Financial Terms | Celebrity Names | Health Keywords | Sports Teams | Tech Terminology | Social Movements |
| | NBS | $With_N$ | $Without_N$ | - | - | - | - |
| | QBS | $With_Q$ | $Without_Q$ | - | - | - | - |
| | WFS | best | find | know | like | people | think |

Table 14: **List of Spurious Correlation Label Values ($a$) in MPS**

| Model | Ag News Imbalanced | Ag News Balance | Amazon Imbalanced | Amazon Balance | Civil C Imbalanced | Civil C Balance | Empathetic D Imbalanced | Empathetic D Balance | IMDB Imbalanced | IMDB Balance | Rotten T Imbalanced | Rotten T Balance | TweetEval Imbalanced | TweetEval Balance | Yahoo! Imbalanced | Yahoo! Balance |
|---|---|---|---|---|---|---|---|---|---|---|---|---|---|---|---|---|
| **SCS** | | | | | | | | | | | | | | | | |
| **BART** | 73.33% | 76.40% | 41.37% | 49.92% | 78.88% | 86.23% | 22.05% | 22.09% | 91.24% | 91.82% | 84.86% | 86.04% | 0.00% | 0.00% | 37.93% | 41.03% |
| **BERT** | 75.32% | 78.38% | 42.63% | 51.07% | 80.90% | 86.07% | 17.04% | 18.87% | 88.92% | 92.09% | 73.47% | 83.06% | 0.00% | 0.00% | 37.33% | 41.29% |
| **DistilBERT** | 69.55% | 71.71% | 37.93% | 51.65% | 77.30% | 86.20% | 13.58% | 9.06% | 85.43% | 90.40% | 73.54% | 80.00% | 0.00% | 0.00% | 33.45% | 40.17% |
| **DeBERTa-v3** | 69.37% | 74.59% | 45.39% | 53.41% | 80.15% | 85.00% | 15.61% | 22.93% | 92.16% | 93.67% | 80.42% | 88.26% | 0.00% | 0.00% | 39.74% | 44.22% |
| **RoBERTa** | 70.27% | 76.94% | 39.79% | 51.17% | 81.35% | 86.61% | 15.61% | 25.50% | 90.78% | 93.82% | 80.42% | 85.83% | 0.00% | 0.00% | 39.14% | 38.28% |
| **ETC** | 68.11% | 75.86% | 40.48% | 49.86% | 78.20% | 86.46% | 8.51% | 16.60% | 88.33% | 91.83% | 74.17% | 82.80% | 0.00% | 0.00% | 37.16% | 38.53% |
| **TCM** | 67.18% | 47.20% | 28.62% | 44.40% | 81.80% | 85.54% | 0.00% | 0.00% | 80.42% | 86.42% | 5.69% | 50.54% | 0.00% | 0.00% | 2.50% | 0.00% |
| **XLnet** | 72.97% | 70.63% | 38.86% | 50.87% | 77.08% | 82.97% | 17.06% | 25.01% | 90.88% | 92.99% | 82.92% | 85.90% | 0.00% | 0.00% | 38.97% | 40.00% |
| **gpt2** | 74.41% | 73.51% | 26.21% | 45.37% | 77.75% | 86.74% | 16.68% | 14.72% | 85.36% | 89.98% | 75.56% | 86.49% | 0.00% | 0.00% | 36.90% | 39.66% |
| **llama2** | 72.43% | 76.22% | 40.15% | 52.03% | 69.89% | 86.43% | 25.93% | 29.10% | 91.42% | 93.12% | 82.43% | 89.51% | 0.00% | 2.99% | 45.69% | 47.95% |
| **Qwen3-8b** | 73.16% | 76.94% | 38.86% | 50.96% | 71.91% | 86.25% | 20.24% | 25.74% | 91.75% | 94.24% | 82.85% | 89.31% | 0.54% | 5.01% | 44.42% | 44.74% |
| **TF-IDF+SVM** | 73.87% | 71.17% | 28.72% | 43.95% | 60.67% | 79.78% | 21.95% | 17.07% | 80.16% | 87.41% | 64.24% | 71.53% | 0.00% | 0.00% | 39.62% | 42.17% |
| **CCS** | | | | | | | | | | | | | | | | |
| **BART** | 87.23% | 88.61% | 41.26% | 50.64% | 83.83% | 86.57% | 0.00% | 0.00% | 92.97% | 93.58% | 84.66% | 83.26% | 0.93% | 0.93% | 46.51% | 49.11% |
| **BERT** | 86.33% | 87.33% | 43.48% | 50.70% | 78.51% | 86.26% | 0.00% | 5.00% | 92.25% | 92.33% | 81.12% | 79.57% | 0.00% | 0.00% | 48.84% | 49.22% |
| **DistilBERT** | 82.81% | 84.48% | 43.08% | 50.57% | 81.70% | 85.09% | 0.00% | 0.00% | 91.27% | 91.45% | 79.31% | 82.06% | 0.00% | 0.00% | 48.31% | 47.65% |
| **DeBERTa-v3** | 85.71% | 87.59% | 46.25% | 53.39% | 84.04% | 84.20% | 0.00% | 6.25% | 94.51% | 95.15% | 89.34% | 88.82% | 0.00% | 0.00% | 47.50% | 47.40% |
| **RoBERTa** | 87.09% | 88.71% | 43.86% | 51.67% | 85.11% | 84.20% | 0.00% | 3.75% | 94.29% | 94.29% | 87.90% | 82.92% | 0.00% | 1.57% | 49.68% | 49.48% |
| **ETC** | 87.48% | 87.48% | 40.83% | 48.33% | 79.57% | 80.50% | 0.00% | 10.36% | 92.17% | 93.48% | 81.41% | 80.94% | 0.00% | 0.00% | 45.73% | 45.82% |
| **TCM** | 75.34% | 71.78% | 32.51% | 45.90% | 81.49% | 87.90% | 0.00% | 0.00% | 85.29% | 84.00% | 62.15% | 66.01% | 0.00% | 0.00% | 1.43% | 0.00% |
| **XLnet** | 88.30% | 87.71% | 41.95% | 49.53% | 82.98% | 81.85% | 0.00% | 8.65% | 93.70% | 94.25% | 87.29% | 88.23% | 0.00% | 0.00% | 46.25% | 46.61% |
| **gpt2** | 87.73% | 87.21% | 38.42% | 44.05% | 77.02% | 84.16% | 0.00% | 3.75% | 93.20% | 89.06% | 82.73% | 82.15% | 0.00% | 0.00% | 45.94% | 45.53% |
| **llama2** | 87.94% | 88.08% | 44.24% | 50.22% | 71.70% | 84.07% | 8.75% | 3.75% | 93.98% | 94.50% | 93.55% | 92.45% | 0.96% | 2.96% | 50.36% | 52.01% |
| **Qwen3-8b** | 86.77% | 87.36% | 42.72% | 46.93% | 70.21% | 86.17% | 10.00% | 8.47% | 93.91% | 94.10% | 89.53% | 87.90% | 0.94% | 4.74% | 48.18% | 49.00% |
| **TF-IDF+SVM** | 77.91% | 77.30% | 33.33% | 42.40% | 63.44% | 76.34% | 11.11% | 0.00% | 87.12% | 87.40% | 70.61% | 73.82% | 0.00% | 0.00% | 44.27% | 40.60% |
| **NBS** | | | | | | | | | | | | | | | | |
| **BART** | 77.48% | 88.65% | 39.69% | 54.72% | 82.98% | 88.82% | 9.88% | 11.65% | 93.96% | 93.93% | 85.79% | 82.45% | 0.43% | 10.32% | 43.32% | 49.14% |
| **BERT** | 75.70% | 87.30% | 42.51% | 53.26% | 80.23% | 90.37% | 8.41% | 2.72% | 91.73% | 92.84% | 79.92% | 83.24% | 0.00% | 7.67% | 42.92% | 48.30% |
| **DistilBERT** | 73.06% | 87.52% | 39.63% | 53.56% | 78.57% | 90.50% | 0.87% | 9.13% | 92.01% | 92.92% | 77.59% | 83.92% | 0.00% | 6.92% | 40.27% | 46.09% |
| **DeBERTa-v3** | 79.96% | 87.70% | 41.53% | 53.06% | 83.42% | 89.20% | 10.83% | 12.62% | 94.64% | 94.88% | 86.90% | 88.81% | 0.00% | 10.33% | 45.71% | 46.21% |
| **RoBERTa** | 74.58% | 89.12% | 45.12% | 55.00% | 83.39% | 89.79% | 12.43% | 7.18% | 94.08% | 94.12% | 83.84% | 83.84% | 0.00% | 9.35% | 44.35% | 50.19% |
| **ETC** | 75.72% | 86.91% | 39.25% | 53.83% | 81.56% | 89.49% | 8.12% | 5.63% | 92.39% | 92.45% | 80.84% | 82.37% | 0.00% | 6.80% | 41.16% | 47.70% |
| **TCM** | 58.74% | 82.16% | 19.96% | 43.72% | 76.52% | 90.36% | 0.00% | 0.00% | 86.87% | 84.52% | 0.00% | 54.09% | 0.00% | 0.00% | 0.00% | 1.07% |
| **XLnet** | 73.02% | 88.05% | 40.23% | 52.31% | 79.13% | 90.10% | 10.43% | 20.29% | 87.16% | 90.97% | 75.84% | 82.58% | 0.00% | 8.73% | 43.40% | 48.29% |
| **gpt2** | 73.02% | 86.83% | 36.16% | 52.19% | 79.13% | 90.10% | 10.43% | 20.29% | 87.16% | 90.97% | 75.84% | 82.58% | 0.00% | 7.52% | 40.11% | 44.40% |
| **llama2** | 75.05% | 90.94% | 42.18% | 53.26% | 77.65% | 89.45% | 17.99% | 28.85% | 93.99% | 94.69% | 87.24% | 90.21% | 3.09% | 9.44% | 45.98% | 47.50% |
| **Qwen3-8b** | 75.98% | 90.26% | 38.51% | 51.06% | 75.65% | 89.93% | 20.20% | 28.94% | 94.12% | 95.12% | 86.98% | 88.21% | 3.42% | 9.35% | 45.16% | 48.96% |
| **TF-IDF+SVM** | 69.85% | 85.48% | 34.56% | 44.88% | 69.43% | 86.19% | 9.80% | 22.43% | 86.49% | 88.54% | 57.22% | 71.00% | 0.00% | 6.81% | 35.48% | 40.58% |
| **QBS** | | | | | | | | | | | | | | | | |
| **BART** | 81.36% | 84.77% | 15.46% | 50.68% | 82.97% | 87.22% | 0.00% | 0.00% | 85.64% | 93.14% | 81.62% | 84.32% | 0.00% | 0.00% | 34.56% | 48.57% |
| **BERT** | 78.86% | 87.84% | 24.62% | 50.62% | 77.64% | 88.75% | 0.00% | 0.00% | 86.26% | 92.07% | 71.33% | 84.77% | 0.00% | 0.00% | 32.05% | 47.92% |
| **DistilBERT** | 82.37% | 87.50% | 25.20% | 50.84% | 77.80% | 89.81% | 0.00% | 0.00% | 84.21% | 91.20% | 62.98% | 83.18% | 0.00% | 0.00% | 31.17% | 45.45% |
| **DeBERTa-v3** | 79.06% | 85.68% | 26.81% | 56.56% | 80.64% | 87.18% | 0.00% | 0.00% | 91.27% | 95.04% | 81.65% | 87.73% | 0.00% | 0.00% | 33.63% | 49.36% |
| **RoBERTa** | 80.97% | 86.79% | 20.95% | 50.11% | 81.90% | 88.35% | 0.00% | 0.00% | 88.87% | 93.21% | 80.70% | 84.05% | 0.00% | 0.00% | 36.43% | 48.72% |
| **ETC** | 82.01% | 86.33% | 19.49% | 48.18% | 79.34% | 88.27% | 0.00% | 0.00% | 87.50% | 91.99% | 72.35% | 82.41% | 0.00% | 0.00% | 31.35% | 46.02% |
| **TCM** | 70.33% | 82.69% | 0.00% | 38.42% | 76.50% | 89.44% | 0.00% | 0.00% | 70.78% | 85.75% | 0.00% | 20.43% | 0.00% | 0.00% | 1.85% | 0.00% |
| **XLnet** | 79.06% | 85.87% | 20.29% | 44.47% | 78.13% | 88.12% | 0.00% | 0.00% | 88.43% | 94.33% | 80.16% | 87.45% | 0.00% | 0.00% | 33.04% | 44.74% |
| **gpt2** | 80.43% | 87.40% | 15.68% | 34.58% | 74.60% | 90.93% | 0.00% | 0.00% | 86.59% | 90.19% | 72.89% | 83.81% | 0.00% | 0.00% | 42.49% | 50.66% |
| **llama2** | 84.99% | 88.56% | 23.81% | 51.56% | 74.60% | 90.93% | 0.00% | 0.00% | 92.81% | 94.75% | 86.51% | 91.05% | 0.00% | 0.00% | 42.49% | 50.66% |
| **Qwen3-8b** | 85.30% | 88.35% | 20.73% | 50.14% | 76.73% | 91.14% | 0.00% | 0.00% | 92.88% | 94.28% | 84.32% | 91.11% | 0.00% | 0.00% | 38.48% | 50.33% |
| **TF-IDF+SVM** | 79.35% | 84.38% | 25.67% | 34.49% | 76.71% | 88.18% | 0.00% | 0.00% | 81.81% | 87.94% | 41.75% | 72.86% | 0.00% | 0.00% | 28.95% | 36.56% |
| **WFS** | | | | | | | | | | | | | | | | |
| **BART** | 83.01% | 89.81% | 40.65% | 50.25% | 86.94% | 89.09% | 0.00% | 0.00% | 94.05% | 94.96% | 72.28% | 84.09% | 0.00% | 0.28% | 42.87% | 46.50% |
| **BERT** | 84.91% | 89.32% | 37.23% | 47.34% | 86.16% | 88.28% | 0.00% | 0.00% | 91.64% | 93.83% | 74.79% | 82.05% | 0.00% | 0.00% | 40.88% | 49.06% |
| **DistilBERT** | 83.40% | 90.42% | 35.94% | 46.88% | 83.68% | 88.31% | 0.00% | 0.00% | 89.11% | 92.88% | 69.77% | 79.35% | 0.00% | 0.00% | 40.96% | 46.08% |
| **DeBERTa-v3** | 83.24% | 87.49% | 41.70% | 50.61% | 88.08% | 89.26% | 0.00% | 0.00% | 95.13% | 96.14% | 76.47% | 86.33% | 0.00% | 0.00% | 45.03% | 47.37% |
| **RoBERTa** | 82.94% | 89.83% | 40.04% | 49.10% | 86.20% | 88.49% | 0.00% | 0.00% | 94.24% | 95.30% | 76.84% | 80.47% | 0.00% | 0.28% | 46.52% | 46.54% |
| **ETC** | 84.94% | 89.02% | 34.42% | 47.42% | 85.61% | 89.38% | 0.00% | 0.00% | 92.95% | 94.39% | 71.53% | 78.88% | 0.00% | 0.00% | 43.67% | 43.83% |
| **TCM** | 80.49% | 82.04% | 29.08% | 41.10% | 86.61% | 89.52% | 0.00% | 0.00% | 84.18% | 88.37% | 42.60% | 33.49% | 0.00% | 0.00% | 0.00% | 0.00% |
| **XLnet** | 84.55% | 88.61% | 36.25% | 48.10% | 86.69% | 89.09% | 0.00% | 0.61% | 86.65% | 95.29% | 77.02% | 85.30% | 0.00% | 0.57% | 45.62% | 44.72% |
| **gpt2** | 83.09% | 90.15% | 29.84% | 45.57% | 84.89% | 89.44% | 0.00% | 0.00% | 86.65% | 90.21% | 71.72% | 80.09% | 0.00% | 0.00% | 38.95% | 43.66% |
| **llama2** | 86.55% | 90.15% | 36.28% | 48.42% | 80.79% | 86.34% | 2.42% | 3.64% | 95.26% | 95.84% | 89.86% | 88.84% | 1.90% | 2.56% | 41.82% | 45.55% |
| **Qwen3-8b** | 84.48% | 90.83% | 34.97% | 49.37% | 82.12% | 86.34% | 1.21% | 1.21% | 94.75% | 95.62% | 86.23% | 89.30% | 0.85% | 4.83% | 41.02% | 43.31% |
| **TF-IDF+SVM** | 80.22% | 86.08% | 21.62% | 36.49% | 76.38% | 80.55% | 0.00% | 0.00% | 85.83% | 89.75% | 49.77% | 66.05% | 0.00% | 0.00% | 31.58% | 30.36% |

Table 15: The average worst-group accuracy results of baselines on MPS

| Model | Ag News | | Amazon | | Civil C | | Empathetic D | | IMDB | | Rotten T | | TweetEval | | Yahoo! | |
|---|---|---|---|---|---|---|---|---|---|---|---|---|---|---|---|---|
| | Imbalanced | Balance | Imbalanced | Balance | Imbalanced | Balance | Imbalanced | Balance | Imbalanced | Balance | Imbalanced | Balance | Imbalanced | Balance | Imbalanced | Balance |
| **SCS** | | | | | | | | | | | | | | | | |
| **BART** | 84.49% | 86.40% | 55.41% | 61.46% | 85.97% | 87.66% | 57.07% | 57.63% | 92.35% | 93.96% | 86.97% | 87.86% | 44.54% | 44.24% | 63.70% | 63.65% |
| **BERT** | 83.48% | 85.56% | 53.72% | 59.60% | 87.56% | 88.16% | 52.56% | 53.72% | 90.36% | 93.29% | 82.90% | 85.24% | 33.58% | 31.82% | 61.21% | 61.98% |
| **DistilBERT** | 81.07% | 83.31% | 52.55% | 58.97% | 85.97% | 88.36% | 50.08% | 50.36% | 88.26% | 91.68% | 81.03% | 84.74% | 33.22% | 32.69% | 61.14% | 61.77% |
| **DeBERTa-v3** | 82.53% | 85.17% | 56.53% | 62.83% | 86.67% | 87.56% | 55.29% | 56.19% | 93.00% | 94.88% | 87.15% | 91.12% | 43.62% | 45.30% | 63.13% | 63.04% |
| **RoBERTa** | 81.29% | 84.83% | 53.55% | 61.14% | 88.36% | 89.75% | 55.76% | 56.55% | 91.91% | 94.30% | 85.27% | 87.68% | 45.04% | 45.31% | 62.37% | 62.58% |
| **ETC** | 82.02% | 85.73% | 54.08% | 59.90% | 85.77% | 87.16% | 51.17% | 53.52% | 90.44% | 93.16% | 81.59% | 84.57% | 42.11% | 44.27% | 61.43% | 61.60% |
| **TCM** | 71.18% | 72.25% | 45.83% | 54.98% | 88.96% | 86.67% | 9.00% | 12.67% | 84.27% | 87.74% | 50.55% | 64.39% | 27.12% | 26.07% | 39.43% | 36.40% |
| **XLnet** | 82.42% | 82.42% | 55.02% | 61.19% | 84.18% | 85.17% | 56.61% | 55.76% | 92.07% | 94.29% | 87.43% | 87.33% | 35.20% | 30.81% | 62.44% | 62.34% |
| **gpt2** | 83.43% | 81.80% | 51.33% | 59.21% | 86.57% | 88.86% | 48.73% | 50.56% | 88.46% | 91.15% | 83.22% | 87.54% | 40.91% | 42.60% | 61.35% | 62.14% |
| **llama2** | 81.80% | 82.70% | 55.12% | 61.38% | 83.58% | 87.56% | 56.03% | 56.65% | 92.96% | 94.37% | 88.81% | 91.22% | 43.28% | 45.04% | 64.76% | 64.25% |
| **Qwen3-8b** | 82.02% | 82.19% | 54.50% | 61.32% | 84.48% | 87.46% | 53.58% | 53.48% | 93.57% | 94.96% | 89.38% | 91.15% | 41.54% | 42.70% | 63.01% | 62.96% |
| **TF-IDF+SVM** | 78.09% | 77.25% | 44.26% | 50.27% | 79.10% | 82.59% | 42.57% | 43.24% | 84.75% | 88.20% | 71.68% | 73.45% | 29.74% | 22.49% | 53.12% | 52.47% |
| **CCS** | | | | | | | | | | | | | | | | |
| **BART** | 89.80% | 91.40% | 56.96% | 61.50% | 86.67% | 88.08% | 62.06% | 54.18% | 93.70% | 94.23% | 87.19% | 88.44% | 41.32% | 41.19% | 64.33% | 64.20% |
| **BERT** | 90.09% | 89.87% | 55.35% | 60.42% | 84.51% | 87.32% | 56.89% | 51.28% | 92.83% | 93.01% | 83.36% | 83.16% | 31.27% | 31.30% | 62.89% | 62.78% |
| **DistilBERT** | 89.04% | 88.84% | 54.11% | 59.63% | 87.79% | 86.38% | 55.23% | 50.92% | 91.96% | 92.34% | 81.91% | 84.80% | 31.28% | 30.86% | 62.49% | 62.02% |
| **DeBERTa-v3** | 89.34% | 89.67% | 57.97% | 62.40% | 90.05% | 88.17% | 60.07% | 54.68% | 95.14% | 95.38% | 90.04% | 89.92% | 39.83% | 40.14% | 63.97% | 63.35% |
| **RoBERTa** | 90.24% | 91.40% | 57.97% | 62.40% | 89.20% | 86.95% | 58.52% | 53.85% | 93.89% | 94.54% | 88.71% | 87.19% | 40.41% | 39.94% | 64.23% | 64.10% |
| **ETC** | 90.48% | 90.64% | 54.90% | 60.06% | 84.51% | 84.23% | 57.65% | 51.86% | 93.02% | 94.02% | 84.06% | 85.74% | 39.35% | 39.61% | 61.91% | 61.82% |
| **TCM** | 84.61% | 84.25% | 49.42% | 54.85% | 87.89% | 89.58% | 18.84% | 11.79% | 86.68% | 87.36% | 66.52% | 71.52% | 20.70% | 20.06% | 44.51% | 41.38% |
| **XLnet** | 90.83% | 90.50% | 57.70% | 61.90% | 86.01% | 84.79% | 57.14% | 51.18% | 94.35% | 94.65% | 88.44% | 88.95% | 32.19% | 32.43% | 62.74% | 62.88% |
| **gpt2** | 91.45% | 89.96% | 55.13% | 60.88% | 85.54% | 86.38% | 53.31% | 48.17% | 90.32% | 91.14% | 83.79% | 85.62% | 38.53% | 38.99% | 62.25% | 61.91% |
| **llama2** | 90.75% | 91.14% | 57.87% | 61.76% | 82.44% | 85.54% | 58.37% | 55.62% | 94.49% | 94.77% | 93.79% | 93.59% | 39.33% | 40.04% | 67.61% | 66.43% |
| **Qwen3-8b** | 91.32% | 90.94% | 58.03% | 61.34% | 82.63% | 87.79% | 55.41% | 50.56% | 94.44% | 94.43% | 91.02% | 91.33% | 36.18% | 37.30% | 66.19% | 65.59% |
| **TF-IDF+SVM** | 86.95% | 86.18% | 45.64% | 51.62% | 78.77% | 83.02% | 43.22% | 40.14% | 87.58% | 88.84% | 75.00% | 75.00% | 23.91% | 22.82% | 54.76% | 53.95% |
| **NBS** | | | | | | | | | | | | | | | | |
| **BART** | 90.47% | 93.17% | 61.60% | 64.68% | 88.12% | 90.80% | 56.91% | 56.81% | 94.44% | 95.07% | 87.41% | 87.67% | 33.42% | 36.52% | 65.33% | 68.22% |
| **BERT** | 89.03% | 92.16% | 60.35% | 61.85% | 87.48% | 91.25% | 53.05% | 53.94% | 92.71% | 93.83% | 84.00% | 85.58% | 25.01% | 28.71% | 64.44% | 67.28% |
| **DistilBERT** | 88.42% | 92.16% | 59.61% | 62.15% | 86.73% | 90.98% | 49.64% | 51.31% | 91.92% | 93.31% | 81.75% | 84.05% | 24.69% | 28.05% | 63.51% | 66.68% |
| **DeBERTa-v3** | 90.18% | 93.11% | 61.36% | 64.38% | 88.84% | 91.46% | 56.76% | 56.54% | 95.29% | 95.59% | 89.24% | 90.68% | 32.69% | 35.36% | 65.64% | 68.11% |
| **RoBERTa** | 89.74% | 93.16% | 61.91% | 64.30% | 88.65% | 91.33% | 56.29% | 56.57% | 94.55% | 95.05% | 86.90% | 87.65% | 32.17% | 35.34% | 65.75% | 67.98% |
| **ETC** | 89.45% | 92.59% | 60.12% | 63.84% | 87.87% | 91.18% | 53.13% | 52.64% | 93.31% | 93.94% | 83.20% | 85.84% | 30.51% | 33.96% | 65.36% | 68.89% |
| **TCM** | 78.55% | 87.70% | 53.32% | 57.58% | 85.68% | 90.57% | 10.34% | 11.23% | 87.71% | 88.84% | 42.68% | 69.50% | 13.07% | 13.57% | 41.70% | 50.88% |
| **XLnet** | 89.75% | 92.56% | 60.27% | 62.95% | 87.67% | 91.22% | 56.11% | 55.95% | 94.31% | 94.96% | 86.82% | 87.77% | 25.76% | 29.25% | 64.71% | 67.23% |
| **gpt2** | 89.14% | 92.64% | 60.19% | 63.25% | 86.98% | 91.13% | 46.23% | 48.23% | 90.28% | 91.85% | 81.67% | 84.50% | 29.92% | 33.27% | 64.07% | 67.07% |
| **llama2** | 89.58% | 94.05% | 60.17% | 64.10% | 86.46% | 90.96% | 54.75% | 55.31% | 94.88% | 95.21% | 90.05% | 91.56% | 31.79% | 34.22% | 68.50% | 70.18% |
| **Qwen3-8b** | 89.69% | 94.00% | 60.32% | 63.48% | 85.62% | 91.06% | 53.21% | 53.98% | 94.78% | 95.52% | 89.81% | 90.93% | 29.94% | 32.72% | 66.89% | 70.52% |
| **TF-IDF+SVM** | 85.43% | 89.71% | 52.88% | 54.42% | 82.21% | 88.17% | 42.35% | 43.45% | 87.83% | 89.32% | 67.68% | 72.27% | 18.41% | 20.62% | 56.42% | 59.26% |
| **QBS** | | | | | | | | | | | | | | | | |
| **BART** | 90.93% | 91.91% | 56.99% | 64.15% | 89.29% | 90.68% | 7.07% | 10.71% | 90.06% | 94.17% | 85.34% | 87.66% | 30.22% | 41.44% | 61.84% | 67.20% |
| **BERT** | 88.58% | 92.22% | 53.64% | 61.03% | 87.83% | 90.97% | 4.44% | 12.12% | 88.74% | 92.94% | 79.78% | 86.24% | 14.08% | 30.40% | 60.31% | 66.24% |
| **DistilBERT** | 89.00% | 92.11% | 51.77% | 60.43% | 87.65% | 91.22% | 4.04% | 6.46% | 87.09% | 91.93% | 74.60% | 84.70% | 14.32% | 29.50% | 60.13% | 65.13% |
| **DeBERTa-v3** | 88.37% | 91.35% | 58.60% | 66.20% | 88.58% | 90.66% | 4.04% | 5.05% | 92.79% | 95.37% | 86.72% | 90.12% | 29.74% | 37.46% | 63.83% | 68.44% |
| **RoBERTa** | 89.58% | 92.10% | 57.12% | 64.36% | 88.83% | 91.16% | 8.28% | 13.33% | 91.08% | 94.09% | 85.36% | 87.58% | 32.16% | 41.82% | 61.57% | 67.11% |
| **ETC** | 89.55% | 91.81% | 55.56% | 60.55% | 88.07% | 90.93% | 13.33% | 13.13% | 89.71% | 93.09% | 79.76% | 84.54% | 27.92% | 37.92% | 59.60% | 65.47% |
| **TCM** | 79.28% | 86.17% | 21.36% | 52.28% | 87.22% | 90.89% | 5.05% | 5.05% | 78.87% | 86.83% | 37.00% | 62.96% | 1.08% | 21.30% | 32.15% | 38.67% |
| **XLnet** | 88.55% | 91.98% | 56.96% | 62.81% | 87.46% | 90.91% | 4.85% | 6.67% | 91.03% | 94.91% | 85.56% | 88.88% | 18.36% | 31.44% | 60.37% | 66.17% |
| **gpt2** | 89.82% | 92.31% | 54.67% | 62.40% | 88.07% | 91.41% | 4.24% | 4.04% | 88.23% | 91.33% | 80.24% | 85.04% | 22.90% | 33.14% | 59.82% | 65.62% |
| **llama2** | 91.53% | 92.94% | 56.20% | 64.45% | 87.15% | 92.16% | 5.25% | 5.45% | 94.01% | 95.18% | 90.34% | 92.26% | 28.50% | 36.00% | 66.15% | 70.35% |
| **Qwen3-8b** | 91.81% | 93.15% | 57.39% | 63.45% | 87.71% | 92.15% | 4.24% | 8.48% | 94.39% | 95.00% | 88.66% | 91.68% | 30.16% | 34.84% | 66.10% | 70.19% |
| **TF-IDF+SVM** | 87.34% | 88.82% | 41.53% | 50.07% | 87.19% | 88.92% | 23.23% | 19.19% | 85.16% | 88.03% | 60.50% | 73.90% | 17.07% | 23.59% | 51.27% | 57.00% |
| **WFS** | | | | | | | | | | | | | | | | |
| **BART** | 91.29% | 93.16% | 60.12% | 62.14% | 87.91% | 89.96% | 41.25% | 46.23% | 94.94% | 95.64% | 79.91% | 85.12% | 24.37% | 30.99% | 61.95% | 65.87% |
| **BERT** | 90.83% | 92.88% | 57.84% | 59.28% | 88.21% | 89.55% | 25.91% | 30.28% | 93.27% | 94.30% | 78.37% | 82.98% | 17.80% | 23.45% | 60.65% | 64.68% |
| **DistilBERT** | 90.73% | 93.08% | 57.14% | 59.06% | 87.03% | 89.45% | 20.04% | 29.20% | 91.93% | 93.45% | 77.44% | 80.14% | 16.83% | 23.60% | 59.30% | 63.30% |
| **DeBERTa-v3** | 91.21% | 92.53% | 60.40% | 62.41% | 88.84% | 90.31% | 22.67% | 36.63% | 95.81% | 96.35% | 84.14% | 87.81% | 20.56% | 30.16% | 61.21% | 65.16% |
| **RoBERTa** | 91.17% | 93.28% | 59.56% | 61.40% | 87.92% | 89.94% | 40.64% | 44.15% | 94.76% | 95.68% | 82.00% | 83.67% | 24.60% | 31.85% | 62.51% | 65.66% |
| **ETC** | 91.21% | 93.06% | 58.38% | 60.66% | 87.45% | 89.98% | 28.86% | 35.42% | 93.71% | 94.74% | 77.77% | 80.33% | 22.34% | 28.84% | 59.54% | 62.89% |
| **TCM** | 87.37% | 89.17% | 52.37% | 55.22% | 88.25% | 90.02% | 4.41% | 5.78% | 87.07% | 88.91% | 51.30% | 49.67% | 7.94% | 11.89% | 22.13% | 25.66% |
| **XLnet** | 91.20% | 93.04% | 58.94% | 61.24% | 87.91% | 89.81% | 37.65% | 42.99% | 94.80% | 95.70% | 81.58% | 86.65% | 17.38% | 24.44% | 61.07% | 64.48% |
| **gpt2** | 90.68% | 93.07% | 58.16% | 61.38% | 86.72% | 89.88% | 12.12% | 9.89% | 90.38% | 92.12% | 76.98% | 81.86% | 19.15% | 27.06% | 58.76% | 62.65% |
| **llama2** | 91.88% | 93.45% | 60.36% | 61.72% | 86.24% | 89.17% | 26.25% | 26.87% | 95.46% | 96.04% | 91.12% | 90.42% | 22.48% | 29.81% | 61.96% | 66.56% |
| **Qwen3-8b** | 91.57% | 93.93% | 59.46% | 60.86% | 86.44% | 88.93% | 23.22% | 19.05% | 95.45% | 95.84% | 88.23% | 90.93% | 22.16% | 28.36% | 62.94% | 66.37% |
| **TF-IDF+SVM** | 87.99% | 90.05% | 47.23% | 48.87% | 82.47% | 85.02% | 29.26% | 30.87% | 88.05% | 89.89% | 58.14% | 70.47% | 11.17% | 16.39% | 43.28% | 49.24% |

Table 16: The average accuracy results of baselines on MPS

| Model | Ag News | | Amazon | | Civil C | | Empathetic D | | IMDB | | Rotten T | | TweetEval | | Yahoo! | |
|---|---|---|---|---|---|---|---|---|---|---|---|---|---|---|---|---|
| | Imbalanced | Balanced | Imbalanced | Balanced | Imbalanced | Balanced | Imbalanced | Balanced | Imbalanced | Balanced | Imbalanced | Balanced | Imbalanced | Balanced | Imbalanced | Balanced |
| **SCS** | | | | | | | | | | | | | | | | |
| NFL-CO | 72.43% | 73.01% | 38.01% | 49.20% | 78.65% | 87.01% | 23.77% | 20.00% | 86.37% | 89.56% | 72.57% | 84.38% | 0.00% | 0.61% | 38.79% | 42.45% |
| NFL-CP | 60.54% | 61.54% | 24.65% | 35.01% | 81.24% | 82.32% | 0.00% | 0.00% | 84.49% | 84.07% | 78.40% | 78.59% | 0.00% | 0.00% | 3.53% | 9.57% |
| DRO | 74.41% | 76.41% | 42.32% | 49.90% | 78.88% | 0.00% | 22.34% | 23.89% | 0.00% | 0.00% | 0.82% | 0.61% | | | 36.47% | 41.38% |
| JTT | 64.95% | 65.81% | 30.55% | 49.19% | 74.23% | 82.08% | 0.03% | 0.00% | 86.29% | 89.06% | 74.03% | 78.10% | 0.03% | 0.00% | 31.92% | 33.22% |
| DFR | 58.56% | 58.56% | 24.28% | 32.70% | 66.20% | 72.47% | 8.51% | 13.70% | 76.76% | 81.42% | 71.35% | 75.69% | 1.75% | 1.09% | 28.23% | 31.45% |
| LLR | 70.27% | 73.87% | 27.90% | 48.23% | 70.79% | 84.82% | 3.66% | 1.89% | 83.30% | 75.48% | 76.39% | 85.07% | 0.00% | 0.00% | 36.64% | 42.24% |
| DownSample | 59.82% | 67.75% | 48.53% | 52.26% | 81.61% | 84.29% | 0.00% | 0.00% | 90.94% | 91.14% | 79.17% | 83.94% | 0.00% | 0.00% | 37.39% | 40.09% |
| **CCS** | | | | | | | | | | | | | | | | |
| NFL-CO | 84.91% | 82.12% | 44.20% | 49.13% | 76.55% | 84.89% | 0.00% | 7.22% | 88.04% | 88.76% | 78.45% | 78.80% | 1.87% | 0.25% | 46.58% | 44.39% |
| NFL-CP | 77.91% | 73.34% | 30.40% | 27.60% | 73.77% | 78.05% | 0.00% | 0.00% | 82.76% | 84.50% | 73.87% | 75.05% | 0.00% | 0.00% | 14.87% | 14.95% |
| DRO | 87.21% | 84.96% | 44.46% | 48.68% | 79.79% | 0.00% | 7.50% | 13.42% | 0.00% | 0.00% | 0.00% | 80.21% | 2.03% | 3.41% | 46.99% | 44.60% |
| JTT | 21.15% | 45.80% | 31.86% | 47.84% | 77.74% | 77.34% | 0.00% | 0.00% | 87.54% | 88.65% | 82.00% | 82.02% | 0.07% | 0.03% | 45.87% | 46.95% |
| DFR | 76.49% | 79.28% | 25.15% | 27.06% | 55.46% | 56.72% | 1.92% | 0.00% | 78.96% | 81.55% | 75.54% | 74.46% | 2.48% | 2.95% | 38.80% | 36.63% |
| LLR | 85.30% | 86.50% | 25.58% | 48.26% | 74.47% | 85.71% | 0.00% | 0.00% | 81.57% | 82.66% | 77.68% | 84.12% | 0.00% | 0.00% | 44.53% | 46.61% |
| DownSample | 82.12% | 80.14% | 50.18% | 47.46% | 81.85% | 83.53% | 0.00% | 0.00% | 89.83% | 91.99% | 79.31% | 80.55% | 1.65% | 4.13% | 45.03% | 43.64% |
| **NBS** | | | | | | | | | | | | | | | | |
| NFL-CO | 73.42% | 86.68% | 37.99% | 53.28% | 71.74% | 85.49% | 17.07% | 22.06% | 89.75% | 89.51% | 77.56% | 82.82% | 4.31% | 4.29% | 42.19% | 48.48% |
| NFL-CP | 84.32% | 84.53% | 19.24% | 21.47% | 77.43% | 80.02% | 0.00% | 0.00% | 84.88% | 83.75% | 78.18% | 80.16% | 0.00% | 0.05% | 11.26% | 16.13% |
| DRO | 71.82% | 86.52% | 43.60% | 53.21% | 79.96% | 0.00% | 20.32% | 22.03% | 0.00% | 0.00% | 2.50% | 8.24% | | | 36.49% | 46.71% |
| JTT | 76.26% | 88.32% | 33.26% | 50.73% | 76.61% | 88.54% | 0.04% | 0.01% | 90.02% | 89.29% | 75.27% | 83.99% | 0.08% | 1.91% | 36.49% | 46.71% |
| DFR | 78.63% | 80.74% | 27.28% | 26.85% | 75.47% | 77.58% | 6.62% | 10.14% | 83.05% | 82.89% | 72.03% | 74.48% | 2.26% | 6.52% | 36.26% | 36.87% |
| LLR | 71.37% | 89.54% | 30.08% | 49.95% | 77.26% | 90.54% | 0.00% | 8.70% | 86.85% | 79.87% | 72.23% | 72.69% | 0.00% | 7.71% | 37.52% | 44.73% |
| DownSample | 87.65% | 87.33% | 52.31% | 51.56% | 88.10% | 88.26% | 1.55% | 11.26% | 92.24% | 92.88% | 81.88% | 82.26% | 4.43% | 8.14% | 48.07% | 47.52% |
| **QBS** | | | | | | | | | | | | | | | | |
| NFL-CO | 81.59% | 86.48% | 18.53% | 51.68% | 73.54% | 87.11% | 0.00% | 0.00% | 79.94% | 89.49% | 70.06% | 84.98% | 0.00% | 0.00% | 36.61% | 44.52% |
| NFL-CP | 80.63% | 83.68% | 16.07% | 28.15% | 77.38% | 81.30% | 0.00% | 0.00% | 80.72% | 84.39% | 74.16% | 74.97% | 0.00% | 0.00% | 14.50% | 11.99% |
| DRO | 78.04% | 85.66% | 35.31% | 0.00% | 77.67% | 88.84% | 0.00% | 0.09% | 0.00% | 0.00% | 71.87% | 0.00% | 0.08% | 0.00% | 33.74% | 44.75% |
| JTT | 78.04% | 86.45% | 0.36% | 29.60% | 77.95% | 87.59% | 0.02% | 0.09% | 83.74% | 89.03% | 64.97% | 82.78% | 0.08% | 0.00% | 19.96% | 38.30% |
| DFR | 72.44% | 78.08% | 10.81% | 29.15% | 67.55% | 74.65% | 0.00% | 0.00% | 78.02% | 81.13% | 67.09% | 75.45% | 0.00% | 1.91% | 28.22% | 26.02% |
| LLR | 70.27% | 73.87% | 27.90% | 48.23% | 70.79% | 84.82% | 3.66% | 1.89% | 83.30% | 75.48% | 76.39% | 85.07% | 0.00% | 0.00% | 36.64% | 42.24% |
| DownSample | 85.00% | 86.03% | 47.84% | 49.03% | 88.43% | 88.15% | 0.00% | 0.00% | 91.35% | 91.54% | 79.57% | 86.23% | 0.00% | 0.00% | 40.76% | 46.76% |
| **WFS** | | | | | | | | | | | | | | | | |
| NFL-CO | 82.43% | 86.42% | 31.12% | 49.27% | 81.13% | 86.51% | 5.45% | 4.85% | 90.27% | 89.95% | 71.53% | 80.19% | 1.61% | 4.45% | 40.39% | 43.20% |
| NFL-CP | 76.76% | 81.97% | 17.32% | 34.83% | 79.24% | 81.95% | 0.00% | 0.00% | 87.47% | 85.62% | 70.70% | 76.84% | 0.00% | 0.00% | 13.39% | 17.19% |
| DRO | 83.72% | 87.98% | 37.75% | 50.02% | 83.69% | 0.00% | 12.12% | 4.85% | 0.00% | 0.00% | 0.00% | 0.00% | 0.66% | 1.61% | 37.77% | 45.44% |
| JTT | 82.73% | 88.46% | 28.02% | 48.30% | 87.12% | 89.70% | 0.04% | 0.09% | 88.49% | 90.91% | 73.08% | 74.01% | 0.04% | 0.02% | 0.00% | 13.57% |
| DFR | 78.35% | 80.86% | 20.53% | 32.85% | 73.34% | 75.42% | 3.03% | 3.03% | 81.79% | 82.90% | 68.84% | 68.37% | 1.18% | 1.90% | 22.52% | 26.93% |
| LLR | 83.96% | 86.76% | 29.13% | 49.13% | 83.43% | 86.46% | 0.00% | 0.00% | 79.02% | 88.79% | 69.30% | 80.93% | 0.00% | 0.00% | 32.27% | 45.90% |
| DownSample | 88.30% | 89.46% | 46.90% | 46.66% | 88.23% | 88.41% | 0.00% | 0.00% | 92.67% | 93.56% | 75.26% | 81.40% | 0.00% | 1.23% | 35.72% | 43.55% |

Table 17: Comparative Analysis of W-ACC Results for Seven Anti-Spurious Correlation Methods with BERT as the Backbone Model

| Model | Ag news | | Amazon | | Civil C | | Empathetic D | | IMDB | | Rotten T | | TweetEval | | Yahoo! | |
|---|---|---|---|---|---|---|---|---|---|---|---|---|---|---|---|---|
| | Imbalanced | Balanced | Imbalanced | Balanced | Imbalanced | Balanced | Imbalanced | Balanced | Imbalanced | Balanced | Imbalanced | Balanced | Imbalanced | Balanced | Imbalanced | Balanced |
| **SCS** | | | | | | | | | | | | | | | | |
| NFL-CO | 79.33% | 80.06% | 52.47% | 58.89% | 86.87% | 89.65% | 49.63% | 50.07% | 88.45% | 90.71% | 82.12% | 85.95% | 20.26% | 20.29% | 59.21% | 58.40% |
| NFL-CP | 74.66% | 74.66% | 45.23% | 50.84% | 84.78% | 85.87% | 13.69% | 13.56% | 86.94% | 88.37% | 82.51% | 82.44% | 15.73% | 11.40% | 51.56% | 52.68% |
| DRO | 82.30% | 81.74% | 55.15% | 60.06% | 86.57% | 48.86% | 52.20% | 53.51% | 46.77% | 51.94% | 49.81% | 50.19% | 31.74% | 28.35% | 58.84% | 61.09% |
| JTT | 78.11% | 79.21% | 49.66% | 60.00% | 83.62% | 84.64% | 11.17% | 18.32% | 87.87% | 90.75% | 83.26% | 83.02% | 27.63% | 27.03% | 62.72% | 62.62% |
| DFR | 69.52% | 69.80% | 43.33% | 44.22% | 70.90% | 73.63% | 33.24% | 32.56% | 78.30% | 82.78% | 76.55% | 77.26% | 9.06% | 11.28% | 51.78% | 51.58% |
| LLR | 82.02% | 83.99% | 49.90% | 60.58% | 83.58% | 86.07% | 48.70% | 51.14% | 84.94% | 85.99% | 82.48% | 86.19% | 33.96% | 34.04% | 63.57% | 63.33% |
| DownSample | 75.67% | 78.71% | 58.09% | 60.29% | 87.06% | 87.86% | 15.06% | 15.04% | 91.59% | 92.61% | 82.55% | 85.35% | 12.21% | 15.59% | 59.47% | 60.02% |
| **CCS** | | | | | | | | | | | | | | | | |
| NFL-CO | 88.42% | 87.11% | 54.31% | 56.54% | 84.88% | 87.14% | 48.68% | 48.61% | 89.75% | 90.10% | 81.91% | 82.85% | 22.24% | 21.44% | 60.78% | 60.06% |
| NFL-CP | 83.79% | 82.85% | 48.42% | 48.70% | 78.50% | 80.47% | 7.88% | 10.42% | 86.54% | 87.77% | 80.66% | 80.86% | 11.02% | 11.74% | 53.51% | 52.34% |
| DRO | 90.44% | 88.60% | 56.13% | 58.96% | 88.08% | 51.17% | 53.67% | 49.44% | 50.44% | 50.44% | 52.70% | 82.85% | 28.92% | 28.21% | 60.65% | 60.15% |
| JTT | 58.36% | 63.86% | 53.66% | 59.57% | 83.17% | 77.58% | 29.73% | 11.85% | 89.10% | 89.75% | 82.03% | 83.08% | 24.47% | 24.85% | 63.84% | 64.09% |
| DFR | 81.09% | 82.13% | 45.71% | 46.32% | 65.49% | 65.02% | 34.81% | 33.09% | 81.30% | 82.72% | 76.76% | 77.34% | 13.32% | 14.15% | 52.59% | 53.75% |
| LLR | 90.90% | 91.34% | 52.24% | 60.00% | 84.98% | 86.38% | 56.06% | 49.73% | 85.85% | 87.14% | 83.40% | 84.38% | 31.41% | 32.70% | 64.55% | 63.55% |
| DownSample | 85.53% | 84.47% | 59.80% | 60.08% | 86.10% | 86.20% | 6.98% | 5.28% | 91.53% | 92.99% | 82.58% | 83.91% | 21.51% | 21.83% | 61.34% | 61.19% |
| **NBS** | | | | | | | | | | | | | | | | |
| NFL-CO | 87.84% | 91.71% | 58.22% | 61.42% | 83.07% | 88.09% | 50.39% | 50.74% | 90.89% | 91.26% | 83.57% | 86.03% | 22.21% | 23.92% | 63.43% | 64.30% |
| NFL-CP | 89.64% | 89.89% | 45.90% | 47.28% | 83.21% | 84.40% | 11.61% | 15.64% | 87.61% | 88.15% | 80.71% | 82.70% | 9.78% | 10.36% | 58.81% | 59.55% |
| DRO | 88.21% | 92.09% | 60.66% | 62.50% | 87.10% | 51.52% | 52.12% | 52.42% | 46.33% | 48.78% | 82.17% | 85.31% | 23.71% | 26.02% | 65.37% | 68.66% |
| JTT | 87.98% | 92.40% | 58.41% | 62.96% | 85.69% | 88.88% | 6.04% | 10.93% | 90.55% | 91.07% | 80.96% | 84.08% | 14.86% | 26.10% | 65.37% | 68.66% |
| DFR | 84.91% | 85.71% | 45.43% | 47.42% | 77.40% | 78.24% | 29.44% | 30.00% | 83.30% | 83.47% | 75.80% | 75.72% | 16.24% | 17.37% | 57.71% | 58.52% |
| LLR | 87.81% | 92.73% | 58.32% | 63.50% | 86.46% | 90.92% | 47.20% | 49.86% | 87.57% | 87.98% | 81.19% | 83.44% | 25.13% | 28.96% | 65.22% | 68.69% |
| DownSample | 92.35% | 91.85% | 61.74% | 61.42% | 90.88% | 91.42% | 46.00% | | 93.27% | 93.95% | 84.23% | 84.39% | 26.51% | 28.45% | 67.06% | 67.54% |
| **QBS** | | | | | | | | | | | | | | | | |
| NFL-CO | 89.37% | 91.80% | 52.64% | 59.15% | 85.12% | 88.44% | 12.12% | 12.73% | 85.39% | 90.91% | 79.28% | 86.22% | 22.36% | 21.66% | 59.45% | 64.16% |
| NFL-CP | 88.59% | 89.90% | 48.47% | 54.07% | 81.86% | 84.07% | 6.46% | 7.68% | 85.21% | 88.24% | 79.64% | 83.58% | 8.42% | 9.40% | 56.98% | 58.41% |
| DRO | 89.02% | 91.36% | 54.56% | 20.36% | 87.74% | 90.79% | 17.37% | 18.79% | 49.29% | 52.12% | 79.48% | 47.40% | 20.98% | 24.84% | 60.27% | 64.35% |
| JTT | 89.31% | 90.92% | 40.16% | 56.10% | 86.95% | 89.41% | 3.06% | 2.09% | 86.92% | 90.09% | 76.72% | 83.48% | 0.73% | 20.98% | 60.07% | 65.84% |
| DFR | 82.38% | 83.16% | 44.23% | 46.30% | 73.69% | 77.46% | 10.61% | 12.12% | 80.07% | 83.11% | 73.95% | 77.25% | 9.45% | 11.95% | 55.25% | 55.25% |
| LLR | 82.02% | 83.99% | 49.90% | 60.58% | 83.58% | 86.07% | 48.70% | 51.14% | 84.94% | 85.99% | 82.48% | 86.19% | 33.96% | 34.04% | 63.57% | 63.33% |
| DownSample | 90.28% | 91.62% | 59.81% | 60.76% | 90.12% | 90.23% | 8.28% | 12.93% | 91.94% | 92.32% | 84.36% | 87.36% | 7.72% | 9.98% | 65.04% | 65.55% |
| **WFS** | | | | | | | | | | | | | | | | |
| NFL-CO | 89.89% | 91.23% | 55.40% | 57.73% | 85.90% | 88.17% | 40.91% | 43.73% | 91.59% | 92.04% | 76.79% | 82.33% | 18.36% | 21.64% | 60.06% | 62.76% |
| NFL-CP | 85.82% | 88.45% | 47.79% | 52.22% | 83.90% | 84.85% | 7.22% | 6.72% | 89.83% | 89.55% | 78.00% | 80.05% | 7.11% | 9.06% | 56.58% | 59.30% |
| DRO | 90.84% | 92.50% | 57.91% | 59.15% | 86.27% | 50.00% | 44.41% | 46.61% | 50.00% | 50.00% | 50.00% | 50.00% | 17.85% | 23.10% | 60.16% | 62.54% |
| JTT | 89.81% | 92.37% | 57.08% | 60.88% | 88.71% | 90.17% | 5.78% | 4.66% | 90.20% | 91.89% | 77.24% | 79.19% | 5.02% | 10.20% | 31.23% | 58.41% |
| DFR | 84.08% | 85.98% | 46.32% | 48.22% | 75.15% | 75.70% | 24.86% | 25.71% | 84.88% | 85.26% | 71.86% | 71.63% | 12.32% | 14.08% | 44.38% | 48.36% |
| LLR | 90.73% | 92.47% | 56.07% | 61.22% | 85.85% | 88.58% | 12.97% | 16.00% | 85.30% | 89.83% | 77.91% | 81.86% | 16.23% | 23.44% | 59.08% | 64.12% |
| DownSample | 91.92% | 92.96% | 58.21% | 58.68% | 89.70% | 89.73% | 6.08% | 5.63% | 93.12% | 94.24% | 80.09% | 82.88% | 14.67% | 21.74% | 61.76% | 64.35% |

Table 18: Comparative Analysis of ACC Results for Seven Anti-Spurious Correlation Methods with BERT as the Backbone Model

| Model | Ag News | | Amazon | | Civil C | | Empathetic D | | IMDB | | Rotten T | | TweetEval | | Yahoo! | |
|---|---|---|---|---|---|---|---|---|---|---|---|---|---|---|---|---|
| | Imbalanced | Balanced | Imbalanced | Balanced | Imbalanced | Balanced | Imbalanced | Balanced | Imbalanced | Balanced | Imbalanced | Balanced | Imbalanced | Balanced | Imbalanced | Balanced |
| **SCS** | | | | | | | | | | | | | | | | |
| **NFL-CO** | 70.45% | 62.70% | 29.14% | 41.14% | 64.72% | 80.90% | 9.19% | 15.65% | 84.53% | 89.30% | 70.62% | 78.96% | 0.00% | 0.00% | 41.23% | 34.24% |
| **NFL-CP** | 65.41% | 68.59% | 36.65% | 41.01% | 77.08% | 82.04% | 9.22% | 8.56% | 86.16% | 88.94% | 76.44% | 82.58% | 0.00% | 0.57% | 39.49% | 33.50% |
| **DRO** | 65.41% | 70.18% | 31.23% | 48.64% | 73.93% | 77.20% | 17.81% | 13.88% | 77.95% | 88.33% | 72.22% | 70.14% | 0.00% | 0.00% | 30.69% | 28.10% |
| **DFR** | 63.06% | 69.82% | 35.52% | 41.11% | 73.21% | 72.32% | 5.88% | 11.68% | 80.06% | 87.32% | 70.20% | 72.60% | 2.68% | 1.05% | 38.36% | 35.99% |
| **LLR** | 63.06% | 64.86% | 36.08% | 53.25% | 58.43% | 81.25% | 21.74% | 4.76% | 67.21% | 79.36% | 77.43% | 75.35% | 0.00% | 2.56% | 38.36% | 44.83% |
| **DownSample** | 69.73% | 72.29% | 55.19% | 58.66% | 83.93% | 83.93% | 29.09% | 23.05% | 92.15% | 92.81% | 84.77% | 83.61% | 15.21% | 19.17% | 49.43% | 53.53% |
| **CCS** | | | | | | | | | | | | | | | | |
| **NFL-CO** | 80.77% | 84.82% | 28.47% | 43.14% | 70.00% | 84.21% | 9.40% | 2.96% | 87.20% | 85.30% | 82.90% | 83.00% | 0.00% | 0.00% | 40.56% | 33.58% |
| **NFL-CP** | 76.10% | 82.30% | 35.61% | 38.31% | 81.57% | 80.64% | 3.47% | 3.93% | 88.00% | 85.10% | 82.51% | 79.71% | 0.00% | 0.61% | 34.47% | 35.18% |
| **DRO** | 83.02% | 78.73% | 33.33% | 34.34% | 68.86% | 71.17% | 2.22% | 4.11% | 86.03% | 86.42% | 69.40% | 71.90% | 0.19% | 1.17% | 38.77% | 34.66% |
| **DFR** | 77.61% | 78.83% | 32.25% | 34.62% | 61.34% | 71.80% | 2.94% | 0.00% | 85.95% | 88.32% | 71.03% | 73.18% | 4.12% | 6.58% | 43.73% | 45.37% |
| **LLR** | 85.28% | 87.12% | 29.46% | 48.41% | 75.53% | 80.85% | 0.00% | 16.67% | 86.08% | 82.61% | 84.12% | 86.02% | 0.00% | 2.33% | 47.66% | 48.44% |
| **DownSample** | 82.56% | 84.66% | 57.97% | 57.97% | 84.80% | 85.11% | 0.00% | 2.71% | 92.69% | 92.15% | 85.30% | 83.18% | 20.90% | 4.56% | 51.63% | 39.76% |
| **NBS** | | | | | | | | | | | | | | | | |
| **NFL-CO** | 80.77% | 88.35% | 35.18% | 46.47% | 72.02% | 86.06% | 12.63% | 18.81% | 89.35% | 89.06% | 76.91% | 81.58% | 0.00% | 3.19% | 43.28% | 43.32% |
| **NFL-CP** | 80.49% | 87.08% | 11.92% | 24.00% | 77.14% | 80.88% | 0.00% | 0.00% | 81.65% | 84.87% | 78.36% | 78.34% | 0.00% | 0.00% | 5.45% | 11.72% |
| **DRO** | 69.02% | 84.29% | 39.55% | 10.42% | 76.69% | 86.73% | 11.09% | 11.69% | 88.34% | 83.76% | 0.00% | 76.16% | 1.38% | 0.00% | 37.61% | 37.26% |
| **DFR** | 85.18% | 85.68% | 32.87% | 40.89% | 80.89% | 81.73% | 7.41% | 14.53% | 88.35% | 88.43% | 70.24% | 73.21% | 5.98% | 7.18% | 44.72% | 44.14% |
| **LLR** | 75.53% | 89.04% | 32.63% | 51.61% | 69.53% | 89.53% | 17.76% | 7.25% | 87.98% | 88.13% | 78.54% | 83.62% | 0.27% | 10.37% | 39.73% | 43.52% |
| **DownSample** | 87.54% | 88.46% | 48.75% | 53.07% | 81.38% | 85.24% | 23.63% | 15.50% | 89.65% | 90.87% | 84.56% | 84.13% | 1.47% | 3.42% | 44.49% | 46.77% |
| **QBS** | | | | | | | | | | | | | | | | |
| **NFL-CO** | 78.50% | 88.94% | 25.94% | 51.88% | 73.93% | 84.19% | 0.00% | 0.00% | 79.85% | 85.00% | 67.68% | 85.00% | 0.00% | 0.00% | 28.40% | 41.50% |
| **NFL-CP** | 82.51% | 85.25% | 8.81% | 23.68% | 74.66% | 75.48% | 0.00% | 0.00% | 73.83% | 85.74% | 73.46% | 76.30% | 0.00% | 0.00% | 6.27% | 3.87% |
| **DRO** | 76.84% | 82.97% | 19.78% | 24.68% | / | / | 0.00% | 0.00% | 83.03% | 88.01% | 66.61% | 71.75% | 0.00% | 0.49% | 37.43% | 40.64% |
| **DFR** | 81.25% | 83.42% | 12.05% | 24.68% | 74.18% | 80.24% | 0.00% | 0.00% | 83.03% | 88.01% | 66.61% | 71.75% | 0.00% | 0.49% | 37.43% | 40.64% |
| **LLR** | 63.06% | 64.86% | 36.08% | 53.25% | 58.43% | 81.25% | 21.74% | 4.76% | 67.21% | 79.36% | 77.43% | 75.35% | 0.00% | 2.56% | 38.36% | 44.83% |
| **DownSample** | 85.09% | 86.08% | 48.96% | 53.24% | 84.83% | 84.18% | 0.00% | 0.00% | 92.58% | 93.32% | 82.57% | 85.90% | 0.00% | 5.33% | 39.86% | 44.08% |
| **WFS** | | | | | | | | | | | | | | | | |
| **NFL-CO** | 83.45% | 87.71% | 33.58% | 46.48% | 80.12% | 84.58% | 5.45% | 5.45% | 91.00% | 91.22% | 73.49% | 80.65% | 1.19% | 1.36% | 39.54% | 44.32% |
| **NFL-CP** | 76.69% | 80.80% | 8.41% | 26.33% | 79.94% | 81.46% | 0.00% | 0.00% | 86.63% | 87.43% | 73.77% | 76.74% | 0.00% | 0.00% | 12.67% | 13.44% |
| **DRO** | 81.99% | 84.47% | 20.79% | 35.37% | 79.94% | 83.95% | 2.42% | 1.82% | 85.82% | 87.35% | 67.44% | 66.28% | 3.55% | 5.21% | 30.02% | 38.47% |
| **DFR** | 84.38% | 86.42% | 23.45% | 37.36% | 74.17% | 77.76% | 9.09% | 7.58% | 85.82% | 87.35% | 67.44% | 66.28% | 3.55% | 5.21% | 30.02% | 38.47% |
| **LLR** | 83.96% | 84.47% | 27.74% | 51.87% | 78.74% | 83.51% | 12.12% | 6.06% | 84.35% | 89.67% | 80.00% | 79.07% | 0.00% | 1.90% | 29.48% | 43.43% |
| **DownSample** | 88.64% | 87.78% | 43.74% | 39.14% | 87.18% | 86.86% | 0.00% | 0.00% | 91.18% | 92.39% | 80.00% | 80.47% | 2.55% | 6.03% | 45.41% | 43.47% |

Table 19: Comparative Analysis of W-ACC Results for Six Anti-Spurious Correlation Methods with Qwen as the Backbone Model

| Model | Ag News | | Amazon | | Civil C | | Empathetic D | | IMDB | | Rotten T | | TweetEval | | Yahoo! | |
|---|---|---|---|---|---|---|---|---|---|---|---|---|---|---|---|---|
| | Imbalanced | Balanced | Imbalanced | Balanced | Imbalanced | Balanced | Imbalanced | Balanced | Imbalanced | Balanced | Imbalanced | Balanced | Imbalanced | Balanced | Imbalanced | Balanced |
| **SCS** | | | | | | | | | | | | | | | | |
| **NFL-CO** | 77.19% | 77.64% | 45.94% | 53.63% | 79.40% | 84.58% | 42.07% | 45.43% | 87.80% | 90.30% | 81.56% | 84.42% | 37.07% | 38.00% | 60.61% | 60.51% |
| **NFL-CP** | 75.39% | 76.24% | 50.26% | 54.14% | 83.98% | 85.27% | 39.89% | 38.17% | 90.15% | 91.32% | 81.84% | 84.96% | 31.19% | 34.00% | 56.94% | 56.25% |
| **DRO** | 77.08% | 79.72% | 46.43% | 53.77% | 82.39% | 82.29% | 47.23% | 47.17% | 82.03% | 89.79% | 79.66% | 78.37% | 37.95% | 32.71% | 55.18% | 54.55% |
| **DFR** | 73.60% | 75.42% | 49.53% | 51.12% | 77.11% | 75.37% | 37.89% | 36.98% | 81.60% | 88.66% | 74.25% | 74.07% | 27.89% | 25.02% | 59.42% | 58.61% |
| **LLR** | 79.49% | 80.06% | 51.29% | 61.00% | 78.61% | 84.08% | 52.08% | 53.17% | 80.01% | 87.74% | 83.72% | 85.66% | 41.43% | 44.98% | 62.80% | 61.67% |
| **DownSample** | 73.03% | 78.09% | 58.35% | 60.50% | 85.57% | 85.07% | 39.27% | 35.14% | 92.31% | 93.25% | 85.13% | 86.87% | 28.48% | 32.54% | 57.16% | 57.84% |
| **CCS** | | | | | | | | | | | | | | | | |
| **NFL-CO** | 88.01% | 88.62% | 47.91% | 51.90% | 81.60% | 85.54% | 45.82% | 38.55% | 88.60% | 87.88% | 85.08% | 85.04% | 32.89% | 32.84% | 58.13% | 56.24% |
| **NFL-CP** | 84.98% | 87.68% | 54.80% | 53.37% | 86.48% | 85.45% | 41.48% | 36.53% | 90.26% | 89.56% | 85.04% | 83.71% | 27.22% | 24.55% | 57.61% | 55.18% |
| **DRO** | 88.55% | 87.06% | 50.86% | 54.86% | 79.81% | 78.69% | 50.53% | 44.99% | 88.37% | 90.19% | 81.03% | 53.79% | 31.19% | 28.27% | 57.37% | 55.54% |
| **DFR** | 85.64% | 85.25% | 52.09% | 53.78% | 69.48% | 73.00% | 38.16% | 37.79% | 87.51% | 89.14% | 74.90% | 75.88% | 26.26% | 26.03% | 60.68% | 61.64% |
| **LLR** | 90.68% | 89.47% | 52.34% | 59.07% | 85.45% | 84.98% | 56.06% | 52.98% | 88.08% | 87.58% | 86.33% | 86.33% | 40.44% | 39.64% | 63.81% | 63.33% |
| **DownSample** | 88.09% | 88.22% | 59.79% | 59.10% | 85.73% | 85.45% | 15.52% | 17.90% | 93.11% | 93.43% | 86.33% | 87.07% | 30.30% | 29.24% | 61.03% | 58.00% |
| **NBS** | | | | | | | | | | | | | | | | |
| **NFL-CO** | 88.25% | 92.87% | 56.91% | 59.14% | 83.03% | 87.56% | 49.83% | 50.62% | 90.12% | 90.62% | 83.01% | 85.55% | 22.82% | 23.27% | 62.90% | 65.51% |
| **NFL-CP** | 89.75% | 91.35% | 45.67% | 47.03% | 82.66% | 84.13% | 6.93% | 7.22% | 86.33% | 86.93% | 81.59% | 82.28% | 7.67% | 8.15% | 54.83% | 58.11% |
| **DRO** | 86.58% | 90.55% | 56.56% | 43.99% | 84.16% | 88.74% | 44.00% | 47.26% | 90.96% | 88.08% | 45.61% | 79.52% | 24.67% | 19.73% | 59.09% | 61.13% |
| **DFR** | 89.41% | 89.65% | 53.00% | 55.47% | 81.51% | 81.85% | 34.75% | 36.63% | 88.69% | 89.14% | 74.84% | 74.28% | 24.93% | 25.21% | 66.20% | 64.57% |
| **LLR** | 88.72% | 92.78% | 59.40% | 63.08% | 82.73% | 90.08% | 51.97% | 54.05% | 88.80% | 89.27% | 83.60% | 86.09% | 31.27% | 34.13% | 65.33% | 67.64% |
| **DownSample** | 92.98% | 92.26% | 62.27% | 63.08% | 87.10% | 88.21% | 50.49% | 52.98% | 92.04% | 92.34% | 86.21% | 86.82% | 28.19% | 28.77% | 64.93% | 64.72% |
| **QBS** | | | | | | | | | | | | | | | | |
| **NFL-CO** | 89.61% | 92.54% | 54.59% | 59.66% | 84.98% | 86.92% | 17.58% | 13.33% | 84.66% | 90.42% | 78.04% | 86.50% | 21.32% | 22.34% | 58.53% | 61.75% |
| **NFL-CP** | 89.54% | 90.02% | 48.07% | 51.64% | 79.98% | 81.75% | 4.24% | 7.68% | 80.90% | 87.89% | 79.32% | 83.72% | 11.66% | 7.12% | 52.63% | 55.85% |
| **DRO** | 86.82% | 90.15% | 50.07% | 51.60% | / | / | 9.49% | 24.24% | 84.40% | 90.18% | 76.00% | 80.16% | 27.54% | 28.90% | 55.35% | 60.75% |
| **DFR** | 89.31% | 89.42% | 48.17% | 49.77% | 79.19% | 82.23% | 16.67% | 22.22% | 85.12% | 88.35% | 70.90% | 73.10% | 25.85% | 22.00% | 62.05% | 63.53% |
| **LLR** | 79.49% | 80.06% | 51.29% | 61.00% | 78.61% | 84.08% | 52.08% | 53.17% | 80.01% | 87.74% | 83.72% | 85.66% | 41.43% | 44.98% | 62.80% | 61.67% |
| **DownSample** | 90.56% | 91.40% | 59.33% | 61.09% | 87.52% | 86.77% | 22.63% | 16.36% | 93.40% | 93.92% | 84.22% | 86.60% | 30.94% | 29.40% | 64.80% | 66.27% |
| **WFS** | | | | | | | | | | | | | | | | |
| **NFL-CO** | 90.73% | 91.64% | 56.33% | 58.53% | 85.41% | 87.19% | 41.02% | 40.72% | 92.65% | 91.96% | 77.02% | 81.77% | 18.73% | 20.14% | 58.85% | 61.75% |
| **NFL-CP** | 87.93% | 88.65% | 48.82% | 51.66% | 83.27% | 83.60% | 4.79% | 4.26% | 89.06% | 89.54% | 78.84% | 80.70% | 5.96% | 7.69% | 50.79% | 53.59% |
| **DRO** | 89.21% | 90.93% | 47.66% | 50.63% | 83.85% | 86.48% | 36.99% | 33.98% | 63.72% | 72.06% | 76.70% | 50.00% | 19.44% | 20.39% | 51.24% | 56.73% |
| **DFR** | 88.85% | 90.27% | 50.21% | 50.92% | 77.57% | 80.22% | 31.44% | 32.43% | 88.14% | 89.43% | 71.05% | 68.14% | 19.70% | 21.23% | 56.96% | 62.92% |
| **LLR** | 90.47% | 92.13% | 56.80% | 60.82% | 84.57% | 87.33% | 45.17% | 45.08% | 87.62% | 90.34% | 82.09% | 83.02% | 23.84% | 29.15% | 54.92% | 64.64% |
| **DownSample** | 92.03% | 91.59% | 57.51% | 56.43% | 88.29% | 87.97% | 23.94% | 22.78% | 93.71% | 93.43% | 81.86% | 83.86% | 23.66% | 25.85% | 62.30% | 63.38% |

Table 20: Comparative Analysis of ACC Results for Six Anti-Spurious Correlation Methods with Qwen as the Backbone Model

