# OpenReview forum: "MPS: A Multi-Perspective Benchmark For Assessing Spurious Correlations in Text Classification"
_ICLR.cc/2026/Conference — Submitted to ICLR 2026_

### Official Review · Reviewer_AtUH · 2025-10-21

**Soundness:** 3
**Presentation:** 2
**Contribution:** 2
**Rating:** 4
**Confidence:** 4

**Summary:**

This paper introduces a multi-perspective benchmark for systematically evaluating how text classifiers rely on spurious correlations, covering five types of spurious cues -- sentence-level, core-word, negation-based, question-based, and word-frequency biases -- across eight popular NLP datasets. The authors propose two new evaluation metrics derived from worst-group accuracy, to measure robustness degradation and potential improvement. Through extensive experiments on eight widely used datasets and multiple model families, they find that existing models and mitigation methods struggle to handle all types of spurious correlations, with question-based biases emerging as particularly challenging.

**Strengths:**

(1) The introduction of the Question-Based Spurious correlation (QBS) category highlights a previously underexplored yet impactful source of model bias.

(2) The proposed δ and Δ metrics, based on worst-group accuracy, allow fine-grained quantification of robustness degradation and potential improvement.

(3) The paper evaluates a wide range of models (traditional, MLMs, LLMs) and robustness methods, revealing consistent vulnerabilities across approaches.

**Weaknesses:**

(1) The idea of benchmarking spurious correlation learning in text classification is not new -- it builds directly on prior frameworks (e.g., Shortcut Maze) without introducing fundamentally new insights. The paper frames itself as the “first comprehensive benchmark”, which is inaccurate given prior large-scale studies.

(2) Unlike earlier work that manipulates correlation intensity (e.g., λ in Shortcut Maze), MPS uses only binary Balanced vs. Imbalanced splits, reducing granularity in robustness measurement.

(3) Although claiming to be comprehensive, the paper omits four spurious types -- synonym, register, author style, and concept correlation -- that were defined in Shortcut Maze.

(4) The new question-based spurious type lacks clear operational boundaries. some questions may genuinely reflect task semantics rather than bias.

(5) The main paper would benefit from a summary figure or table defining, explaining, and exemplifying each spurious type.

(6) Because MPS constructs artificially balanced and imbalanced splits, it may not fully capture naturally occurring spurious correlations present in real-world data.

(7) The redistribution used to create these splits may unintentionally alter label distributions and subgroup sizes, introducing confounding factors beyond the intended spurious correlations. Consequently, performance gaps between splits might reflect class imbalance or distributional shifts, rather than true model sensitivity to spurious features.

**Questions:**

(1) Could the authors explicitly articulate how MPS advances beyond prior benchmarks such as Shortcut Maze? In particular, how do the proposed five spurious types and eight datasets provide new analytical insights rather than simply broader coverage? A clearer statement of conceptual novelty would help calibrate expectations.

(2) In constructing Balanced and Imbalanced splits, how do the authors control for changes in label distributions, subgroup sizes, or topic diversity that could introduce confounding effects?

(3) The human study is an interesting addition—could the authors elaborate on participant demographics, task instructions, and agreement rates?

(4) Since MPS focuses solely on text classification, do the authors foresee extending the benchmark to generation or reasoning tasks?

---

> ### Author Response · Authors · 2025-11-14
> **Response to Reviewer AtUH**
>
> We sincerely appreciate all the suggestions you have provided regarding our work. We have thoroughly reviewed each comment and will now address them in detail, hoping to resolve any concerns you may have. We believe all your recommendations have positively influenced the improvement of our study.
>
> To weakness 1:We acknowledge that our initial explanation may not have fully captured the comprehensive nature of our work. Thank you for this valuable clarification. While our benchmark encompasses all four types of spurious correlations discussed in the paper, they are not parallel but rather hierarchical in relationship. More significantly, this paper introduces a novel evaluation framework for assessing spurious correlations in text classification. Our primary contribution lies in fundamentally reimagining the conventional W-ACC computation methodology. We have developed an innovative W-ACC calculation approach and, building upon this foundation, proposed two more granular metrics for quantifying the influence of spurious correlations. Please refer to Appendix A.5 for detailed formulations (we recognize this was an oversight in our initial submission - we failed to adequately highlight these methodological innovations in the main text due to space constraints, and the highly mathematical presentation in the appendix may have obscured the comprehensive nature of our improvements). We have now incorporated a clear summary of these contributions in the main body of the paper.
>
> To weakness 2:The λ parameter in the referenced paper is a data-generation hyperparameter that controls the strength of spurious correlations between shortcut features (e.g., specific words, styles, or concepts) and target labels across different categorical groups. In contrast, our work not only reformulates the W-ACC computation but fundamentally addresses the "insufficient subpopulation samples" issue that compromises measurement authenticity in NLP spurious correlation assessment. In traditional text classification datasets, the assessment of worst-group accuracy often becomes unfeasible when certain subpopulations contain insufficient samples. Our refined W-ACC metric effectively addresses this limitation. By adopting a dataset construction methodology analogous to ours, any dataset can now quantitatively measure its degree of spurious correlations without being constrained by subpopulation sample sizes.These two studies diverge in their primary focus—our approach demonstrates greater granularity and comprehensiveness while employing a more profound measurement architecture.
>
> To weakness 3:All four types of spurious correlations identified in Shortcut Maze are incorporated within our benchmark, albeit under different nomenclature. For instance, what they term "synonym" corresponds to our proposed CCS-type spurious correlation. Furthermore, their conceptual spurious correlations align with our second major category of spurious associations, which we have further refined into SCS and CCS subcategories with greater granularity.We acknowledge that our previous explanations may have lacked clarity, and we sincerely apologize for any confusion this may have caused.
>
> To weakness 4:This issue warrants profound consideration. While conceptual spurious correlations have increasingly become a research focus, we have sought to identify novel research directions within statistical spurious correlations. Through systematic analysis, we discovered that punctuation marks may exert distinctive influences on model learning, which inspired our investigation into interrogative and negative semantics from a symbolic perspective. Given that spurious correlations related to negation semantics have been extensively studied in NLI tasks, we pioneered the exploration of interrogative spurious correlations (QBS). Experimental results demonstrate that QBS indeed exists and broadly impairs model learning.The reviewer's question raises a particularly valuable point, and we should consider providing further demonstration of the significance of this spurious correlation.
>
> To weakness 5:This content has been incorporated at the top of page two in the main text.

---

> ### Author Response · Authors · 2025-11-14
> **Response to Reviewer AtUH （Part 2）**
>
> To weakness 6:It is inherently challenging for any dataset or benchmark to comprehensively encompass the full spectrum of spurious correlations present in real-world scenarios. By definition, spurious correlations represent false associations that can neither be entirely eliminated nor exhaustively detected—we can only strive to collect as many representative instances as possible. The fundamental objective of our benchmark is to systematically categorize various types of spurious correlations at a conceptual level, analyze their distinct characteristics in influencing model learning, and thereby provide guidance for future research directions.Therefore, comprehensive detection of real-world spurious correlations may not represent the primary focus of our research.
>
> To weakness 7:We sincerely apologize for any potential misunderstanding. We have now explicitly clarified our improvements to the metric system in both the main text and Appendix A.5 of the paper. Specifically, we have refined the W-ACC calculation method to eliminate its dependency on subpopulation sample sizes, thereby addressing a critical challenge in spurious correlation measurement within natural language processing.
>
> To Question 1:We would like to reiterate the fundamental structure of our dataset. Our benchmark comprises eight distinct datasets, which differ from conventional approaches in that while we annotate subpopulation labels (referred to as conceptual or statistical labels in the main text, varying by spurious correlation type), we systematically restructure the label distribution (specifically, classification labels) accordingly. Through this process, various features within the dataset establish direct, strong associations with certain labels—creating precisely the spurious correlations we examine.
> The key innovation lies in how these spurious correlations are no longer dependent on traditional subpopulation definitions but are instead reflected through classification labels. Building on this principle, our MPS dataset directly employs classification labels as groups for worst-group accuracy measurement—an approach that would be methodologically unsound in other contexts but is justified by our specific construction methodology. The validation for this approach is detailed in Appendix A.5.
> Subsequently, we introduce two more granular metrics that enable finer-grained analysis of different spurious correlation types, advancing beyond conventional measurement limitations.
>
> To Question 2:The relevant content has been updated in Appendix A.8 of the revised manuscript. Regarding the confounding factors you mentioned, our elaborated explanation on "the redefinition of W-ACC" should help address your concerns.
>
> To Question 3:Thank you for your valuable feedback regarding the insufficient documentation of our annotation process in the initial submission. We have now comprehensively detailed the human annotation procedure in Appendix A.3 of the revised manuscript.
>
> To Question 4:The principal innovation of our benchmark lies in its comprehensive coverage and analytical depth. It encompasses an extensive range of spurious correlation types while maintaining a high-level conceptual perspective, rather than focusing on specific granular variants (as we contend that spurious correlations can be infinitely subdivided—even proper names could constitute a category). Consequently, our work concentrates on systematizing spurious correlations at a macroscopic level. We believe other research domains could similarly benefit from summarizing and elucidating spurious correlations from higher conceptual vantage points, rather than remaining constrained by narrowly defined subtypes.
>
> We extend our sincere appreciation for all your valuable comments and suggestions.

---

> ### Author Response · Authors · 2025-11-17
> **Response to Reviewer AtUH （Part 3）**
>
> We summarize here the distinctions between our benchmark MPS and prior studies, which we hope will address your question.
>
> 1.The λ parameter employed in the referenced paper serves as a tool for dataset construction, yet the benchmark relies solely on accuracy as the evaluation metric—an approach we deem insufficient. In spurious correlation research, the most critical metric, worst-group accuracy, was not utilized. Although Δ was adopted as a substitute, this leads to inadequate granularity in their analysis.
> In contrast, our benchmark not only refines the conventional worst-group accuracy metric but also introduces two secondary metrics—δ and ∆—derived from optimized worst-group accuracy and overall accuracy. These metrics offer significantly finer granularity and capture the impact of spurious correlations at a deeper level.
>
> 2.Our benchmark not only expands in terms of dataset scale and model evaluation scope, but its core strength lies in the significantly broadened coverage of domains, the increased number of classification categories, and the heightened difficulty level. It also incorporates a more diverse range of model types for measurement, including evaluations of more methods specifically designed for studying spurious correlations.
> More importantly, the five categories of spurious correlations we propose are conceptualized at a macro level, representing more abstract types of spurious correlations. Analyzing spurious correlations at this level helps elucidate the specific ways in which different types exert their influence. In contrast, the spurious correlation types discussed in the referenced paper are fewer in number and more concrete in nature.
> We believe that spurious correlation types are virtually limitless, and studying only a limited set of individual types makes it difficult to grasp their overall characteristics. Therefore, our investigation of multiple categories of spurious correlations at a higher level of abstraction may offer new perspectives for future research.

---

> > ### Comment · Reviewer_AtUH · 2025-11-26
> >
> > Thank the authors for providing additional explanations. However, some of my concerns are not fully addressed.
> >
> > (1) The benchmark’s shortcut coverage is still not complete: several well-studied shortcut types from prior work (e.g., synonym, stylistic/register cues, author-style, and concept-correlation variants) are not faithfully reproduced in MPS. The authors’ assertion that these categories merely “rename” into SCS/CCS is unsupported. These shortcut mechanisms are semantically and operationally distinct and require different annotation pipelines. Furthermore, the contribution remains incremental rather than novel: MPS follows the same core paradigm as Shortcut Maze: annotating spurious attributes and constructing controlled splits, without introducing a fundamentally new conceptual framework or experimental dimension. The new metrics are straightforward differences derived from W-ACC, not innovations in benchmark design or methodology.
> >
> > (2) The response does not explain why removing a continuous strength parameter (lambda in Shortcut Maze) does not reduce analytical granularity. The discussion about subgroup sample sizes is orthogonal to the core issue. The authors provide no justification or evidence showing that two discrete levels can adequately probe model sensitivity to varying spurious correlation strengths.
> >
> > (3) The response does not address the core issue of confounding factors introduced during dataset construction. Redefining W-ACC does not prevent label-distribution shifts, topical drift, or semantic distortions caused by forced re-labeling. The authors provide no quantitative analysis showing that redistribution isolates spurious features rather than altering the underlying task. Thus, the validity of comparisons between Balanced and Imbalanced splits remains questionable.
> >
> > (4) The authors do not clarify when interrogative markers represent spurious cues rather than genuine semantics.
> > Pointing to punctuation or symbolic reasoning does not establish that QBS constitutes a spurious association rather than a task-relevant feature.

---

> > > ### Author Response · Authors · 2025-11-26
> > > **Response to Reviewer AtUH （Round 2）**
> > >
> > > Thank you very much for the reviewer's feedback. We have provided some clarifications regarding these comments, hoping that they will help enhance your understanding of the issues addressed in our work.
> > >
> > > To （1）:Our work, MPS, primarily investigates textual classification at a relatively abstract content level, encompassing comprehensive categories, with the main objective of examining the impact of various types of spurious correlations on model learning in text classification. The Shortcuts Maze dataset and its associated evaluation metrics do not adequately address this aspect. Here, we provide a summary of the four types of spurious correlations you raised: Synonyms can be understood as a categorization of words sharing similar meanings. For instance, "happy" and "bliss" are synonyms, both expressing a positive emotional attitude. This falls under the CCS-type spurious correlations we proposed, where words in the text belong to the same conceptual group. Stylistic and authorial features cannot be summarized by individual words; the manifestation of style or register requires the comprehensive expression of the entire text segment. This characteristic aligns with SCS-type spurious correlations, which pertain to the core semantic concepts conveyed by the entire sentence. Finally, the notion of concept-correlation variants is the primary focus of the ACL 2024 paper "Explore Spurious Correlations at the Concept Level in Language Models for Text Classification", i.e., concept-level spurious correlations. In summary, we contend that the Shortcuts Maze paper primarily investigated concept-level spurious correlations and their derivative variants, without extending the research further into other types of spurious correlations in text classification.
> > >
> > > In the field of spurious correlation evaluation, there exists a prevalent research bottleneck: the issue of sample size in worst-group accuracy. This problem significantly constrains further investigation into various types of spurious correlations, as insufficient samples in minority subgroups make it difficult for metrics to reflect such correlations adequately. This also serves as a key reason why current spurious correlation benchmarks (e.g., Shortcut Maze) exhibit fragmented and limited coverage of spurious correlation types. Our benchmark construction process eliminates the dependency on subgroup labels. By directly associating worst-group performance with task labels, we enable all types of spurious correlations to be captured by the "worst-group accuracy" metric based on task labels. In other words, through our benchmark construction methodology, any type of spurious correlation can be evaluated using our defined W-ACC, free from the constraints of subgroup sample size limitations.We believe this represents a fundamental distinction between our benchmark and other comparable benchmarks in the field.
> > >
> > > To （2）:While continuous strength parameters could indeed provide more granular insights into the magnitude of specific spurious correlations, this approach does not align with the primary research direction of our benchmark. The extensive categorization of spurious correlation types across multiple datasets already enables comprehensive assessment of various spurious correlation patterns. The elimination of the hyperparameter λ can be compensated for through comparative analyses across these diverse datasets. This methodological choice offers a distinct advantage: it demonstrates the pervasive existence of the five macro-categories of spurious correlations we have identified across multiple text classification domains.
> > >
> > > To （3）:We acknowledge the validity of this concern and will address it in subsequent revisions by incorporating additional experimental evidence to provide further clarification. We appreciate the reviewer's thorough consideration of this matter.
> > >
> > > To （4）:This pertains to the fundamental definition of spurious correlations. By definition, any type of spurious correlation could potentially serve as valid reasoning evidence rather than being spurious in specific contexts. For instance, the NBS-type spurious correlation - where the presence of negation semantics could indeed constitute crucial reasoning clues - has nevertheless been consistently treated as spurious in prior research. This classification stems from the observation that models may fail to learn the actual semantic function of negation in text, instead merely capturing the superficial phenomenon of its "presence or absence," consequently leading to the negative impact of spurious correlations on model learning. The same rationale applies to QBS-type spurious correlations.

---

### Official Review · Reviewer_3pFQ · 2025-10-28

**Soundness:** 3
**Presentation:** 2
**Contribution:** 2
**Rating:** 4
**Confidence:** 2

**Summary:**

The paper introduces MPS, a benchmark built from 8 text classification datasets to evaluate model robustness under five spurious correlation types, namely SCS, CCS, NBS, QBS, and WFS. It defines two analysis metrics, which are δ (change in worst-group accuracy when balancing a given shortcut type) and Δ (headroom from worst-group under shortcut to overall accuracy after balancing), to quantify each shortcut’s impact. Across models and mitigation methods, results show no single method is robust across all five types, and the newly defined QBS is particularly challenging.

**Strengths:**

The paper’s core strength lies in its scope and standardization. A unified benchmark across eight datasets and five shortcut types), with paired balanced/imbalanced splits, and clear robustness metrics (W-ACC, δ/Δ) that isolate and quantify shortcut effects. Empirically, it’s broad and careful, covering classic baselines, pretrained encoders, multiple mitigation families, and LLM backbones, plus a human reference, yielding actionable findings.

**Weaknesses:**

1. The distinction between sentence-level concepts (SCS) and core-word concepts (CCS) is not fully transparent from the main text.
2. It is not clear how static attributes of SCS are selected and the relationship with their corresponding labels.
3. Some ambiguous parts, like the usage of W-ACC and δ, are listed in the following Questions section.

**Questions:**

1. Table 7’s descriptions do not make the distinction between SCS and CCS sufficiently clear. Could you provide precise definitions and concrete, dataset-specific examples for each to clarify the difference?
2. When δ < 0 (e.g., strong results on an imbalanced split), how do you conclude that the model exploits spurious correlations rather than demonstrating genuine understanding?
3. How to explain the large negative δ of TCM on Ag News in Table 1?
4. Could the low W-ACC simply because of small sample sizes in certain (𝑦, 𝑎) groups, rather than true vulnerability to the spurious attribute.

---

> ### Author Response · Authors · 2025-11-14
> **Response to Reviewer 3pFQ**
>
> We sincerely appreciate all the suggestions you have provided regarding our work. We have thoroughly reviewed each comment and will now address them in detail, hoping to resolve any concerns you may have. We believe all your recommendations have positively influenced the improvement of our study.
>
> To weakness 1:Thank you for your suggestion. We have identified the issue you raised regarding the somewhat unclear distinction between SCS and CCS. In response, we have relocated the explanatory table from the appendix to the main body of the text and further refined its content. The updated version is now available in the revised manuscript.  In essence, SCS refers to the overarching theme of the entire text segment, whereas CCS denotes the thematic essence derived from the core vocabulary of the text. Both represent conceptual abstractions, albeit at differing levels of granularity.
>
> To weakness 2:As illustrated in the appendix (page thirty-three of the revised manuscript), the selected spurious correlations labels for SCS-type spurious correlations across various datasets are presented. For both SCS and CCS-type spurious correlations, we first obtained conceptual labels, then performed distribution reconstruction within the subpopulations formed by these conceptual labels. This process enables the benchmark to incorporate spurious correlations and equips it with the capability to evaluate models' resistance to various types of spurious correlations.
>
> To weakness 3:To Question part.
>
> To Question 1:Similarly to the response provided for the first limitation, an explanatory table has been incorporated into the main text.
>
> To Question 2:It is possible that our explanation has caused some confusion. We intend to convey that when a model exhibits a negative δ metric, it indicates that the model has correctly acquired certain features. This form of "learning" represents an intriguing phenomenon: it may be interpreted as genuine knowledge acquisition, or alternatively, as the assimilation of superficial patterns rather than substantive understanding. This is analogous to students' tendency to select longer options in multiple-choice questions—a strategy that may indeed improve their probability of answering correctly, without necessarily reflecting deep comprehension.What can be ascertained is that the model has acquired beneficial factors, though it remains undetermined whether it has genuinely assimilated substantive knowledge — analogous to a student correctly answering a multiple-choice question without necessarily demonstrating deep comprehension.
>
> To Question 3:Based on the computational methodology of this metric, it can be inferred that the worst-group accuracy achieved by TCM on the imbalanced data subset significantly surpasses that obtained on the balanced subset, resulting in a substantially negative value (-19.98%). We posit two plausible explanations, with the latter being more probable:
> 1. It is highly improbable that the model has learned this particular type of spurious correlation and successfully applied it to the test set, let alone generalized such spurious associations to other parameter layers, thereby achieving exceptional worst-group performance. This scenario remains unlikely, as evidenced by the scarcity of such instances across all datasets.
> 2. The observed spurious correlations may possess a certain degree of deceptive nature. As indicated in the table, other models also demonstrate a tendency toward positive δ values (highlighted in green), suggesting a systematic pattern worth further investigation.
>
>
> To Question 4:As elaborated in our Appendix, the W-ACC metric employed in our study deviates from conventional formulations. The specific computational methodology and theoretical justification are detailed in Appendix A.5 of the new manuscript. Our benchmark utilizes an optimized W-ACC calculation that addresses the inherent limitations of subpopulation sample scarcity typically encountered in traditional NLP approaches for quantifying spurious correlations. This constitutes a fundamental design rationale underlying our distribution reconstruction paradigm. By systematically aligning subpopulation characteristics with task labels through data distribution restructuring, our framework enables the quantification of spurious correlation strength via worst-group accuracy measurement across task-labeled categories. It is crucial to emphasize that this measurement approach is intrinsically tied to our benchmark's architectural specificity and cannot be directly extrapolated to other experimental contexts.
>
> We extend our sincere appreciation for all your valuable comments and suggestions.

---

### Official Review · Reviewer_Tbyw · 2025-11-02

**Soundness:** 2
**Presentation:** 3
**Contribution:** 2
**Rating:** 4
**Confidence:** 4

**Summary:**

This paper introduces MPS (Multi-Perspective Benchmark For Assessing Spurious Correlations in Text Classification), a comprehensive benchmark for evaluating model robustness against spurious correlations in text classification. The authors systematically categorize spurious correlations into five types: SCS (Sentence-level Concept Spurious), CCS (Core-word Concept Spurious), NBS (Negation-Based Spurious), QBS (Question-Based Spurious), and WFS (Word-Frequency-based Spurious). Using 8 widely-used datasets, they create 40 dataset variants and conduct extensive evaluations of various models (MLMs, LLMs, traditional ML) and anti-spurious correlation methods. The work includes human performance comparisons and introduces novel metrics ($\delta$ and $\Delta$) to quantify spurious correlation effects.

**Strengths:**

1. The five-type taxonomy of spurious correlations (SCS, CCS, NBS, QBS, WFS) provides a clear framework for analysis. QBS (Question-Based Spurious correlations) appears to be a genuinely new contribution that existing methods struggle with.
2. Testing 12 models and 5 anti-spurious correlation methods across 40 dataset variants represents substantial empirical work. The inclusion of human performance baselines (Table 4) provides valuable context and reveals interesting patterns (e.g., humans outperform models on emotional tasks but under-perform on contextual classification).

**Weaknesses:**

1. The 10% manual verification rate is too low for a benchmark paper. The paper should provide: (a) inter-annotator agreement scores, (b) error analysis of LLM mistakes, (c) validation on a larger sample or full validation on at least one dataset. Sample sizes, annotator selection criteria, training procedures, and inter-annotator agreement are not provided. This makes it difficult to assess the reliability of human performance claims.
2. The Imbalanced/Balanced construction procedure needs algorithmic detail. What constitutes "overwhelmingly dominant"? What are the exact class distribution targets? Without this, reproduction is difficult.
3. The paper doesn't ablate key design choices. For example: How sensitive are results to the choice of 6 spurious attributes for SCS/CCS? What about the training epoch selection strategy?
4. The paper evaluates existing methods but doesn't propose new solutions. While benchmark papers need not introduce new methods, some guidance on promising directions would strengthen the contribution.

**Questions:**

1. Can you provide pseudocode or precise algorithmic descriptions for constructing Imbalanced and Balanced subsets? What specific thresholds define "overwhelmingly dominant"?
2. Beyond the 10% validation, can you provide error analysis? What types of mistakes does Llama 3.1 make in concept annotation? How do error rates vary across datasets?
3. What are the sample sizes for human evaluation? How many annotators? What was their expertise level? What was inter-annotator agreement?
4. Have you tested whether models trained to be robust on one spurious correlation type show improved robustness on other types?

---

> ### Author Response · Authors · 2025-11-14
> **Response to Reviewer Tbyw**
>
> We sincerely appreciate all the suggestions you have provided regarding our work. We have thoroughly reviewed each comment and will now address them in detail, hoping to resolve any concerns you may have. We believe all your recommendations have positively influenced the improvement of our study.
>
> To weakness 1:We performed a 10% stratified sampling validation across all dataset subgroups. As the accuracy rate (as denoted in the appendix) reached 99%—indicating that 99% of labels conformed to our annotation standards—we deemed further resource-intensive validation unnecessary. However, we acknowledge the validity of your suggestion and, if required, will conduct more extensive verification of dataset labels. Additionally, we have incorporated the requested validator demographics and detailed verification protocols into Appendix A.2 of the revised manuscript, expanding upon the original A.2 with comprehensive procedural descriptions and validation metrics.
>
> To weakness 2:This issue has been a subject of our prolonged consideration. Your suggestion is highly pertinent and valuable. In the first revision, we included comprehensive distribution visualizations and have now further augmented this by adding a detailed explanation of the "overwhelmingly dominant" phenomenon in Appendix A.8. Specifically, through systematic experimentation, we identified several distribution patterns that demonstrably exacerbate the impact of spurious correlations on model training. These empirically validated patterns form the foundational basis for our construction of the MPS framework.
>
> To weakness 3:The issue raised falls partially outside the scope of our current research, which primarily investigates how conceptual spurious correlations affect overall model learning and proposes a reformed evaluation framework. It should be emphasized that our newly introduced worst-group accuracy metric, along with two additional metrics, effectively assesses spurious correlations at the categorical level—constituting the core focus of our study. However, these metrics are not designed to evaluate the individual impact of specific instantiations within each spurious correlation type, as such granular analysis was not part of our original research design. Our work concentrates on identifying broader trends across five categories of spurious correlations in mainstream architectures and various mitigation methods, rather than examining individual instantiations. Nevertheless, your suggestion is highly insightful, and we will consider incorporating it into our future research endeavors.
>
> To weakness 4:A comprehensive summary of the paper's conclusions, along with recommendations for future technological development, has been incorporated into Section 6 of the main text and can be located therein.
>
> To Question 1:We have refined the description of the construction process in the main text. The core issue regarding the definition of "overwhelmingly dominant" has been explicitly addressed in the appendix, corresponding to the section "To weakness 2".
>
> To Question 2:As addressed in "To weakness 1".
>
> To Question 3:We acknowledge that this aspect was not adequately addressed in the initial submission. We thank the reviewer for this meticulous observation and have now compiled comprehensive documentation of the human text annotation process, which is presented in Appendix A.3.（First, we selected 10% of the total data from each pair of Balanced and Imbalanced datasets (corresponding to the 5 types of spurious correlations) across the aforementioned 8 MPS datasets, based on the same label distribution ratio, to constitute the human testing dataset.A total of 10 human annotators were recruited to conduct labeling tests for text classification, aiming to evaluate human performance in terms of robustness to spurious correlations in text classification tasks. All annotators demonstrated proficiency in reading and comprehending English text, satisfying the requirements for performing text classification. Each human annotator was required to directly perform the text classification task. When a text segment was displayed on the screen, the annotator was required to select the corresponding text classification label for that text. Furthermore, to mitigate errors arising from randomness, we performed 3 tests per dataset and computed the average to obtain the final metric results.）
>
> To Question 4:This is a particularly insightful question. We have conducted additional experiments to address this issue, which are detailed in Appendix A.7. Notably, these experiments further substantiate our key finding that 'spurious correlations exhibit independence—effectively mitigating one category of spurious correlations does not meaningfully contribute to resolving other types.' This outcome significantly enriches the conclusions of our benchmark study.
>
> We extend our sincere appreciation for all your valuable comments and suggestions.

---

### Author Response · Authors · 2025-11-14
**The collective comments from the three reviewers.**

We sincerely thank the three reviewers for their thorough and insightful comments on our manuscript. While we will provide point-by-point responses to each reviewer separately, we would first like to clarify several key aspects that may not have been sufficiently articulated in the initial version of the paper. These points are elaborated in detail below.

1.In the first round of experiments, we evaluated several anti-spurious-correlation loss functions using BERT and Qwen as backbones. However, for the Qwen backbone, measurements were only conducted for the NFL loss function family and downsampling. In this revision, we have expanded the benchmark by incorporating DFR and LLR loss functions to ensure a more comprehensive evaluation. Additionally, we have completed the measurements for the Qwen backbone across these methods, with the exception of JTT, which is not compatible with the Qwen architecture.

2.Our benchmark employs a novel evaluation framework for assessing spurious correlations. The definition of W-ACC (Worst-group Accuracy) was initially presented in the appendix of the first-round submission, where its heavily symbolic notation may have caused some ambiguity. This version rectifies that issue. It is important to note that our implementation of worst-group accuracy is not the conventional one; rather, it represents an optimized version of the metric. The detailed definition can be found in Appendix A.4 of the paper.Since our variant constitutes an optimization that remains fundamentally grounded in the group-based framework, we continue to refer to it as W-ACC.

3.For the evaluation of standard pre-trained models on the benchmark, while the results for SCS-type spurious correlations are presented in the main text, the experimental outcomes for the remaining four types of spurious correlations are provided in the appendix. It should be noted that the color intensity in the table cells is designed to be interpreted through vertical comparison.

---

### Author Response · Authors · 2025-12-01
**A Comprehensive Introduction to Writing Rebuttals in Academic Peer Review**

$Key·Revisions·1$

We have added validation data for the labels generated by MPS (Appendix A.2). This supplementary dataset demonstrates that the labels produced by the large-scale model exhibit a high degree of reliability, thereby addressing several reviewers' concerns regarding the credibility of MPS-generated labels. Additionally, we have elaborated on the description of the human annotation experiment (Appendix A.3), which investigates human performance in handling spurious correlations within text classification tasks. Through the first round of rebuttal, we have refined the experimental setup descriptions, thereby enhancing the credibility of the experimental data presented in the paper.

$Key·Revisions·2$

In response to the reviewers' comments, we have incorporated several comparative experiments, the results of which are included in the appendix of the revised manuscript. These experiments include cross-validation of spurious correlations (Appendix  A.7), which investigates whether addressing a single type of spurious correlation can effectively enhance robustness against other types. The experimental findings indicate that this is not the case, indirectly demonstrating that the five categories of spurious correlations constructed in our dataset are independent. This, to some extent, suggests that MPS does not involve "altering the underlying task" as raised by the reviewers.

$Key·Revisions·3$

To demonstrate the rationale behind our W-ACC (worst-group accuracy) calculation method and to justify why W-ACC computed under our benchmark—MPS—holds significant value for evaluating spurious correlations, we have designed new experiments and incorporated both the experimental results and corresponding analysis into the revised version of the paper (Appendix A5.2). The experiments confirm that our newly proposed W-ACC calculation method effectively mitigates the dependency on subgroup sample sizes inherent in traditional worst-group accuracy metrics, while robustly reflecting the extent of a model’s resilience against spurious correlations.

$Key·Revisions·4$

The reviewer referenced a paper titled "Navigating the Shortcut Maze: A Comprehensive Analysis of Shortcut Learning in Text Classification by Language Models," suggesting that our work lacks substantial innovation. In response, we outline the distinctions between the two studies as follows:

1. The spurious correlation types examined in that paper are relatively specific, whereas our MPS framework synthesizes the spurious correlations prevalent in current research into five categories, each studied separately (including a newly proposed type, QBS spurious correlation). Experimental results demonstrate that QBS spurious correlation is not a "necessary learning feature" as mentioned by the reviewer; instead, it substantially impairs model learning performance, thereby exacerbating spurious correlation effects. Our research objective is to analyze the varying degrees of impact that different types of spurious correlations exert on model learning. In contrast, the spurious correlation types covered in that paper do not encompass all categories (e.g., NBS type, i.e., negation-based spurious correlations).

2. The paper employs hyperparameter $\lambda$ to adjust the strength of spurious correlations in the dataset and performs comparisons using test sets. However, it utilizes only three datasets, which cannot adequately represent diverse scenarios of varying intensity and varied linguistic corpora. Although our dataset does not incorporate a spurious correlation strength adjustment metric, it conducts comparative analyses across five variants of spurious correlations on eight distinct datasets, each paired with corresponding training sets.

3. The dataset released in that paper does not include subgroup labels, making it incompatible with traditional spurious correlation research methodologies. This is also one of the reasons why the paper employs a limited number of baselines.

4. A key innovation of our work lies in the dataset construction process and the novel calculation of W-ACC. Through our construction method, we establish spurious correlations between subgroups and task labels, thereby enabling the transfer of W-ACC computation to a task-label-based group calculation. The rationale and superiority of this calculation approach are detailed in Appendix A.5. In contrast, the sole metric used in that paper to evaluate spurious correlation strength is $\triangle$. However, when accuracy levels are inconsistent, the utility of $\triangle$ for assessing spurious correlations is considerably limited.


We have incorporated the aforementioned comparative discussion into the related work section (Appendix A.9).

---

> ### Author Response · Authors · 2025-12-01
> **Add**
>
> Furthermore, we have introduced two metrics, $\delta$ and $\Delta$, calculated from two distinct training datasets, to assess the degree of impact of spurious correlations. Contrary to the reviewer's suggestion that they are merely simple, basic computations, these metrics are analytically substantiated. As demonstrated in our paper's analysis, different approaches for addressing spurious correlations exhibit varying degrees of learning capacity for different types of spurious patterns. In certain scenarios, 'spurious correlations' can be effectively learned and leveraged by specific methods, a phenomenon precisely captured and quantified by our metrics $\delta$ and $\Delta$. Our evaluation framework has yielded several findings of guiding significance for future research: some features may act as spurious correlations for a model, yet with effective utilization (e.g., via certain spurious correlation mitigation techniques), these spurious features can be transformed into constructive elements for feature learning.

---

### Meta-Review · Area_Chair_SEHv · 2026-01-07

**Summary:**

The paper introduces MPS, a benchmark built from 8 text classification datasets to evaluate model robustness under five spurious correlation types. Through extensive experiments on eight widely used datasets and multiple model families, they find that existing models and mitigation methods struggle to handle all types of spurious correlations, with question-based biases emerging as particularly challenging. All three reviewers are leaning towards rejection, with reasonable concerns that were not adequately addressed in the rebuttal phase. Therefore, I recommend rejection of this paper. I encourage the authors to revise the paper and resubmit it to future conferences.

**Reviewer Concerns:**

Reviewer Tbyw:

W1: insufficient manual verification. Partially addressed, while the authors still conduct manual verification on a small subset of data.
W2: Data construction details. Fully addressed.
W3: Ablation study: Partially addressed.
W4: New solution: Not addressed.

Reviewer 3pFQ raised questions on some technical details and presentation issues, which seems to be fully addressed by the authors.

Reviewer AtUH raised concerns on the novelty, benchmark settings, missing spurious types, presentations, etc. As stated by Reviewer AtUH, the authors  provided some clarifications but did not address these concerns adequately.

**Reviewer Scores:**

Given that some concerns remain unresolved, I think the reviewer Tbyw and AtUH's scores remain the same. I think reviewer 3pFQ's score may be slightly increased after the clarifications.

---

### Decision · Program_Chairs · 2026-01-26

Reject